# Membrane curvature initiates Cdc42-FBP17-N-WASP clustering and actin nucleation

Kexin Zhu[1,6], Xiangfu Guo[2,6], Aravind Chandrasekaran[3,6], Xinwen Miao[2], Padmini Rangamani 🆔[3,4✉],
Wenting Zhao 🆔[2,5✉] & Yansong Miao 🆔[1,5✉]

## Abstract

**The architecture of actin networks at the cell surface is regulated by local membrane topology. However, how actin nucleation can respond sensitively to the degree of membrane curvature remains incompletely understood. Using nanolithography to precisely control local membrane curvature, we reconstituted the dynamic interplay of the tri-component Cdc42/FBP17/N-WASP system on a series of deformed membrane sites, resulting in differential actin nucleation. We found that high-curvature sensing is primarily mediated by FBP17 through its intrinsic BAR-domain activity, which then induces the hierarchical assembly of FBP17/N-WASP clusters to activate N-WASP in synergy with Cdc42. This nucleation boost is fine-tuned by modulating the FBP17-to-N-WASP stoichiometry within multivalent macromolecular assemblies according to local curvature radii. At lower-curvature regions, Cdc42 enhances basal FBP17 recruitment to the membrane, enabling detection of shallow curvatures and initiating actin polymerization before high-curvature effects dominate. This establishes a dynamic, curvature radius-dependent cooperativity that links geometric cues to the regulation of actin polymerization, highlighting their interplay in coordinating membrane and actin morphodynamics during complex cellular processes.**

**Keywords** Actin Polymerization; Membrane Curvature; Multivalent Interaction
**Subject Categories** Cell Adhesion, Polarity & Cytoskeleton; Membranes & Trafficking

## Introduction

N-WASP's multivalent and multicomponent interactions enable its activation through various signaling pathways to drive actin polymerization spatiotemporally. Among these, Rho family GTPases play a crucial role in linking WASP family proteins to the Arp2/3 complex for actin assembly and organization across all eukaryotic cells (Lee et al, 2015; Machacek et al, 2009; Marston et al, 2020; Noh et al, 2022; Pardo-Pastor et al, 2018; Yang et al, 2016). Despite these well-established mechanisms regulating actin assembly in response to stimuli, the molecular processes driving localized actin polymerization and organization at the subcellular level during morphodynamics remain unclear. Dynamic membrane curvature underpins vital cellular functions, orchestrating complex biophysical and biochemical activities (Al-Aghbar et al, 2022; Cail and Drubin, 2023; Lu et al, 2022; Yoshida et al, 2018). Membrane shape alterations are intricately connected to the quick remodeling of the actin cytoskeleton, offering spatial and temporal governance of associated biological functions (Brunetti et al, 2022; Gaertner et al, 2022; Ledoux et al, 2023; Lou et al, 2019; McMahon and Gallop, 2005; Su et al, 2020; Wu et al, 2018; Zhao et al, 2017). Proteins sensitive to membrane curvature respond dynamically to nanoscale membrane deformations, setting off intricate lipid-protein interactions at these sites. The mechanisms by which signal-triggered biomolecular cascades are locally intensified on the membrane and influence actin remodeling remain elusive.

The BAR (Bin/Amphiphysin/Rvs) domain-containing superfamily proteins provide an attractive template for studying the interplay between membrane curvature and curvature-sensitive biochemical reactions. BAR domain proteins are known for their curvature-sensing and -shaping capabilities in cellular membranes, forming crescent dimers with lipid affinity (Bhatia et al, 2009; Jin et al, 2022; Peter et al, 2004). These proteins, including subfamilies like BAR, N-BAR, F-BAR, and I-BAR, are central to processes such as endocytosis, membrane trafficking, and cell division (Liu et al, 2015; Qualmann et al, 2011; Ren et al, 2006; Salzer et al, 2017). F-BAR proteins such as FNBP1L (TOCA-1), FBP17(TOCA-2), CIP4(TOCA-3), and other BAR proteins, including PACSIN, SNX9, amphiphysin, endophilin and IRSp53 were shown to interact with actin nucleator and nucleation-promoting factors (NPFs) and affect actin polymerization (Feng et al, 2022; Ho et al, 2004; Otsuki et al, 2003; Prehoda, 2000; Shin et al, 2007; Suetsugu and Gautreau, 2012; Takano et al, 2008; Yamada et al, 2009). In addition, actin cytoskeleton-generated forces are also known to facilitate membrane morphogenesis locally (Almeida-Souza et al, 2018; Carman and Dominguez, 2018; Graziano et al, 2014;

[1]School of Biological Sciences, NTU, 60 Nanyang Drive, 637551 Singapore, Singapore. [2]School of Chemistry, Chemical Engineering and Biotechnology, NTU, 70 Nanyang Drive, Singapore 637457, Singapore. [3]Department of Mechanical and Aerospace Engineering (MAE), University of California, San Diego, 9500 Gilman Drive MC 0411, La Jolla, CA 92093-0411, USA. [4]Department of Pharmacology, School of Medicine, University of California, San Diego, 9500 Gilman Drive MC 0636, La Jolla, CA 92093-0636, USA. [5]Institute for Digital Molecular Analytics and Science (IDMxS), NTU, 59 Nanyang Drive, Singapore 636921, Singapore. [6]These authors contributed equally: Kexin Zhu, Xiangfu Guo, Aravind Chandrasekaran. ✉E-mail: prangamani@ucsd.edu; wtzhao@ntu.edu.sg; yansongm@ntu.edu.sg

Mcdonald et al, 2015; Suetsugu, 2009; Takano et al, 2008; Tsujita et al, 2006; Wu et al, 2018). The direct role of membrane sculpting in triggering actin remodeling or perpetuating actin assembly around curved regions remains unclear, as decoupling the processes of curvature formation and actin polymerization is challenging.

The recent development of nanotopography manipulation technologies, including nanopillars, nanobars, nanoridges, and nanobeads, each with a feature diameter of less than 500 nm, has allowed for exquisite control of substrate curvature (Bettinger et al, 2009; Ledoux et al, 2023; Lou et al, 2019; Zhao et al, 2017). Such curvature control has demonstrated the selective recruitment of Wiskott-Aldrich syndrome protein (WASP) family proteins as a function of curvature (Brunetti et al, 2022; Gaertner et al, 2022; Lou et al, 2019). N-WASP is a membrane-bound actin nucleation-promoting factor in an auto-inhibited state. Activation occurs when it undergoes a conformational shift induced by its interaction with the small GTPase Cdc42 (Rohatgi et al, 2000). Recent evidence suggests that N-WASP activation may also stem from multi-component nanoclustering, which was facilitated by its intrinsically disordered region (IDR)-mediated multivalent interactions (Case et al, 2019; Harmon et al, 2017; Li et al, 2012; Su et al, 2016). However, the role of multivalent interactions in controlling downstream actin signaling is poorly understood.

This study deciphered the spatiotemporal regulation of actin network formation at curved membrane sites with an FBP17-defined range of nanoscale radius. We reconstituted the curvature-dependent actin polymerization on a nanochip-supported lipid bilayer with membrane curvature precisely defined using electron-beam lithography. We discovered the step-by-step cascade reactions leading to curvature-mediated actin polymerization using mathematical modeling and quantitative biochemical and biophysical analysis. Our findings indicate that, within Cdc42-regulated overarching actin assembly, FBP17's curvature-sensing function triggers the local nanoclustering of FBP17/N-WASP complexes at an optimal stoichiometry in a curvature-dependent manner, thereby enhancing the local activation of N-WASP for on-site actin nucleation. Activated Cdc42 enhances FBP17 membrane recruitment at an earlier stage of curvature progression by enabling sensing at lower-curvature regions, preceding high-curvature-driven FBP17-N-WASP clustering for actin nucleation. Our study reveals a mechanism in which nanoscale curvature precisely directs protein recruitment and macromolecular nanoclustering to locally trigger actin nucleation, coordinated and guided by Cdc42 in a curvature radius-dependent manner. This work uncovers the intricate, dynamic interaction between membrane curvature and the biochemical cascade of actin polymerization fundamental to various cellular functions.

# Results

## The curvature radius-dependent recruitment of N-WASP requires FBP17

First, we established a minimal curvature-sensing system to investigate how FBP17 and N-WASP respond to nanoscale membrane deformations. We generated supported lipid bilayers (SLBs) on quartz nanobar arrays fabricated by electron-beam lithography, in which the bar ends were defined with radii ranging from 100 to 500 nm, and the bar height is 600 nm (Fig. 1A–C). The bilayers spread continuously across the structures and maintained high fluidity, as confirmed by rapid fluorescence recovery after photobleaching (FRAP) (Fig. EV1A,B). Previously, similar nanoarrays demonstrated the recruitment of curvature-sensing proteins seen both in living cell systems (Lou et al, 2019; Mu et al, 2022; Zhao et al, 2017) and in vitro on SLB (Miao et al, 2022; Su et al, 2020). To probe protein distribution, we introduced fluorescently labeled recombinant proteins of nucleation-promoting factor (NPF) of Arp2/3 N-WASP (unless specifically stated, all the N-WASP used in our study were in an auto-inhibited state) and F-BAR family FBP17 and examined them over SLB on nanobar arrays. (Fig. EV1C). To quantify curvature sensitivity, we averaged the fluorescent intensity of AF488-N-WASP and AF647-FBP17 over 50 nanobars and normalized it by the lipid bilayer signal to account for surface area differences among different-sized nanobars. While the mean SLB signal was evenly distributed on the surfaces of nanobar, FBP17 and N-WASP exhibited an ends-enriched distribution dependent on the curvature radius (Fig. 1D). The end density of FBP17 signal increases on sharper nanobars (<500 nm) (Fig. 1E), consistent with its established curvature sensing ranges (Lou et al, 2019; Su et al, 2020). Interestingly, N-WASP exhibited an even more prominent curvature response within the same regions (Fig. 1E,F). On the contrary, N-WASP alone does not exhibit curvature-dependent recruitment (Fig. 1G,H), suggesting that curvature sensing by N-WASP emerges only in the context of FBP17 scaffolding.

## High membrane curvature promotes N-WASP binding by tuning (N-WASP: FBP17) bound stoichiometry

We next sought to investigate how changes in stoichiometry might regulate the formation of the FBP17/N-WASP complex on varying curvatures (Fig. EV2A). We first titrated N-WASP across a range of concentrations and molar ratios with FBP17 on flat SLBs (Fig. EV2B–D). These assays revealed that clustering became prominent at higher N-WASP concentrations, and the F: N ratio of 4:1 produced the most robust enrichment (Fig. EV2A). Then, we adopted this ratio and titrated N-WASP across a range of concentrations with FBP17 on nanobar-patterned SLBs (Fig. 2A–F). We quantified the density of FBP17 and N-WASP at the curved bar ends and normalized these measurements to lipid channel signals. As the curvature increased, both FBP17 (Fig. 2A,B) and N-WASP (Fig. 2C,D) exhibited greater accumulation at the curved ends. We observed that N-WASP accumulation is particularly sensitive to changes in curvature radius (Fig. 2A,C). This suggests a hierarchical recruitment mechanism, where N-WASP's accumulation is contingent on FBP17's primary detection of membrane curvature. Consistent with the above (Fig. 1G,H), N-WASP no longer exhibited end-accumulation and showed less membrane association across the different concentrations without FBP17 (Fig. 2E,F).

We next examined how curvature and protein abundance together influence the progressive assembly of the FBP17–N-WASP complex. We quantified the N/F ratio, which reflects the relative number of N-WASP molecules associated with each FBP17 molecule. Our results showed that the N/F ratio increases with membrane curvature under all the protein concentration

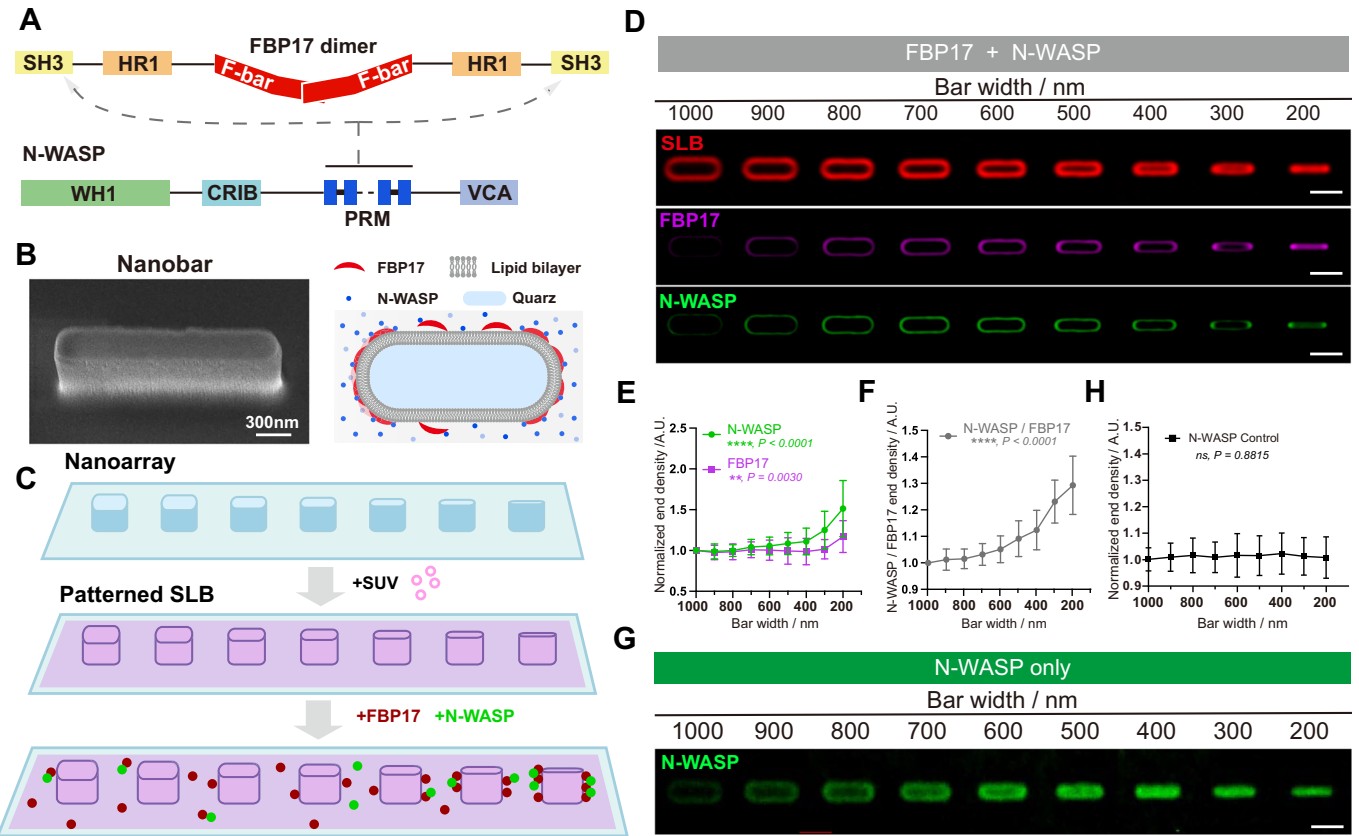

**Figure 1. Understanding the curvature sorting of FBP17 and N-WASP through nano-patterning and spatial simulations.**

(A) A domain diagram of FBP17 and N-WASP. The arrows indicate the interaction between SH3 domain of the dimeric FBP17 and the polyproline-rich region of N-WASP. (B) Left: Scanning electron microscopy image of a single nanobar. Tilted 30°. Scale bar: 300 nm. Right: A representation illustrating the in vitro reconstitution of FBP17-mediated N-WASP recruitment at membrane curvatures. (C) Schematic workflow illustration of SLB patterning and protein binding on nanobar arrays. SLBs were formed by fusion of SUVs, followed by incubation with fluorescently labeled proteins (AF647-FBP17 and AF488-N-WASP). (D) Averaged confocal images of signal intensity showing curvature responses of SLB (red), with additional 200 nM of AF647-FBP17 (magenta), or additional 50 nM of AF488-N-WASP (green), respectively, on top of nanobars of 200–1000 nm width. Scale bar, 2 μm. (E) Quantification of normalized nanobar-end density of FBP17 (magenta) and N-WASP (green) in (D). $N = 11$ averaged nanobars, each point represents mean ± SD. Linear regression was performed using the reciprocal of bar width (1/width) as a measure of curvature. Both proteins showed significant curvature sensitivity (N-WASP: $P < 0.0001$; FBP17: $P = 0.0030$), and the slopes differed significantly ($F_{(1,14)} = 58.98$, $P < 0.0001$). (F) Ratio of N-WASP end density to FBP17 end density at nanobars of 200–1000 nm width in (E). $N = 11$ averaged nanobars, each point represents mean ± SEM. Linear regression was performed using the reciprocal of bar width (1/width) to represent curvature. The ratio of N-WASP / FBP17 end density showed a highly significant positive correlation with curvature (slope = 88.4 ± 5.3, $P < 0.0001$). (G) Averaged confocal images showing curvature responses of 50 nM of AF488-N-WASP on top of nanobars of 200–1000 nm width. Scale bar, 2 μm. (H) Quantification of normalized N-WASP end density in (G). $N = 15$ averaged nanobars, each point represents mean ± SD from over 50 nanobars. Linear regression was performed using the reciprocal of bar width (1/width) to represent curvature. The slope was not significantly different from zero (slope = 0.28 ± 1.84, $P = 0.8815$). Source data are available online for this figure.

conditions we examined (Fig. 2G), indicating that higher curvature promotes more N-WASP recruitment to FBP17. Consistent with this, the slope of N/F increase—the extent of the local stoichiometry shift across curvature—was modulated by protein concentration (Fig. 2G). This also revealed an optimal range, where at ~20 nM N-WASP with 80 nM FBP17, curvature produced the strongest stoichiometric response, whereas above these concentrations, the responsiveness of the N/F ratio to curvature decreases (Fig. 2H). However, on a 2D flat membrane system, N-WASP did not exhibit a saturation state and continued accumulating over the same stoichiometry range with FBP17 (Fig. 2I). The results indicate that both N-WASP recruitment and the local stoichiometry of FBP17–N-WASP complexes are regulated by the curvature radius.

## Cdc42 orchestrates curvature-mediated N-WASP recruitment and activation

Next, we investigated the functional relationship between membrane curvature and dynamic actin remodeling in living cells. Previous studies have shown that F-actin in U2OS cells exhibits greater accumulation at high membrane curvatures (Lou et al, 2019). To dissect the curvature-guided initiation of actin polymerization and network organization, we first treated U2OS cells with Latrunculin A (Lat A) to depolymerize the F-actin and then monitored F-actin recovery by following the Lat A washout (Fig. 3A). Before LatA treatment, Lifeact-mApple labeled F-actin bundles were observed in cells on flat surfaces outside of curvature zones and also highlighted F-actin clusters at the end of

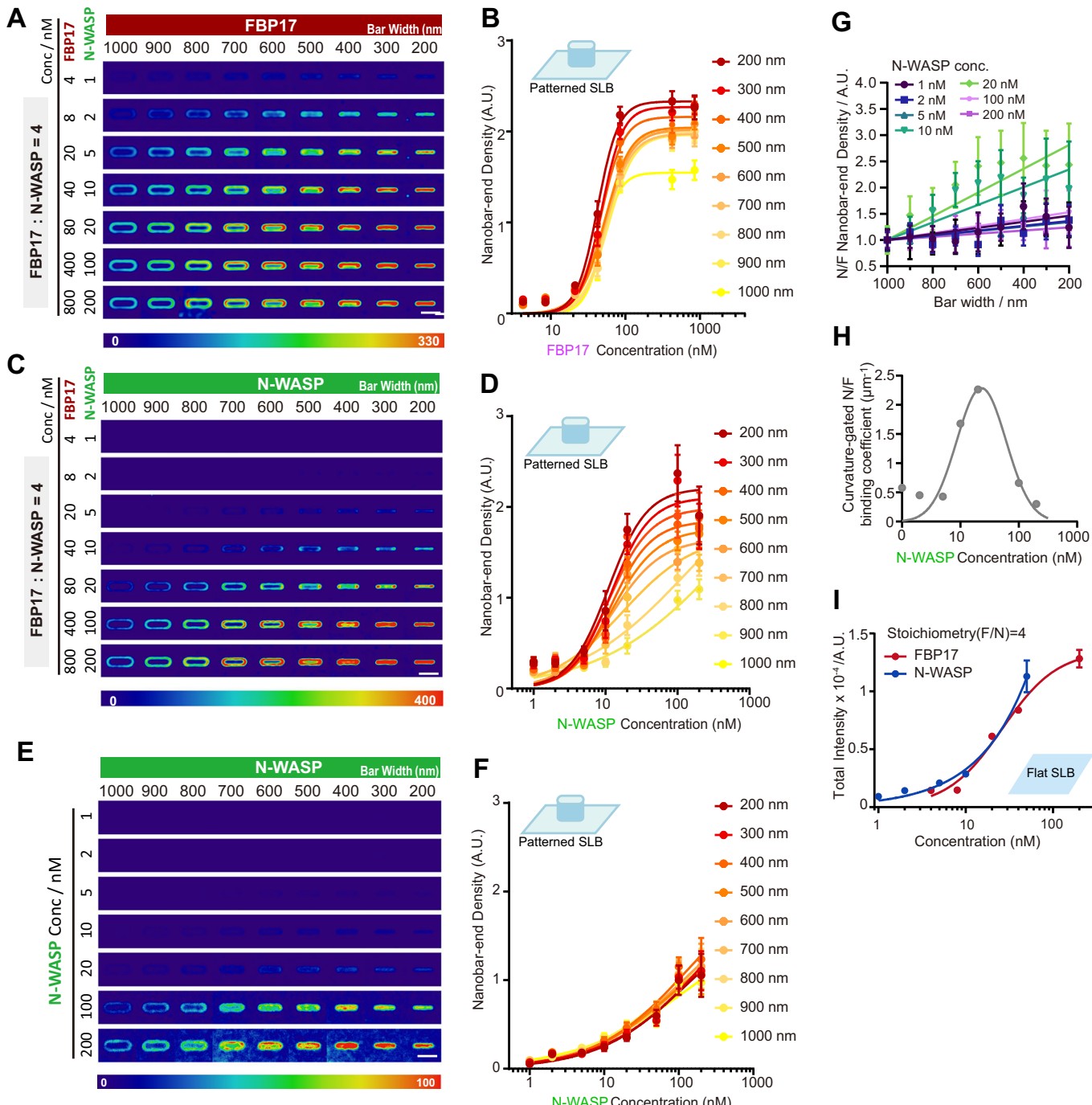

high-curvature regions above the nanobars (Figs. 3B and EV3A, left panels). If we slowly depolymerize F-actin by treating cells at a non-toxic low dose of LatA (100 nM), F-actin signals were no longer visible at the curved ends (Fig. EV3A, middle panels). When repolymerizing the actin network upon Lat A washout, we observed a pronounced F-actin recovery around the nano-pattern in a radius-dependent manner (Figs. 3B–D and EV3A, right panels) with smaller radii (higher curvature) showing greater F-actin enrichment, indicating efficient onsite nucleation of the actin cytoskeleton at the high curvature regions.

To explore how curvature mediates local F-actin assembly, we first examined the role of the well-studied N-WASP activator, Cdc42, in curvature-mediated actin nucleation. We treated cells with ML141, a specific Cdc42 inhibitor (Surviladze et al, 2010), at concentrations of 20 µM and 50 µM (Ledoux et al, 2023). Fluorescence microscopy analysis of N-WASP recruitment and activation at membrane curvatures induced by nanobar structures showed that 20 µM ML141 partially compromised actin polymerization at curvature sites (Fig. 3E,F). Despite this partial inhibition in F-actin accumulation, N-WASP remained robustly localized to

Figure 2. Curvature radius-dependent N-WASP and FBP17 distribution and condensation.

(A) Averaged confocal images of 4, 8, 20, 40, 80, 400, 800 nM AF647-FBP17 (with 1, 2, 5, 10, 20, 100, 200 μM AF488-N-WASP under the molar ratio of F:N = 4:1) on the lipid bilayer on top of nanobars of 200–1000 nm width. Scale bar, 2 μm. (B) Plot of nanobar-end density of AF647-FBP17 in (A) as a function of concentration. Lines are binding curves fitted with the Hill equation. Sample sizes for conditions from low to high concentrations were $N = 16, 17, 14, 10, 11, 11$, and 10 averaged nanobars, respectively. Each data point represents the mean ± SEM. (C) Averaged confocal images of 1, 2, 5, 10, 20, 100, 200 nM AF488-N-WASP (with 4, 8, 20, 40, 80, 400, 800 μM AF647-FBP17 under the molar ratio of F:N = 4:1) on the lipid bilayer on top of nanobars of 200–1000 nm width. Scale bar, 2 μm. (D) Plot of nanobar-end density of AF488-N-WASP in (C) as a function of concentration. Lines are binding curves fitted with the Hill equation. Sample sizes for conditions from low to high concentrations were $N = 16, 17, 14, 10, 11, 11$, and 10 averaged nanobars, respectively. Each data point represents the mean ± SEM. (E) Averaged confocal images of 1, 2, 5, 10, 20, 100, 200 nM AF488-N-WASP control on the lipid bilayer on top of nanobars of 200–1000 nm width. Scale bar, 2 μm. (F) Plot of nanobar-end density of AF488-N-WASP in (E) as a function of concentration. Lines are binding curves fitted with the Hill equation. Sample sizes for conditions from low to high concentrations were $N = 12, 14, 18, 16, 18$, 13, and 15 averaged nanobars, respectively. Each data point represents the mean ± SEM. (G) Normalized N-WASP end density divided by FBP17 end density at nanobars of 200–1000 nm width under the stoichiometry N/F = 1/4. Sample sizes for conditions from low to high concentrations were $N = 16, 17, 14, 10, 11, 11$, and 10 averaged nanobars, respectively. Each data point represents the mean ± SEM. (H) Curvature responsive coefficient plot (slope of N-WASP/FBP17 ratio as a function of concentration). Lines are binding curves fitted with the Gaussian equation. Each point represents the mean. (I) Plot of single particle total intensity of 4, 8, 20, 40, 200 nM AF647-FBP17 (red) with 1, 2, 5, 10, 50 nM AF488-N-WASP (blue) under the molar ratio of F:N = 4:1 on flat lipid bilayer (10% POPS + 89.5% POPC + 0.5% Rhodamine-PE). Each point represents mean ± SEM, $N = 7447, 22777, 28608, 25589$, and 13419 single particles for FBP17 and 1236, 734, 573, 1145, and 658 single particles for N-WASP. See also Fig. EV2. Source data are available online for this figure.

the ends of nanobars, similar to the DMSO control (Fig. 3E,G), indicating that curvature-guided local N-WASP accumulation for actin assembly persists with reduced Cdc42 activity. In contrast, treatment with 50 μM ML141 significantly reduced both N-WASP recruitment and actin enrichment at nanobar ends (Fig. 3E,G). This impaired localization at higher ML141 concentrations suggests that Cdc42 might also be involved in the association of N-WASP with BAR proteins at curvature sites through tricomponent weak multivalent interactions, which were reported by NMR-based intermolecular interaction work (Watson et al, 2016). We next examined FBP17 distribution under ML141 treatment. FBP17 retained its ends enrichment at nanobar ends under both inhibitor conditions, confirming its intrinsic curvature-sensing ability (Fig. 3H,I). However, overall membrane association of FBP17 was reduced, correlating with diminished actin accumulation (Fig. 3J,K). Thus, Cdc42 does not regulate FBP17 curvature sensing but its basal membrane recruitment, thereby reinforcing curvature-guided actin assembly. These findings also indicate that Cdc42 is involved in curvature-guided actin assembly through two roles: directly activating N-WASP and orchestrating the interactions locally between N-WASP and FBP17 via controlling surface dose. Such Cdc42 regulation of N-WASP-FBP17 accumulation at curvature sites is independent of F-actin assembly. Inhibiting local actin formation with the Arp2/3 inhibitor CK666 did not affect N-WASP accumulation at the ends of nanobars, indicating that the curvature-FBP17 cascade enables N-WASP recruitment on its own (Fig. EV3B–D). Overall, FBP17 intrinsically guides curvature sensing and N-WASP clustering. Cdc42 enhances this complex formation for curvature-mediated actin assembly by improving FBP17 and N-WASP recruitment, in addition to its established role in activating N-WASP through conformational changes.

We next dissected the cellular biochemical reactions underlying the radius-dependent actin polymerization via in vitro reconstitution. To define how Cdc42 roles cope with BAR protein's curvature sensing to regulate actin nucleation, we reconstituted their interplay on SLBs patterned with nanobar (Fig. 4A). To distinguish intrinsic curvature sensing from Cdc42-dependent regulation, we compared full-length FBP17 (FL-FBP17) with an HR1-deleted mutant, FdHR1, which cannot bind Cdc42 (Fig. 4B). We examined FBP17 recruitment with and without Cdc42 across various curvature gradients by quantifying two complementary parameters: the end

signal density, which indicates the absolute accumulation of protein at the ends of the bars, and the curvature preference, assessed by the end-to-center ratio, which is influenced by their baseline signal intensities. Both FL-FBP17 and FdHR1 displayed enrichment at the ends of the nanobars; however, only FL-FBP17 showed increased overall recruitment in response to Cdc42, with a more significant effect observed in low-curvature regions compared to high-curvature areas (Fig. 4C–E). In contrast, while FdHR1 was unaffected by Cdc42 as anticipated, it still exhibited end enrichment on the nanobars (Fig. 4F–H), confirming the intrinsic curvature-sensing ability of the BAR domain. We next applied G-actin and recombinant proteins to nanopatterned SLBs, including FBP17, N-WASP, Arp2/3, and CapZ, to examine NPF activation for actin nucleation. The presence of capping protein limited the initial elongation and spontaneous polymerization of F-actin, allowing a nucleation-specific assessment of its efficiency. Thus, we can monitor the initiation and accumulation of actin signals around nanobars, spanning from 200 to 1000 nm, and over time to observe the initiation period (within 180 s) of actin polymerization (Fig. 4I; Movie EV1). We noticed that 200–400 nm nanobars showed rapid actin polymerization starting from the 90 s. In contrast, wider ones took longer, which indicates that actin nucleation is accelerated on highly curved membranes (Fig. 4I). When the actin signal was within the initial linear increase period, curvature-dependent actin assembly was shown by quantifying polymerization rate (Fig. 4J,K). In contrast, the non-polymerizable(NP)-G-actin-based polymerization assay showed no noticeable actin signal increase over the curvature radius (Fig. EV4A–C).

We further explored the molecular interactions involved in Cdc42- and curvature-mediated N-WASP activation by visualizing bulky actin assembly on nanobar-curved SLBs. In the in vitro reconstitution assay, all references to Cdc42 refer specifically to GTP-loaded Cdc42 G12V, a constitutively active variant (Wu et al, 2013). Control groups lacking N-WASP (F + A + C) or Arp2/3 (F + N + C) showed no actin assembly or curvature sensitivity (Fig. 4L–N). Although Cdc42 alone can activate N-WASP for actin polymerization (N + A + C), this activation does not depend on curvature unless FBP17 is also present (F + N + A + C). Remarkably, actin polymerization was significantly enhanced when all components—Cdc42, FBP17, N-WASP, and Arp2/3—were present (F + N + A + C). This demonstrates strong cooperativity during

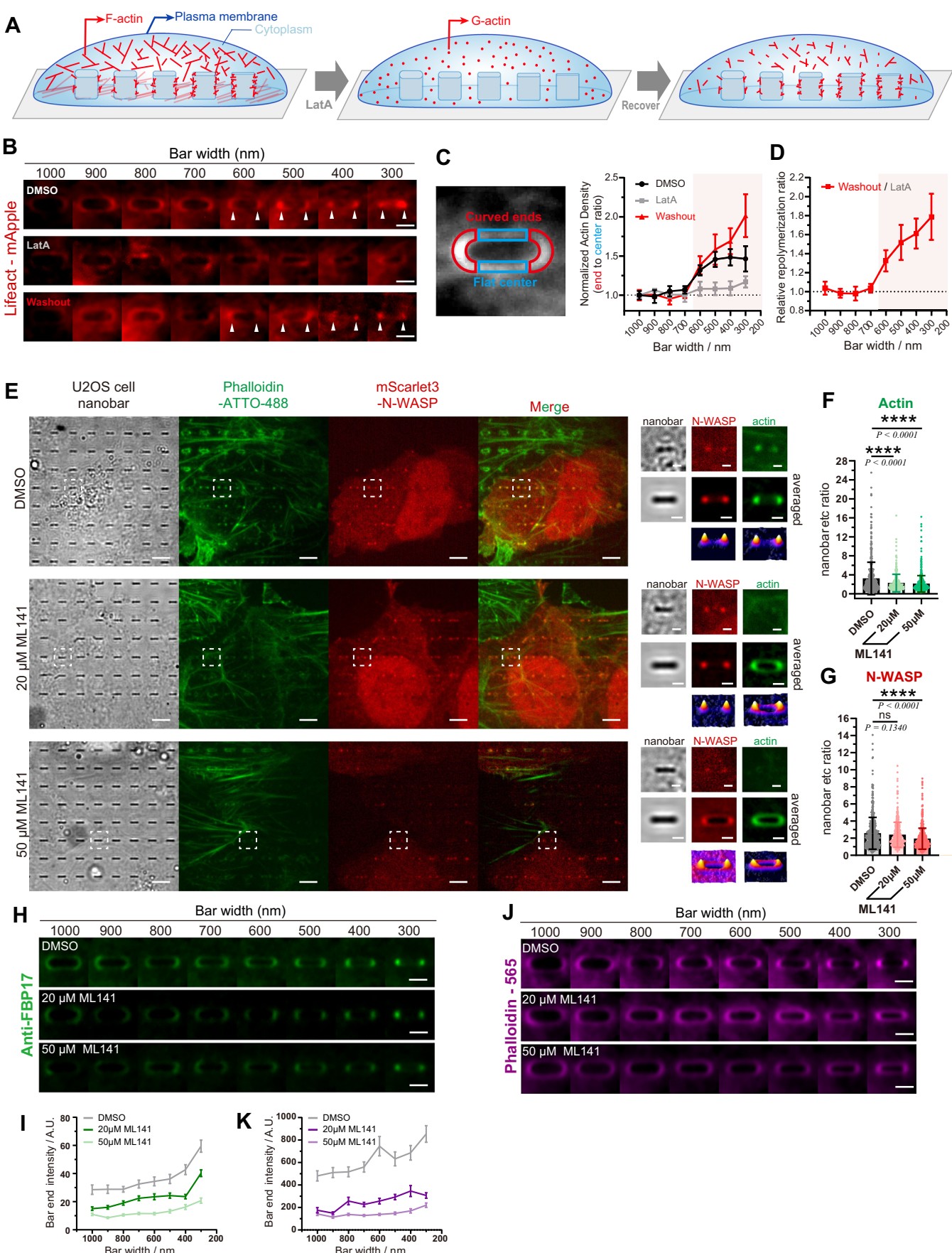

**Figure 3.  Cdc42 inhibitor attenuates actin assembly at the curved area.**

(A) A schematic illustrates the process of LatA treatment assay in U2OS cell lines on nanostructures. (B) Average confocal images of LifeAct-mApple-expressed U2OS live cells cultured on nanobars ranging from 300 to 1000 nm in width demonstrate F-actin recovery following a 1 h treatment with 100 nM LatA and a 20 min washout. Scale bar, 2 μm. (C) Normalized actin signal density was calculated based on the actin intensity in the flat area, at nanobars ranging from 300–1000 nm in width. Sample sizes for treatments under DMSO, LatA and washout were $N = 6$, 6, and 8 averaged nanobar ends, respectively. Each data point represents the mean ± SEM. (D) The normalized actin repolymerization ratio (Washout/LatA) in (C) at nanobars ranging from 300–1000 nm in width. $N = 8$ averaged nanobars, each point represents mean ± SEM. (E) Original and averaged confocal images of U2OS fixed cells cultured on 300-nm-wide nanobars with the expression of mScarlet3-N-WASP and the staining with Phalloidin-ATTO-488. Cells were fixed after a 1 h treatment with DMSO, 20 or 50 μM ML141. Scale bar, 10 μm (left) and 1 μm (right). Contrast(averaged): 12–30. (F) Actin signal end-to-center ratio at nanobars 300 nm in width. Sample sizes for treatments under DMSO, 20 or 50 μM ML141 were $N = 634$, 324, 689 nanobar ends, respectively. Each data point represents the mean ± SD. (****$P < 0.0001$). (G) mScarlet-N-WASP signal end-to-center ratio at nanobars 300 nm in width. Sample sizes for treatments under DMSO, 20 or 50 μM ML141 were $N = 634$, 324, 689 nanobar ends, respectively. Each data point represents the mean ± SD. (H) Average confocal images of U2OS cells cultured on nanobar arrays with bar widths ranging from 1000 to 300 nm, immuno-stained with anti-FBP17. Cells were treated with DMSO, 20 μM ML141, or 50 μM ML141. Scale bars: 2 μm. (I) Quantification of FBP17 intensity at nanobar ends across bar widths for each treatment condition in (H). Data represent mean ± SEM from $N = 45–57$ averaged nanobar ends per bar width per condition (exact $n$ values are provided in the Source Data). (J) Average confocal images of U2OS cells cultured on nanobar arrays with bar widths ranging from 1000 to 300 nm, stained with phalloidin-565. Cells were treated with DMSO, 20 μM ML141, or 50 μM ML141. Scale bars: 2 μm. (K) Quantification of Phalloidin-565(F-actin) intensity at nanobar ends across bar widths for each treatment condition in (J). Data represent mean ± SEM from $N = 34–54$ averaged nanobar ends per bar width per condition (exact $n$ values are provided in the Source Data). Statistical analysis was performed using one-way ANOVA followed by Tukey's multiple comparisons test. See also Fig. EV3. Source data are available online for this figure.

curvature-guided nucleation of actin polymerization curvature sensing, where Cdc42-mediated N-WASP activation, coupled with a preferential assembly of FBP17/N-WASP at highly curved sites to enable effective modulation of local actin polymerization (Fig. 4L,M). Then, we quantified the curvature sensing coefficient of actin assembly for all combinations by linearly fitting the actin signal slope as a function of curvature (Fig. 4N). We found that although Cdc42 boosts actin assembly at nanobar ends through N-WASP activation, it does not affect curvature sensitivity when comparing F + N + A and F + N + A + C, indicating two distinct yet interconnected mechanisms for curvature-based actin polymerization. It should be noted that nanobars with a curvature gradient share the same flat area, which supports the observation of actin assembly that depends on curvature radii, as measured by intensity, even though densely packed actin branches cannot be resolved by imaging. Importantly, substituting FBP17 with FdHR1 (FdHR1+N + A + C) preserved curvature sensitivity at higher curvature but abolished the Cdc42-dependent enhancement at low-curvature regions (Fig. 4L,M), consistent with the curvature-sensing pattern of FdHR1. The above FBP17 oligomerization potentiates N-WASP recruitment at high curvature, and then their macromolecular assembly in vivo and in vitro reconstitution results suggest that FBP17-mediated activation of N-WASP at curvature sites is closely coupled with Cdc42-driven basal actin polymerization. However, Cdc42 enhanced FBP17's membrane association has more pronounced roles at low-curvature regions, while at high curvature, FBP17 enables direct curvature sensing, facilitating N-WASP clustering and actin assembly activation even in the absence of Cdc42. This curvature radius-dependent cooperativity explains how global Cdc42 signaling is coupled to local geometric hotspots and also provides a mechanistic basis for curvature-driven actin assembly that often progresses from low to high curvature during membrane bending (Fig. 4O).

## Reconstituting the cooperative regulation of N-WASP–mediated actin polymerization by Cdc42 and FBP17

To elucidate how Cdc42 and FBP17 coordinate the multivalent assembly and activity of N-WASP, we assessed their individual and combined roles in promoting Arp2/3-mediated actin polymerization on SLBs using Cdc42 (C), FBP17 (F), and N-WASP (N) through TIRF time-lapse imaging and quantitative fluorescence intensity analysis (Fig. 5A–D; Movie EV2). In both the "+N + A" and "+F + A" conditions, spontaneous actin assembly occurred due to the absence of NPF activity, resulting in growth patterns dominated by unbranched, parallel filaments, similar to the control. When FBP17 and N-WASP coexisted ("+F + N + A"), initial nucleation and overall polymerization were enhanced, producing more branched networks and fewer parallel filaments. Furthermore, the well-known N-WASP activator Cdc42 directly promoted Arp2/3-based actin assembly ("+N + A + C"). Remarkably, when both FBP17 and Cdc42 were included ("+F + N + A + C"), we observed highly robust actin polymerization characterized by the highest fluorescence intensity and the densest F-actin network. This suggests a cooperative effect of Cdc42 and FBP17 in enhancing N-WASP-mediated actin nucleation through their direct inter-molecular interactions. To dissect this complex interplay, we disrupted the association between Cdc42 and FBP17 using FdHR1. In the "+FdHR1+N + A + C" condition, actin polymerization remained more efficient than "+N + A + C" but was markedly reduced compared with "+F + N + A + C" (Fig. 5B–D). The increased activity observed with FdHR1 indicates that curvature binding by the BAR domain and multivalent interactions between FdHR1 and N-WASP can enhance actin nucleation on their own, although this effect is weaker than the optimal synergy of Cdc42–FBP17–N-WASP tri-component assemblies. After 30 min of actin polymerization, the resulting cytoskeleton networks exhibited pronounced differences in organization, highlighting how the initial coordination of nucleation shapes the final network architecture (Fig. 5E). Specifically, the "+F + N + A + C" condition produced a highly dense and interconnected actin network originating from widely distributed asters. This suggests that multivalent interactions among FBP17, N-WASP, and Cdc42 facilitate cluster formation as effective nucleation centers. Actin networks formed with CapZ (Fig. EV5) also confirmed that the formation of clusters under the "+F + N + A + C" condition served as the nucleation nodes of branched actin filaments during network construction. Together, these data indicate that the formation of FBP17/N-WASP/Cdc42 tripartite is critical for maximal synergistic activation of N-WASP.

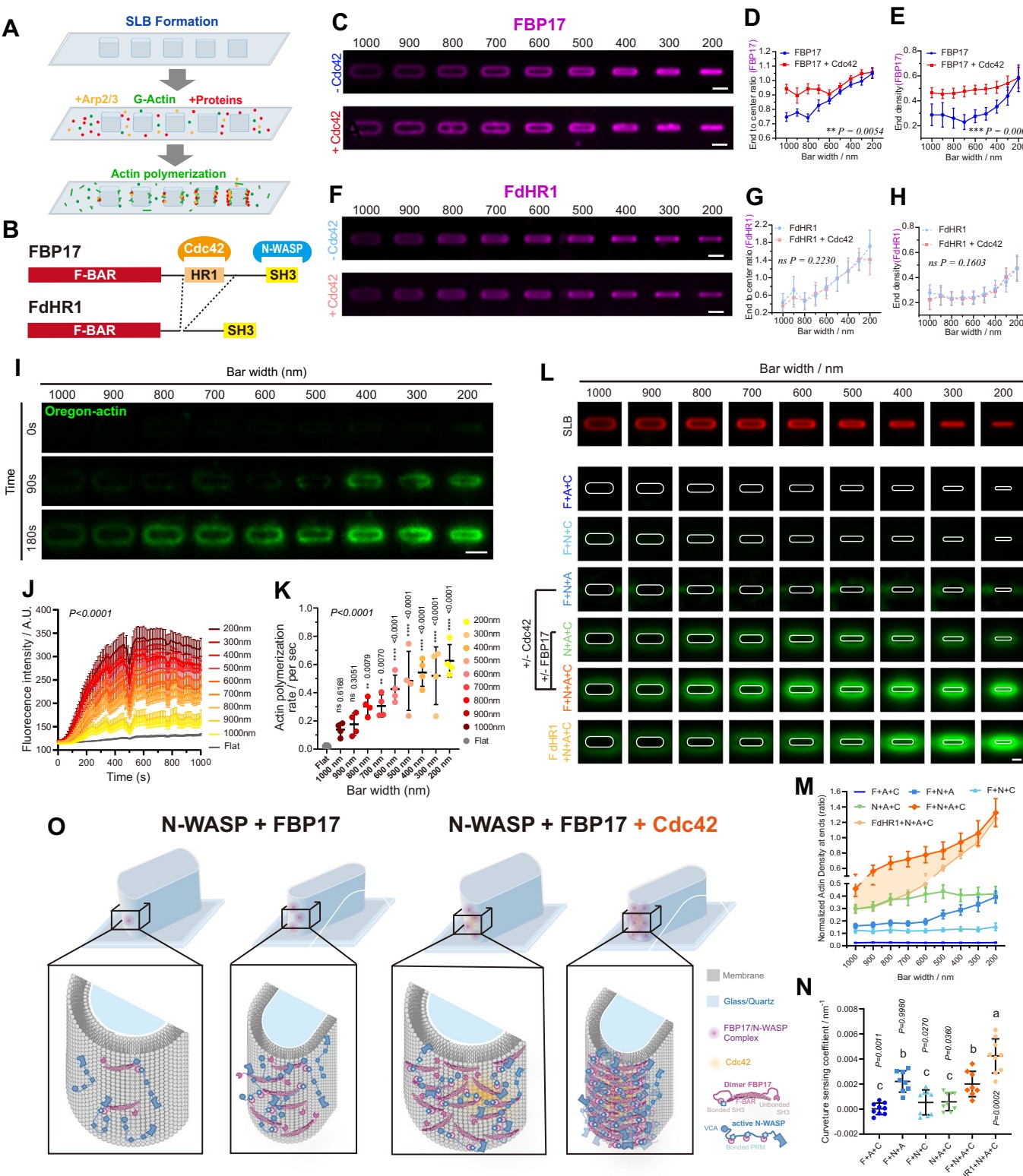

◀ **Figure 4. Cdc42 and FBP17 cooperatively facilitate radius-dependent actin assembly on nanobar-patterned SLB.**

(A) Schematic of SLB reconstitution assay on nanobar arrays. SLBs were incubated with G-actin, Arp2/3, and indicated proteins to assess curvature-dependent recruitment and actin polymerization. (B) Domain illustration of FBP17 and the HR1-deleted truncation mutant (FdHR1). FBP17 contains an F-BAR domain, HR1 domain for Cdc42 binding, and SH3 domain for N-WASP binding. (C) Average confocal images of FBP17 recruitment to nanobars in the absence (−Cdc42) or presence (+Cdc42). Scale bars, 2 μm. (D) Quantification of nanobar end enrichment ratio for FBP17 with or without Cdc42. Data represent mean ± SEM from $N = 36$ averaged nanobar ends. (E) Quantification of nanobar end density for FBP17 with or without Cdc42. Data represent mean ± SEM from $N = 36$ averaged nanobar ends. (F) Average images of FdHR1 recruitment to nanobars in the absence or presence of Cdc42. Scale bars, 2 μm. (G) Quantification of nanobar end enrichment ratio for FdHR1 with or without Cdc42. Data represent mean ± SEM from $N = 30$ or 39 averaged nanobar ends with or without Cdc42, respectively. (H) Quantification of nanobar end density for FdHR1 with or without Cdc42. Data represent mean ± SEM from $N = 30$ or 39 averaged nanobar ends with or without Cdc42, respectively. (I) Averaged confocal images of in vitro reconstitution of 1.5 μM actin (10% Oregon-labeled) polymerization at 0 s/90 s/180 s on the bilayer at nanobar of 200–1000 nm width with the presence of 200 nM FBP17, 50 nM N-WASP, 12 nM CapZ, and 5 nM Arp2/3. Scale bar, 2 μm. (J) Normalized actin signal intensity based on their corresponding lipid bilayer intensity at nanobars of 200–1000 nm width. Sample sizes for nanobars and flat regions were $N = 4$ nanobars, and five ROIs, respectively. Each point represents mean ± SEM. (K) Actin polymerization rate at nanobars of 200–1000 nm width. (The linear fitted slope within 105–350 s range in (J). Sample sizes for nanobars and flat regions were $N = 4$ nanobars, and five ROIs, respectively. Each point represents mean ± SEM. (L) Averaged confocal images of in vitro reconstitution of 3 μM actin (10% Oregon labeled) polymerization on the bilayer at nanobar of 200–1000 nm width with the combination of 200 nM FBP17(F), 200 nM FBP17 dHR1(**FdHR1**), 50 nM N-WASP(N), 100 nM Cdc42(C) and 5 nM Arp2/3(A). Scale bar, 1 μm. (M). Normalized actin signal intensity based on their corresponding lipid bilayer intensity at nanobars of 200–1000 nm width. Sample sizes were $N = 9, 9, 9, 9, 8$, and 11 averaged nanobar ends, respectively. Each point represents mean ± SEM. ($P < 0.0001$). (N) Curvature sensing coefficient at nanobars of 200–1000 nm width. (The linear fitted slope in M). Sample sizes were $N = 9, 8, 9, 9, 8$, and 9 averaged nanobars, respectively. Each point represents mean ± SEM. (O) Schematic of the proposed molecular interactions on nanobar-SLBs. FBP17 (red), N-WASP (blue), Cdc42 (orange), and their complexes are shown. Statistical analysis was performed using a paired two-tailed *t*-test(C, D, F, G), two-way RM ANOVA with Geisser–Greenhouse correction(J), ordinary two-way ANOVA(M), and ordinary one-way ANOVA followed by Tukey's multiple comparisons test (K, N). See also Fig. EV4; Movie EV1. Source data are available online for this figure.

## FBP17 amplifies N-WASP-mediated actin assembly by forming condensed nucleator complex zones

Given the demonstrated multivalent interactions for the FBP17/N-WASP complex (Fig. 1A), along with the notably increased F-actin polymerization, which occurs in a curvature radius-dependent manner (Fig. 4K), we are motivated to explore the specific molecular mechanisms that direct this cascade of recruitment and regulate the biochemical activity of actin polymerization (Fig. 6A). Upon investigating the high curvature nanobar, we noted that FBP17 exhibited a ~1.2-fold stronger signal end density at 200 nm in comparison to that observed at a nanobar size of 1000 nm. In the case of N-WASP, this ratio saw an increase to ~1.5-fold, and for the rate of actin polymerization, the ratio further rose to around 2.7-fold (Fig. 6B). Thus, we hypothesize that FBP17-driven N-WASP localization is critical to amplifying actin nucleation.

To test if N-WASP localization can enhance Arp2/3-driven nucleation, we generated $8 \times 8 \mu m^2$ copy number maps of N-WASP that were spatially discretized into compartments of size $0.5 \times 0.5 \mu m^2$ for different levels of localization. The simulations play a pivotal role in explaining the mechanistic origin of enhanced nucleation from N-WASP clustering. Experimental studies cannot tune the extent of N-WASP localization on SLBs. To overcome this limit, our model generates concentration profiles of N-WASP at various levels of localization. Towards this, we quantify localization based on the Shannon Entropy, $S(x,y)$, of the corresponding probability density functions. We generated probability density functions corresponding to various levels of N-WASP-localization using least squares optimization of the objective function $G := (S - f \times S_{max})^2, f \in [0, 1]$. Here, the factor $f$ is termed the localization factor and S represents the Shannon entropy of a given probability density. Uniform distributions result in the maximum value $S_{max}$ of the Shannon entropy function. Please refer to Methods for a detailed description of the model. The resulting probability density maps were sampled to generate copy number maps corresponding to $[N - WASP] = 1 \mu M$ (Fig. 6C). We then simulated the resulting Arp2/3 nucleation using MEDYAN (mechanochemical dynamics of active

matter), a C++ code to study the stochastic mechano-chemistry of active networks (Fig. 6D) (Chandrasekaran et al, 2019; Ni and Papoian, 2021; Popov et al, 2016). The initial condition for simulations at each localization factor consists of the N-WASP copy number map generated along with uniformly distributed G-actin ($[G - actin] = 1 \mu M$) and inactive Arp2/3 ($[Arp2/3_{inactive}] = 10 nM$) molecules. We consider N-WASP-dependent activation of Arp2/3 molecules along with polymerization, depolymerization, and dendritic nucleation reactions as described in Fig. EV6A. We use this framework to study how N-WASP localization affects Arp2/3 activation and filament nucleation. We quantify enhanced nucleation using both the number of filaments and the number of branches per μm². Our simulations reveal that the number of filaments nucleated increases with the localization of N-WASP-Arp2/3 (lower $f$ value, Fig. 6E). The simulations show that an adequate activation rate of Arp2/3 is necessary to see the localization-driven enhancement of Arp2/3 nucleation (Fig. EV6). Finally, we identified the branch clusters in the reaction volume and calculated the number of branches per $\mu m^2$. Our findings show that the branch density also increases with increased localization (Fig. 6F).

We also experimentally examine these model predictions by reconstituting an N-WASP-mediated Arp2/3 complex activation to study actin nucleation (Fig. 6G). We first sought to investigate if FBP17 could facilitate N-WASP-mediated activation of actin assembly. By experimenting with various mixtures of FBP17, N-WASP, and the Arp2/3 complex, we found that FBP17 triggers the formation of aster-shaped clusters of actin filaments (Fig. 6H,I; Movie EV3). Each cluster of actin filaments had an FBP17-enriched focus acting as the origin of nucleation (Fig. 6J). This observation suggests that FBP17 initiates actin polymerization by forming individual N-WASP-based nucleation centers through macromolecular assembly. Furthermore, we noted that the aster-shaped actin clusters would still form even in the presence of CapZ (Fig. 6H). CapZ binds to the actin barbed end and hinders the elongation of actin seeds outside of the FBP17/N-WASP nucleation center, around which the filaments are also shorter (Fig. 6H). Additionally, we also see that the increase in the number of branches in the

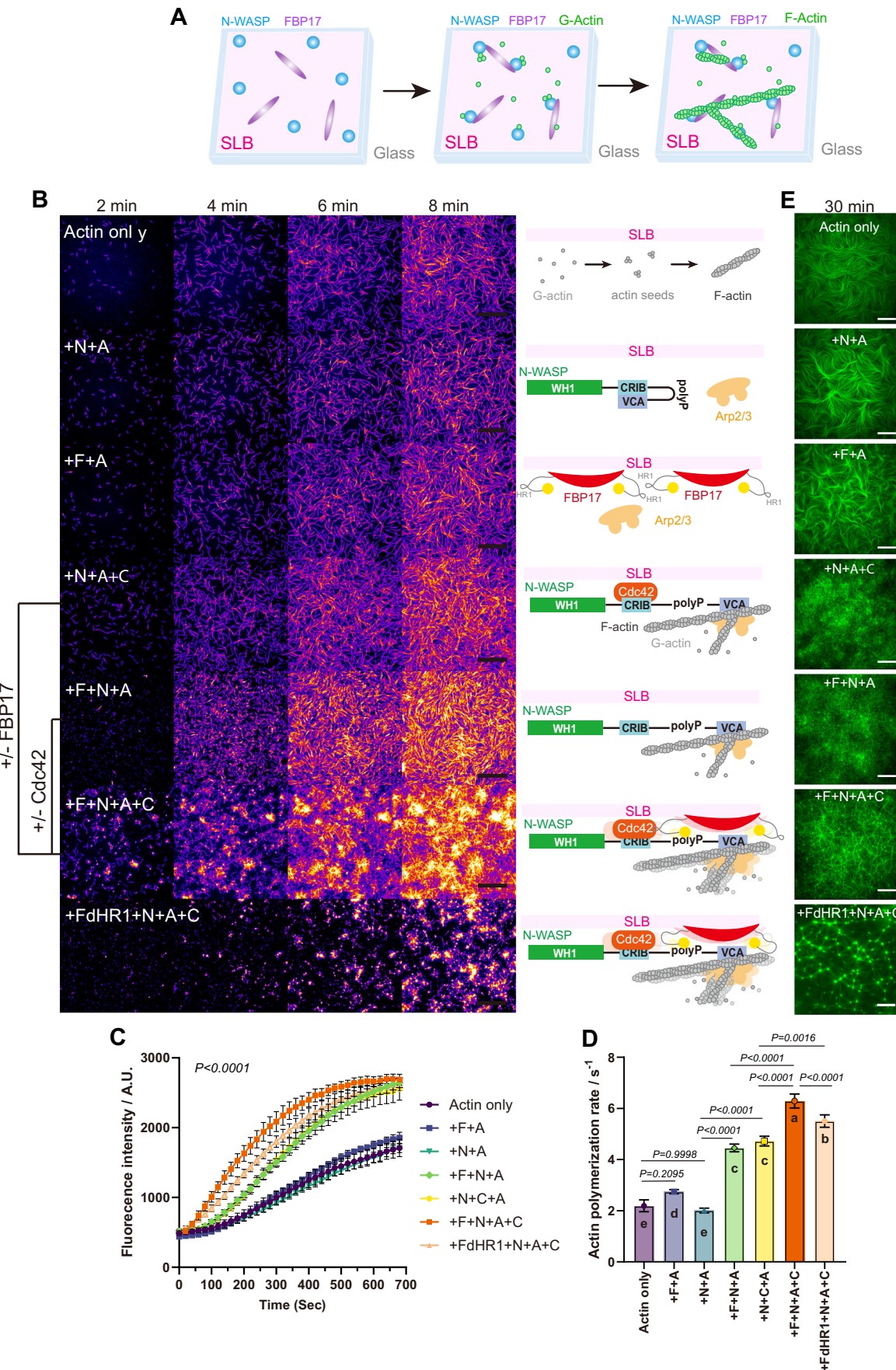

**Figure 5. Orchestrated N-WASP activations by FBP17-mediated clustering and Cdc42.**

(A) A schematic illustrates the process of in vitro reconstitution of FBP17-mediated actin polymerization at flat SLBs. (B) TIRF images and illustrations of actin polymerization (10% Oregon-labeled) with different combinations of 200 nM FBP17(**F**), 200 nM FBP17 dHR1(**FdHR1**), 50 nM N-WASP(**N**), 100 nM Cdc42(**C**), and 5 nM Arp2/3(**A**) at 0/2/4/6/8 and 10 min timeframe. Scale bar, 1 μm. Scale bar, 10 μm. (C) Normalized actin signal intensity. Sample sizes for nanobars and flat regions were $N = 4$ ROIs. Each point represents mean ± SD. (D) Actin polymerization rate. (The linear fitted slope within 105–350 s range in C). $N = 4$ ROIs. Each point represents mean ± SD. (E) TIRF images of actin polymerization (10% Oregon-labeled) with different combinations of 200 nM FBP17(**F**), 200 nM FBP17 dHR1(**FdHR1**), 50 nM N-WASP(**N**), 100 nM Cdc42(**C**), and 5 nM Arp2/3(**A**) at 30 min timeframe. Scale bar, 20 μm. Statistical analysis was performed using two-way RM ANOVA and one-way ANOVA followed by Tukey's multiple comparisons test ($p < 0.05$). See also Fig. EV5; Movie EV2. Source data are available online for this figure.

presence of FBP17-driven N-WASP clusters (Fig. 6G–J) is consistent with predictions from our model (Fig. 6C–F).

## N-WASP nanoclustering for actin assembly is mediated by multivalent interaction with FBP17

To investigate the cascade reaction in nanoclustering FBP17 and N-WASP complex for actin polymerization, we next examined whether homo-dimeric FBP17 can cluster full-length N-WASP through the direct interaction between its SH3 domains and the disordered PolyP region (271–391aa) of N-WASP (Fig. 1A). Using TIRFM imaging on SLB with the same lipid components as above, the size and total intensity of N-WASP foci increased with the addition of nanomolar FBP17 after 15 min incubation (Fig. 7A,B). Although the N-WASP signal is partially colocalized with FBP17, their colocalization generates bigger and more condensed foci (Fig. 7C,D). Meanwhile, we found that FBP17 also exhibits clustering behavior after adding N-WASP, and the size of FBP17 clusters grows stoichiometrically (Fig. EV7A). This observation revealed that FBP17 can induce the clustering of N-WASP molecules, which could be a potential mechanism for FBP17-mediated activation of N-WASP.

We next asked whether the multivalent interaction of SH3 and PolyP plays a determinative role in this nanoclustering-mediated N-WASP activation. We de novo engineered a trimeric SH3 protein by fusing SH3$^{FBP17}$ with a well-defined trimeric coiled-coil motif (Han et al, 2023; Khairil Anuar et al, 2019) (Fig. 7E). The recombinant trimer-SH3$^{FBP17}$ was purified and incubated with N-WASP on Flat SLB. We first characterized N-WASP oligomerization by single-particle imaging via TIRFM. We found that the control monomeric SH3$^{FBP17}$ doesn't induce obvious clustering of N-WASP, while trimerCC-SH3$^{FBP17}$ resulted in higher signal intensity in N-WASP clusters (Fig. 7F,G). We then assessed whether the multivalent SH3 induces N-WASP clustering and activates actin assembly. Via the TIRFM actin polymerization assay, we compared actin spontaneous polymerization using auto-inhibited N-WASP FL with Arp2/3 complex, using monomeric or trimeric SH3 domain. A significant increase in F-actin production and the formation of star-shaped actin clusters were observed in the presence of trimer-SH3$^{FBP17}$ but not in monomeric-SH3$^{FBP17}$, indicating a valency-dependent activation machinery of N-WASP FL (Figs. 7H,I and EV7C; Movie EV4).

## FBP17 oligomerization is critical for curvature-mediated actin assembly in cooperation with Cdc42

To investigate how hierarchical homotypic FBP17-FBP17 and heterotypic FBP17–N-WASP assembly on curved membranes is organized and functions in the presence of Cdc42, we included an additional oligomerization-deficient FBP17 mutant (K166A), which disrupts tip-to-tip F-BAR assembly (Fig. 8A,B) (Frost et al, 2008). On a flat SLB, FBP17 forms distinct fluorescent puncta, whereas the K166A mutant shows substantially reduced puncta intensity, which is corroborated by the decreased total fluorescence per particle and the shifted intensity distribution (Fig. 8C–E). We then compared the curvature sensing of FL-FBP17, FdHR1, and K166A, along with their ability to recruit N-WASP and promote actin polymerization in U2OS cells where Cdc42 is present (Fig. 8F–K). All FBP17 variants accumulated at nanobar ends, indicating that curvature sensing is primarily mediated by the BAR domain alone (Fig. 8F,I). However, N-WASP recruitment (Fig. 8G,J) and local actin polymerization (Fig. 8H,K) were both impaired in FdHR1 and K166A, suggesting that Cdc42 and FBP17 oligomerization are both important for potentiating N-WASP recruitment and actin assembly on curved membranes. Together, our results suggest that neither BAR-based curvature sensing nor activation of Cdc42 alone is sufficient to drive productive actin assembly at high curvature. FBP17 oligomerization potentiates N-WASP recruitment at high curvature and promotes their macromolecular assembly for activating actin polymerization.

Collectively, our in-cell and in vitro reconstitution data demonstrate that while membrane curvature sensing is intrinsically mediated by the BAR domain, full activation of N-WASP and actin nucleation depends synergistically on FBP17 oligomerization, FBP17–N-WASP co-assembly, and Cdc42 binding. These functional interplays are dynamically tuned by curvature radius. At high curvature, FBP17 oligomerization is the dominant factor, promoting N-WASP clustering and actin polymerization, whereas at lower curvature, Cdc42 plays the primary role by increasing FBP17 surface concentration to boost curvature-mediated nucleation efficiency (Fig. 9A,B). Thus, curvature-gated actin polymerization emerges from the integration of three layers: intrinsic BAR-domain geometry sensing, Cdc42 signaling, and oligomerization-driven scaffold clustering—ensuring that global Cdc42 activation couples with local curvature hotspots to spatially control actin assembly during membrane remodeling.

## Discussion

Our work elucidates how topographical cues guide stoichiometric interactions between curvature-sensing modules, nucleation-promoting factors, and small GTPases to orchestrate actin assembly. We demonstrate that N-WASP activation is facilitated by multivalent clustering with the F-BAR protein FBP17 via PRM-SH3 binding, while Cdc42 provides an additional bridging input

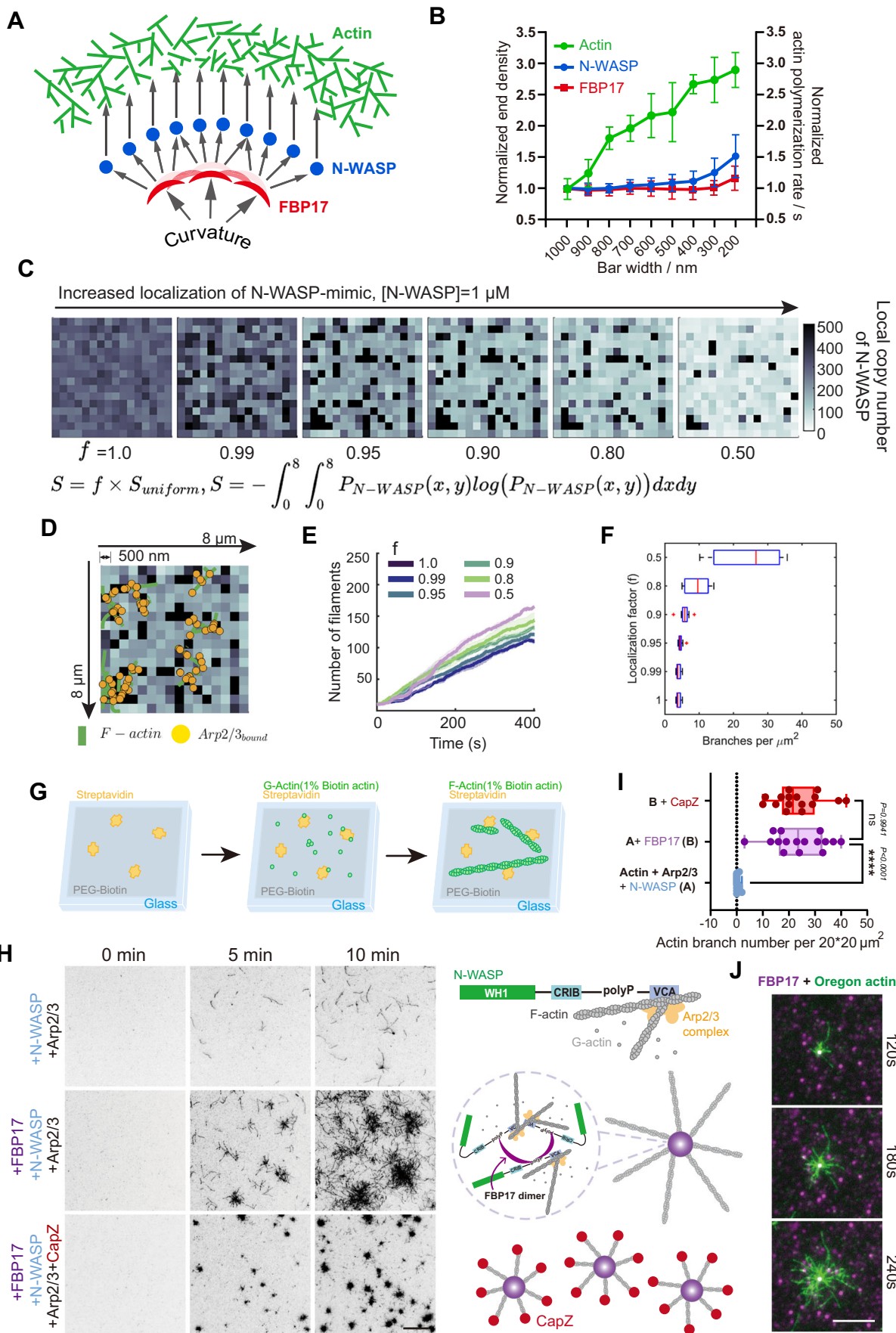

◄ **Figure 6. FBP17 localization activates N-WASP clustering, resulting in enhanced actin nucleation.**

(A) A schematic illustrates the in vitro reconstitution of amplification effects for actin polymerization at membrane curvatures. (B) Normalized nanobar-end density of FBP17(red) and N-WASP(blue) based on their corresponding lipid bilayer intensity (each point represents mean ± SEM from over 50 nanobars) and normalized actin polymerization rate(green) at nanobars of 200–1000 nm width (each point represents mean ± SEM from 4 nanobars). The FBP17, N-WASP, and Actin curves are integrated data from Fig. 1E and Fig. 4K here to elaborate the parameterization of the model in (A). (C) Spatial clustering of N-WASP is mathematically represented using the Shannon entropy function. Spatial density maps of N-WASP-mimics were subsequently generated at various values of the spatial localization factor (f) using constrained optimization technique (Please refer to Methods for detailed information). Colormaps show resulting N-WASP densities are shown for 8 μm × 8 μm space divided into compartments of size 0.5 μm × 0.5 μm. (D) Representative final snapshot ($t = 400$ s) corresponding to MEDYAN simulation of actin chemical dynamics at $f = 0.9$ in shown. The underlying N-WASP-mimic density is shown along with overlay of F-actin filaments (green) and filament-bound Arp2/3 (yellow). (E) Mean and standard error of mean time series profiles of number of filaments are shown colored by spatial localization factor (f) ($N = 5$). (F) Box plot shows distribution of branches per cluster computed from simulations with varying spatial localization of N-WASP. Data shown is scaled for $20 \times 20$ μm² cluster for ease of comparison against experiments. Red line represents median, the box bounds represent quartiles, and the whiskers represent bounds corresponding to 95% confidence intervals. Outliers are shown as red crosses. (G) A schematic illustrates the process of in vitro reconstitution of FBP17-mediated actin polymerization at PEG-biotin- and streptavidin-coated coverslips. (H) TIRF images of actin polymerization (10% Oregon-labeled) with different combinations of 80 nM FBP17, 20 nM N-WASP and 5 nM Arp2/3. Scale bar, 10 μm. (I) Box plot for the number of filaments per $37.44 \times 37.44$ um². $N = 16$ ROIs, mean ± SEM shown. The center line denotes the median, the bounds of the box represent the 25th and 75th percentiles, and the whiskers indicate the minimum and maximum values; all individual data points are shown. Statistical analysis was performed using one-way ANOVA followed by Dunnett's multiple comparisons test (ns $p = 0.9941$; ****$p < 0.0001$). (J) Dual-color TIRF images of actin polymerization (10% Oregon-labeled, green) with 80 nM AF647-FBP17(purple), 20 nM N-WASP and 5 nM Arp2/3. Scale bar, 5 μm. See also Fig. EV6; Movie EV3. Source data are available online for this figure.

through enhancing their local dose using the CRIB domain of N-WASP and the HR1 domain of FBP17. This tripartite assembly ensures that global Cdc42 activation is translated into local nucleation hotspots at sites of membrane curvature, while its regulation is modulated in a curvature radius-dependent manner. While curvature sensing is intrinsic to F-BAR domains, F-BAR protein scaffolding, hierarchical assembly with N-WASP, Cdc42-mediated membrane association, and Cdc42's direct activation of N-WASP synergize to promote actin polymerization on nano-curved membranes. This is different from canonical N-WASP activation via conformation changes with flat membranes, where Cdc42 predominantly drives actin polymerization, illustrating topology-guided orchestration: curvature boosts local actin polymerization for local force generation, cooperating with global lateral polymerization on flat surfaces, demonstrating orchestration of global cues with local membrane curvature to nucleate actin.

## Coordinated curvature sensing, nanoclustering, and membrane recruitment in local N-WASP activation by FBP17-N-WASP–Cdc42 complex

While curvature-guided macromolecular assembly defines the local hotspot for N-WASP activation via FBP17-guided clustering, the involvement of Cdc42 is critical to boost the actin nucleation in a curvature radius-dependent manner. In general, the auto-inhibited state of N-WASP is typically relieved via a conformational release facilitated by Cdc42 and PI(4,5)P2, enabling interaction with the Arp2/3 complex and subsequent basal activation of actin polymerization (Kim et al, 2000; Rohatgi et al, 2000). In addition, N-WASP is also known to respond to various cellular signals and supports multiple activation mechanisms, though whether and how these mechanisms orchestrate N-WASP activation spatiotemporally is sophisticated and not fully understood. The N-WASP/Arp2/3 complex can be activated in cell extracts, even in the absence of the typical plasma membrane environment where activated Cdc42 normally recruits N-WASP (Ho et al, 2004). In vitro studies have also shown that the SH3 domain of these BAR proteins can directly activate N-WASP-mediated actin polymerization on liposomes

with a specific range of diameter (Ho et al, 2004; Takano et al, 2008). Furthermore, molecular condensation-based N-WASP activation is also substantiated by phase separation studies that suggest a stoichiometry- and dwell-time-dependent enhancement of actin nucleation (Case et al, 2019; Ma et al, 2022; Su et al, 2016; Sun et al, 2021). Such condensation-driven activation events have also been observed in other systems. In plants, the activation of formin nucleators, single transmembrane domain proteins situated on the PM, was also found to rely on nano-scale clustering during immune responses and bypasses the need for a GTPase-regulated mechanism (Ma et al, 2021; Ma et al, 2022).

Here, our curvature-centric study elucidates how diverse regulatory mechanisms of N-WASP activation are coordinated in a curvature radius-dependent manner, mimicking spatiotemporal regulation as local membrane bending emerges and evolves from low to high curvature. We have taken a comprehensive approach, systematically dissecting the combined effects of key factors—including Cdc42-triggered conformational changes, Cdc42-dependent membrane recruitment, curvature-induced local clustering by FBP17, and stoichiometric ensembles of the FBP17-N-WASP complex—using FBP17 mutants that decouple from Cdc42 or fail to self-oligomerize, a synthetic engineered SH3 oligomer, and mathematical modeling. The findings not only contribute to the fundamental understanding of N-WASP activation but also hold promise for practical applications in the field of membrane dynamics and actin regulation.

The cellular and biochemical evidence converge on a unified model: progressive topological changes from low- to high-curvatures provide the spatial template, Cdc42 supplies the signaling input, and FBP17 oligomerization amplifies the assembly. This integration ensures that global Cdc42 activity is locally focused into discrete, curvature-gated nucleation hotspots, thereby coupling membrane curvature to efficient actin network formation. On flat surfaces, Cdc42 canonically regulates actin polymerization by exposing the VCA domain as a basal master regulation. Starting from shallow curvature, Cdc42 starts to facilitate FBP17-N-WASP recruitment to increase local concentration, priming the threshold for hierarchical assembly as membrane topology evolves to high

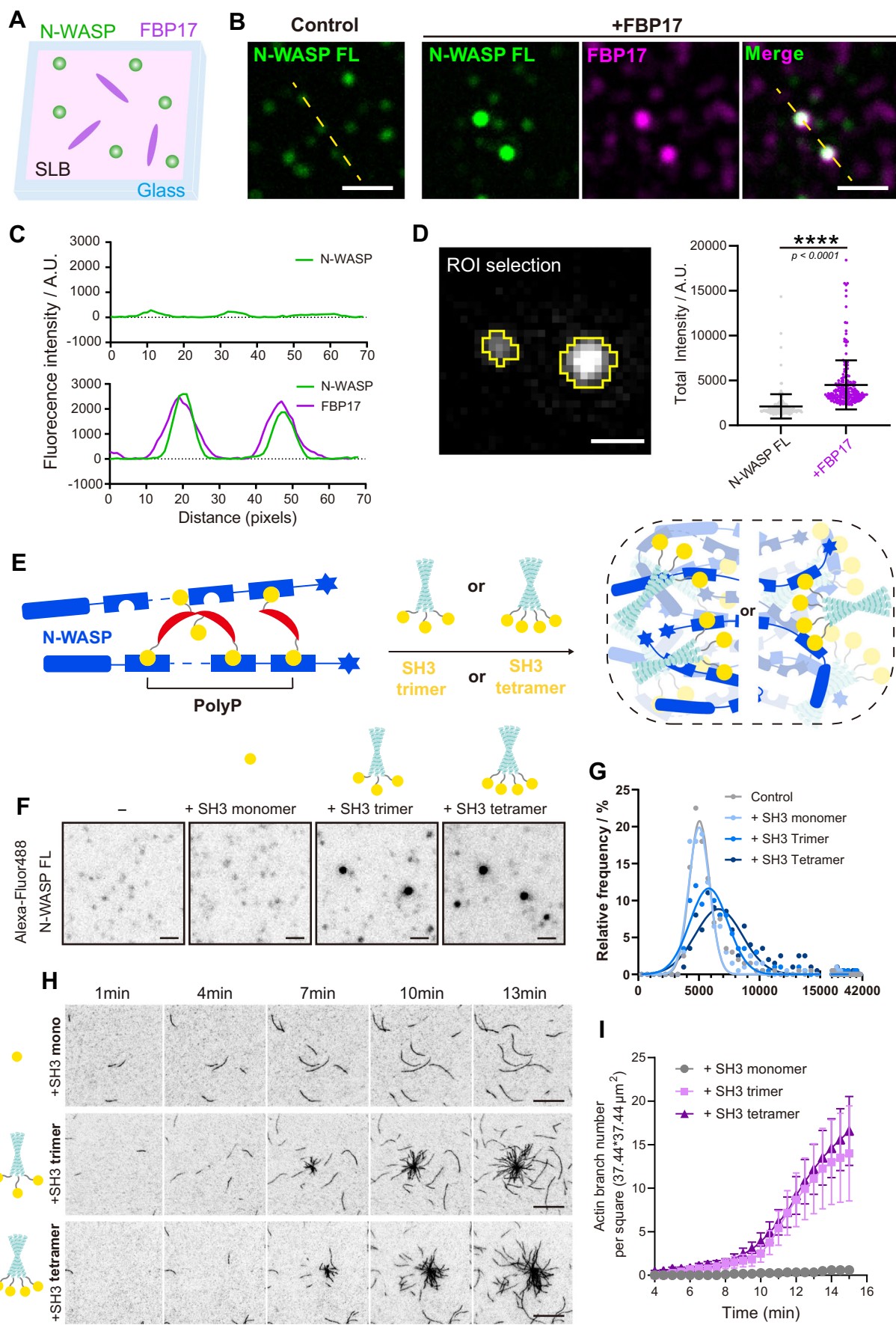

**Figure 7. N-WASP condensates mediated by FBP17 multivalent clustering and coarsening.**

(A) A schematic illustration of protein assembly assay on flat SLB. (B) Single-particle TIRF images of 5 nM AF488-N-WASP with 20 nM AF647-FBP17 on SLB. Scale bar, 2 nm. (C) Intensity profiles along the dashed yellow lines shown in (B) showing that clustering of N-WASP occurred in the presence of FBP17 and N-WASP colocalized with FBP17 in the clusters. (D) Left: the ROI selection of single particle by threshold. Right: total intensity plot of N-WASP single particles on SLB. $N = 200$ single particles out of over 10,000 particles, mean ± SD showed. Data were compared using Welch's unpaired $t$-test ($F$-test, $P < 0.0001$). (E) A schematic illustrates the multivalent interaction between N-WASP (blue) and FBP17 (red) or SH3 trimer/tetramer (yellow). (F) Single particle TIRF images of 5 nM AF488-N-WASP with 20 nM AF647-SH3, 20 nM AF647-oTri-SH3, or 20 nM AF647-oTet-SH3 on SLB. Scale bar, 1 μm. (G) Distribution of N-WASP single particle population on SLB. $N = 200$ single particles out of over 5000 particles, Gaussian normalization applied. Distributions were analyzed using Welch's ANOVA with Dunnett's T3 multiple comparisons test. Compared to the control, SH3 monomer shows adjusted $P = 0.3987$, while SH3 trimer (adjusted $P = 0.0002$), and SH3 tetramer (adjusted $P < 0.0001$). (H) TIRF images of actin polymerization (10% Oregon-labeled) with different combinations of 80 nM SH3, oTri-SH3 or oTet-SH3, 20 nM N-WASP and 5 nM Arp2/3. Scale bar, 10 μm. See also Movie EV4. (I) Number of filaments at $t = 60$–960 s per 37.44 × 37.44 um². $N = 12$ ROIs, mean ± SEM shown. Two-way repeated-measures ANOVA with Geisser–Greenhouse correction (factors: SH3 oligomerization × time). There was a main effect of SH3 oligomerization ($P = 0.0075$). See also Fig. EV7; Movie EV4. Source data are available online for this figure.

curvature. Then, FBP17's BAR domain intrinsically senses high curvature, reinforced by FBP17 oligomerization and multivalent SH3-PolyP interactions with N-WASP, leading to macromolecular assembly into local condensates that boost nucleation activity. This enables stepwise recruitment and condensation of N-WASP in a curvature progression-guided manner. Our findings reveal a dynamic, ensemble-dependent mode of N-WASP activation by integrating and dissecting all these factors, offering a spatiotemporal regulatory framework of how membrane topology orchestrates associated actin polymerization in living systems—a process previously challenging to examine due to the difficulty in capturing seconds-resolution dynamics of membrane topology changes via imaging or bulk actin polymerization assays (Fig. 9).

Different BAR-domain proteins sense different curvature radii and shapes to cope with membrane remodeling (Frost et al, 2008; McMahon and Gallop, 2005). For example, F-BAR proteins such as FBP17, CIP4, and FNBP1L preferentially bind shallow positive curvature, while I-BAR proteins engage convex geometries and recruit distinct actin regulators (Feng et al, 2022; Qualmann et al, 2011). The curvature-radius dependency of FBP17–N-WASP at high curvature may differ for other BAR domains, owing to variations in curvature-binding conformations and their distinct capacities for homo- and hetero-assembly, which in turn alter their cooperation with Cdc42 in regulating Arp2/3 nucleation efficiency.

## Implications for cortical actin waves and membrane morphodynamics

Curvature radius-dependent regulation highlights the spatiotemporal control underlying cellular dynamic membrane invaginations and oscillatory patterns that align with topographical changes over time (Day et al, 2021; Itoh et al, 2005; Lappalainen et al, 2022; Mondal et al, 2022; Tsujita et al, 2006; Wu et al, 2013; Wu et al, 2018; Yang et al, 2017). For example, phenomena such as PM cortical wave propagation depend on the dynamic binding of BAR domain proteins, with curvature guiding actin polymerization in response to local topographical changes on the membrane surface (Su et al, 2020; Yang et al, 2017), and Rho GTPase oscillations (Allard and Mogilner, 2013; Wu et al, 2013; Wu et al, 2018). Using our well-defined gradient curvature arrays, we precisely differentiate between the roles of Cdc42 and curvature-mediated clustering based on curvature radius. This mirrors the differential cooperation

modes at the initiation of cortical waves and membrane invaginations, where small deformations arise. Cdc42 provides early inputs that enhance FBP17 and N-WASP recruitment to initiate basal actin polymerization and topological stabilization, priming the system for large-scale actin remodeling triggered by orchestrated FBP17–N-WASP clustering as the membrane evolves to high curvature. It offers a plausible explanation for how actin waves can propagate across relatively flat membrane regions (Wu et al, 2018). A similar principle may operate during endocytosis (Day et al, 2021; Mondal et al, 2022; Tsujita et al, 2006), lamellipodial ruffling (Sitarska et al, 2023), and immunological synapse formation (Fooksman et al, 2010; Fritzsche et al, 2017), where actin assembly must initiate at shallow deformations before expanding into more pronounced morphological structures. Thus, our study provides a unifying framework in which nanoscale curvature and Cdc42 signaling cooperate to regulate not only local actin nucleation but also the spatial and temporal progression of complex morphodynamic events.

*Limitation of the study:* the modulation of N-WASP activity by FBP17 clustering, whether by altering intramolecular interactions as suggested for Toca-1 or by instigating direct conformational changes to expose the C-terminus, remains to be conclusively delineated by structural studies. Adding to the complexity is the challenge of understanding how nano-scale curvature determines N-WASP's biochemical activity in living cells—a promising yet experimentally complex area for future research. Although our minimal reconstitution systems provide mechanistic clarity, they lack the full regulatory complexity of living cells. Additional competing BAR proteins, and other NPFs such as WAVE or WHAMM, may modulate curvature responsiveness.

Cellular membrane curvatures range from tens to thousands of nanometers. Current electron-beam lithography (EBL) struggles to fabricate features below 100 nm radius—especially with high aspect ratios—due to proximity effects and etching limitations. Nonetheless, nanostructured systems that generate controlled curvature gradients now enable studies of protein interactions at curvatures up to 100 nm. This capability to observe biomolecule-nanoscale curvature dynamics expands research potential. Advances in EBL resolution and etching techniques could capture a broader range of curvatures, matching the full cellular spectrum. Integrating these with high-resolution imaging will greatly enhance understanding of biomolecular assembly.

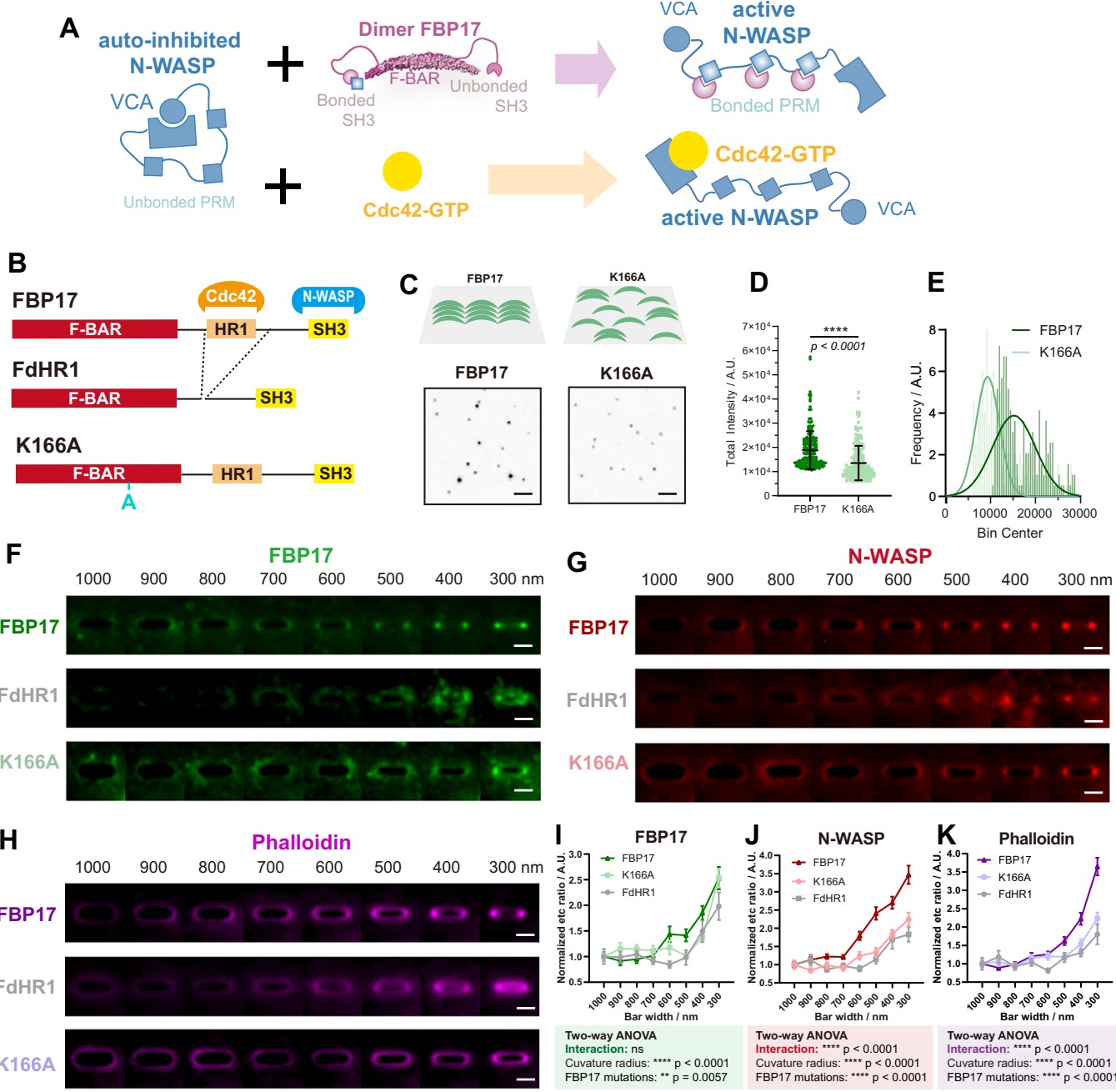

**Figure 8. FBP17 multivalent interactions via HR1–Cdc42 binding and tip-to-tip oligomerization are required for N-WASP recruitment and actin assembly at curvature sites.**

(A) Model of N-WASP activation by FBP17 (upper) and Cdc42 binding (lower). (B) Domain organization of FBP17, the HR1-deleted mutant (FdHR1), and the tip-to-tip oligomerization-deficient mutant (K166A). (C) Schematic and representative single-molecule imaging of FBP17 and K166A clustering (10% AF647 labeled). Scale bars, 1 μm. (D) Quantification of total intensity for FBP17 and K166A clusters. (E) Frequency distribution of total intensity for FBP17 and K166A clusters. (F) Average confocal images of U2OS cells seeded on nanobar arrays, expressing FBP17, FdHR1, or K166A. Scale bars, 2 μm. (G) Average confocal images of N-WASP recruitment in U2OS cells expressing FBP17, FdHR1, or K166A. Scale bars, 2 μm. (H) Average confocal images of phalloidin staining (F-actin) in U2OS cells expressing N-WASP with FBP17, FdHR1, or K166A. Scale bars, 2 μm. (I) Quantification of nanobar end-to-center ratio for FBP17, FdHR1, and K166A in (F). Data represent mean ± SEM from $N = 22$–64 averaged nanobar ends per bar width per condition (exact $n$ values are provided in the Source Data). (J) Quantification of nanobar end-to-center ratio for N-WASP in (G). Data represent mean ± SEM from $N = 23$–64 averaged nanobar ends per bar width per condition (exact $n$ values are provided in the Source Data). (K) Quantification of nanobar end-to-center ratio for phalloidin in (H). Data represent mean ± SEM from $N = 20$–70 averaged nanobar ends per bar width per condition (exact $n$ values are provided in the Source Data). Statistical analysis was performed using unpaired $t$-test or two-way ANOVA followed by Tukey's multiple comparisons test. Outliers were identified and excluded using the ROUT method ($Q = 1\%$). Source data are available online for this figure.

# Methods

### Reagents and tools table

| Reagent/resource | Reference or source | Identifier or catalog number |
|---|---|---|
| **Experimental models** | | |
| *Homo sapiens* bone osteosarcoma U2OS cells | ATCC | HTB-96 ™ |
| **Recombinant DNA** | | |
| Plasmids | N/A | See Tables EV2 and EV3 |
| **Antibodies** | | |
| FNBP1 Polyclonal Antibody | Thermo Scientific™ | Product Number: 5139887 |
| Goat anti-Mouse IgG(H + L) Highly Cross-Adsorbed Secondary Antibody, Alexa Fluor™ 488 | Thermo Scientific™ | Catalog number: A11029 |
| **Chemicals, enzymes and other reagents** | | |
| DMEM, high glucose, GlutaMAX™ Supplement | Gibco | Cat#10566016 |
| Fetal Bovine Serum, Research Grade, Heat Inactivated | GE Hyclone | Cat#SV30160.03HI |
| Protease inhibitor cocktail IV | Bioworld | Cat#22020010-1 |
| Trichloroacetic acid | Sigma | Product Number: T6399-250G; CAS:76-03-9 |
| Rabbit muscle acetone powder | Pel-Freez, LLC | Product code:41995-2 |
| Oregon Green™ 488 Iodoacetamide, mixed isomers | Invitrogen | Cat#O6010 |
| NHS-dPEG4-biotin | Sigma | Product Number: QBD10200-50MG |
| Hydrogen peroxide solution (30% w/w) | Sigma | Product Number: H1009- |
| Sulfuric acid | Sigma | Product Number: 258105-500ML-PC; CAS:7664-93-9 |
| mPEG-silane | Laysan Bio Inc | Lot#157-118 |
| Biotin-PEG-saline | Laysan Bio Inc | Lot#154-174 |
| Streptavidin | Sigma | Product Number: 11721666001 |
| Glucose oxidase from *Aspergillus niger* | Sigma | Product Number: G7141-10KU; CAS:9001-37-0 |
| Methylcellulose, viscosity 4000CP | Sigma | Product Number: M0512-250G; |
| Catalase from bovine liver | Sigma | Product Number: C40-100MG CAS:9001-05-2 |
| GelCode™ Blue Safe Protein Stain | Thermo Scientific™ | Cat#24596 |
| Arp2/3 protein complex | Hypermol | Cat#8413-01 |
| CapZ (non-muscle, human recombinant) | Hypermol | Cat#8322-01 |
| Latrunculin A | Sigma | Product Number: L5163-100UG |

| Reagent/resource | Reference or source | Identifier or catalog number |
|---|---|---|
| Non-polymerizing actin | Hypermol | Cat#8105-01 |
| PreScission Protease | Cytiva | GE27-0843-01 |
| Guanosine 5'-triphosphate sodium salt hydrate (GTP) | Sigma | Product Number: 51120-25MG CAS Number: 36051-31-7 |
| Phanta Max Super-Fidelity DNA Polymerase | Vazyme | Product serial number: P505-d1 |
| T4 DNA ligase | New England Biolabs | Cat#M0202S |
| Lipofectamine™ 3000 Transfection Reagent | Invitrogen | Cat#L3000001 |
| **Bacterial and viral strains** | | |
| *Escherichia coli* | N/A | Rosetta(DE3) |
| *Escherichia coli* | N/A | DH5a |
| **Software** | | |
| ImageJ | Open source | https://imagej.nih.gov/ij/ |
| Huygens Essential | SVI | https://svi.nl/Huygens-Essential |
| Trackmate | (Tinevez et al, 2017) | https://imagej.net/TrackMate |
| Graphpad Prism 10 | Graphpad Software | https://www.graphpad.com/scientific-software/prism/ |
| MATLABvR2021 | The Math Works, Natick, MA | https://www.mathworks.com/ |
| Metamorph software | Molecular Devices, USA | https://www.moleculardevices.com/products/cellular-imaging-systems/acquisition-and-analysis-software/metamorph-microscopy |
| **Other** | | |
| Glass-based dish,27 mm | Thermo Scientific™ | Cat#150682 |
| Microscope coverglass, thickness No.1.5, size: 24 × 50 mm | Marienfeld Superior, Germany | Cat#0102222 |
| HisTrap HP column | Cytiva | Product No:17524802 |
| HiLoad 16/600 Superdex 200 pg column | Cytiva | Product No:28989335 |
| Superdex 200 Increase 10/300GL column | Cytiva | Product No:28990944 |
| HiPrep 16/60 Sephacryl S-300 HR column | Cytiva | Product No:17116701 |
| Glutathione Sepharose™ 4B beads | Cytiva | GE17-0756-01 |
| Plastic flow cell chamber (sticky-slideVI0.4) | Ibidi GmbH | Cat#80608 |

## Recombinant protein purification and fluorescent labeling

The proteins FBP17 (aa 1–617) and FBP17 ΔHR1 (deleted aa 401–485) with an N-terminal His6 tag, N-WASP (aa 1–505) with

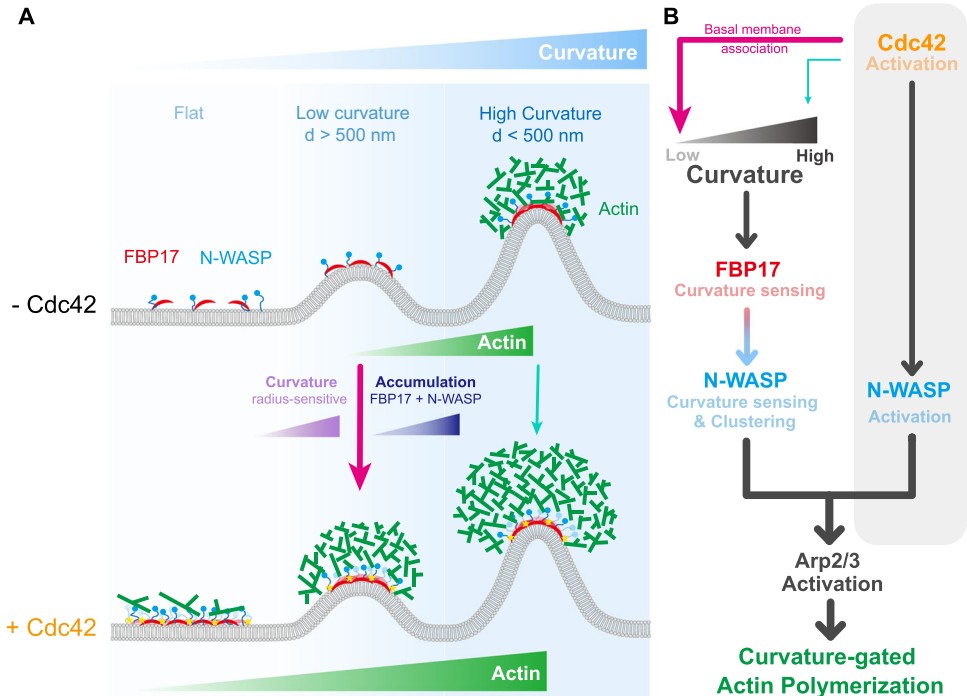

**Figure 9. Hierarchical regulation of FBP17-dependent actin assembly by membrane curvature and Cdc42.**

(A) Schematic of curvature-dependent actin assembly with and without Cdc42. At high curvature ($d < 500\,\text{nm}$), FBP17 intrinsically enriches and recruits N-WASP to nucleate actin, whereas at low curvature ($d > 500\,\text{nm}$), Cdc42 enhances FBP17 membrane association and stabilizes FBP17–N-WASP clusters, boosting actin nucleation efficiency. (B) Mechanistic model integrating curvature sensing, Cdc42 activation, and multivalent interactions. The gray shaded region indicates established mechanisms: Cdc42 relieves N-WASP autoinhibition through CRIB–VCA binding and activates Arp2/3 for basal actin assembly. The colored pathways represent new findings: FBP17 recruitment is curvature-sensitive and further amplified by Cdc42 at low curvatures, and multivalent interactions between FBP17 and N-WASP stabilize clusters that reinforce curvature-gated actin polymerization.

an N-terminal His6 tag, monomer-SH3, Trimer-SH3 with an N-terminal His6-Avi tag and Cdc42 G12V with a N-terminal GST tag were produced and isolated from BL21(Rosetta) cells. The Arp2/3 complex and the CapZ protein were procured from Hypermol.

BL21(DE3) Rosetta cells transformed with plasmids were selected using antibiotics. Subsequently, BL21 Rosetta cells were inoculated into 2 L of TB medium using a 50 mL overnight culture. The cells were then grown at 37 °C until they reached an optical density (O.D.) of 1–2 at 600 nm absorbance, and protein expression was induced with isopropyl-thio-β-D-galactoside (IPTG, 1 mM) at 16 °C for 16 h. After induction, cells were harvested by centrifugation and resuspended in 45 mL of Binding Buffer (20 mM HEPES, pH 7.4, 500 mM NaCl, 20 mM imidazole). Cell disruption was achieved using a homogenizer (LM20 Microfluidizer) in Binding Buffer supplemented with 1 mM PMSF, 0.1% (v/v) Triton X-100, and a protease inhibitor cocktail tablet. The supernatant was obtained by centrifugation at 20,000 rpm for 1.5 h at 4 °C, filtered (0.22 μm), and then applied onto a 5 mL HisTrap HP column (GE Healthcare) connected to an FPLC AKTA system (GE Healthcare). The column was washed with Washing Buffer (20 mM HEPES, pH 7.4, 50 mM imidazole, and 500 mM NaCl), and proteins were eluted with Elution Buffer (20 mM HEPES, pH 7.4, 500 mM imidazole, and 500 mM NaCl). The proteins were further purified by size exclusion chromatography on a HiLoad 16/600 Superdex 200 pg column (G.E. Healthcare) in Gelfiltration Buffer (20 mM

HEPES, pH 7.4, 500 mM NaCl, 10% glycerol, and 1 mM DTT), and finally concentrated using 15 mL 50 kDa cut-off concentrators (Amicon Inc.) to a concentration of ~5 mg/mL.

The purification of Cdc42 G12V was briefly described in (Wu et al, 2013). In brief, GST-Cdc42 G12V was bacterially expressed and purified using Glutathione Sepharose 4B beads (Cytiva) following the manufacturer's protocol. The protein was eluted from the beads and subjected to a second incubation with fresh beads to ensure saturation of bead capacity. For nucleotide loading, the GST-Cdc42–bound beads were incubated in buffer A (PBS supplemented with 2.5 mM $MgCl_2$) containing 2.5 mM GTP at 30 °C for 20 min. After two washes with buffer A, the supernatant was removed. The GST tag was cleaved using PreScission protease (Cytiva), and free GST, along with any uncleaved GST-Cdc42 G12V, was eliminated by incubation with fresh beads.

The primary amine groups of FBP17, FBP17 ΔHR1, monomer/trimer/tetramer-SH3, and N-WASP were labeled using an Alexa Fluor™647 and Alexa Fluor™488 labeling kit (Thermo Scientific). Briefly, the protein of interest was prepared at ~2 mg/mL in a 0.1 M sodium bicarbonate buffer. The Alexa dye was added to the mixture and incubated overnight at 4 °C. Excess dye was removed by using a 5 mL HiTrap Desalting column (GE Healthcare) in a Gelfiltration Buffer (20 mM HEPES, pH 7.4, 500 mM NaCl, 10% glycerol, and 1 mM DTT). Labeling efficiency was determined using a Nanodrop 2000 (Thermo Scientific). Since labeling efficiency varies from batch to batch, the exact amount of labeled protein in the reaction

was calculated according to the efficiency and mixed with unlabeled protein to reach the indicated labeling fraction (10% unless stated otherwise).

## Rabbit skeletal muscle actin purification and labeling

To obtain monomeric ATP-bound rabbit muscle actin (RMA) for the SLB actin reconstitution assay and TIRF actin assembly assay, two grams of rabbit muscle acetone powder (Pel-Freez, LLC) were dissolved in 200 mL of cold G-buffer (2 mM Tris, pH 8.0, 0.2 mM ATP, 0.5 mM DTT, and 0.1 mM CaCl$_2$) and stirred at 4 °C overnight. Subsequently, the mixture was filtered with cheesecloth to remove muscle powder, and the actin-dissolved solution underwent centrifugation at 2600 × $g$ and 4 °C (Type 45 Ti rotor, Beckman Coulter) for 30 min to collect the supernatant. Actin in the supernatant was then polymerized with slow stirring for 1 h at 4 °C by adding KCl and MgCl$_2$ solutions to final concentrations of 50 and 2 mM, respectively. To remove tropomyosin and other actin-binding proteins, fine KCl powder was slowly added to reach a final concentration of 0.6 M, and the solution was stirred for another 30 min. The solution was centrifuged at 14,000 × $g$ (Type 45 Ti rotor, Beckman Coulter) for 3 h at 4 °C to collect the filamentous actin pellet. The pellet was then rinsed with a cold G-buffer and homogenized with a homogenizer in 7 mL of cold G-buffer, followed by a short sonication time.

The sample was then dialyzed in 2 L G-buffer at 4 °C for 48 h to induce depolymerization, with G-buffer changed every 12 h. After buffer exchange, the sample underwent centrifugation at 167,000 × $g$ (S.W. 55 Ti swinging-bucket rotor, Beckman Coulter) at 4 °C for 2.5 h, and 5 mL of supernatant was collected and loaded onto a Sephacryl S-300 HR column (GE Healthcare) pre-balanced with G-buffer. Peak fractions were collected and combined, and then 0.01% (final) sodium azide (Sigma) was added to inhibit fungal contamination, maintaining the sample at 4 °C. Actin concentration was measured by assessing the OD290 with a Nanodrop 2000 (Thermo Scientific).

For actin labeling with Oregon Green™ 488 iodoacetamide (Invitrogen), the same purification steps were followed as for RMA until the pelleted filamentous actin was homogenized and sonicated. After this, the sample was dialyzed in 1 L G-buffer at 4 °C overnight. The next day, the sample was changed to 1 L G-buffer without DTT and dialyzed for 4 h at 4 °C, with the buffer changed once. Oregon Green™ 488 iodoacetamide was dissolved in dimethylformamide to a final concentration of 10 mM. Before labeling, actin concentration was measured by reading the OD290 with a Nanodrop 2000 (Thermo Scientific).

Actin was first diluted with an equal volume of cold 2X labeling buffer (50 mM imidazole, pH 7.5, 200 mM KCl, 0.6 mM ATP, and 4 mM MgCl$_2$) and further diluted to a final concentration of 23 mM with cold 1X labeling buffer. Then, a tenfold molar excess of Oregon Green™ 488 iodoacetamide was added dropwise while gently vortexing. The mixture was covered with aluminum foil and rotated at 4 °C overnight. The following day, labeled filamentous actin was centrifuged at 167,000 × $g$ (Type 50.2 rotor, Beckman Coulter) for 3 h at 4 °C. Pellets were collected, homogenized in 4 mL G-buffer, left on ice for 1 h, and homogenized again. Actin was then dialyzed in 1 L G-buffer at 4 °C for 48 h to induce depolymerization (in darkness, with G-buffer changed every 12 h). After buffer exchange, actin was centrifuged at 436,000 × $g$

(TLA100 rotor, Beckman Coulter) at 4 °C for 1 h. The supernatant was collected and further purified by a Sephacryl S-300 HR column (GE Healthcare) pre-balanced with a G-buffer. Peak fractions were collected and combined, then dialyzed in 500 mL G-buffer with 50% (v/v) glycerol at 4 °C overnight to reduce volume. Small aliquots were frozen in liquid nitrogen and stored in a −80 °C freezer.

## Fabrication of nanobar Chips

The nanobar arrays utilized in this study were fabricated on square quartz coverslips using electron-beam lithography, as described by (Zhao et al, 2017). Briefly, 15 × 15 × 0.2 mm square cover-slips were initially spin-coated with positive electron-beam resist polymethyl methacrylate (Allresist) to a height of approximately 300 nm. A conductive protective coating was subsequently applied using AR-PC 5090.02 (Allresist). Nanobar patterns were then delineated via electron-beam lithography (F.E.I. Helios NanoLab) and developed in the developer AR 600-56 (Allresist). A 300-nm-height chromium mask was generated through thermal evaporation (UNIVEX 250 Benchtop) and lifted off using acetone. The nanobars were then created via reactive ion etching with a mixture of CF$_4$ and CHF$_3$ (Oxford Plasmalab 80). Following the application of a 10 nm Cr layer, the nanostructures underwent characterization through scanning electron microscopy to examine their curvature, diameter, and structural height. Before use, the nanobar chips underwent immersion in Chromium Etchant (Sigma-Aldrich) until the chromium masks were removed entirely.

## Preparation of lipid vesicles

The lipid vesicles were composed of 1-palmitoyl-2-oleoyl-glycero-3-phosphocholine (POPC) mixed with 0.5 mol% of 18:1 Rhoda-mine-PE, 1 mol% of phosphatidylinositol 4,5-bisphosphate (PI(4,5) P$_2$), and 10 mol% of 1-palmitoyl-2-oleoyl-*sn*-glycero-3-phospho-L-serine (POPS). Initially, the lipids were dissolved in chloroform and mixed at the specified molar ratios. The lipid mixture was then dried in a brown glass vial using 99.9% nitrogen gas for 5 min, followed by vacuum drying for 1.5 h to eliminate residual chloroform. Subsequently, the dried lipid film was resuspended in phosphate-buffered saline (PBS) at 0.5 mg/mL and sonicated for 30 min. The resulting lipid mixture was transferred to a 1.5 mL tube and subjected to freeze-thaw cycles 15 times (20 s in liquid nitrogen followed by 2 min in a 42 °C water bath). The lipid mixture was further extruded through a polycarbonate membrane with a pore size of 100 nm using an extruder equipped with a holder/heating block (610000-1EA, Sigma-Aldrich). Finally, the lipid vesicle solution was stored at 4 °C and utilized within 7 days.

## Formation of protein-bound SLBs on nanobar chips

The nanobar chips underwent cleaning with piranha solution (composed of seven parts concentrated sulfuric acid and one part 30% hydrogen peroxide solution) overnight, followed by rinsing with a continuous stream of deionized water to eliminate the acids. Subsequently, the nanobar chips were dried using 99.9% nitrogen gas and subjected to cleaning with air plasma in a plasma cleaner (Harrick Plasma) for 1 h to eliminate any remaining impurities on the surfaces before lipid bilayer formation. Once cleaned, the chips

were attached to a polydimethylsiloxane (PDMS) chamber, and lipid vesicles were introduced into the PDMS channel, where they were allowed to incubate for 15 min to facilitate the formation of the lipid bilayer. PBS was employed to rinse away any unbound vesicles within the chamber. Following this, the protein solution of the desired concentration was premixed with the indicated molar ratios in PBS and incubated for 2 min before introducing it onto the lipid bilayer-coated nanobar. Microscopy imaging was conducted after 30 min of incubation at room temperature.

## Imaging protein-nanobar interaction

The distribution of purified proteins on lipid-bilayer-coated nanobar arrays was investigated using a spinning disc confocal (SDC) system built around a Nikon Ti2 inverted microscope equipped with a Yokogawa CSU-W1 confocal spinning head and a 100X/1.4NA oil immersion objective. The Rhodamine-PE-containing lipid bilayer was excited at 561 nm and detected at 570–645 nm. Alexa Fluor 488-labeled N-WASP was excited at 488 nm and detected at 490–570 nm, while Alexa Fluor 647-labeled FBP17 was excited at 633 nm and detected at 645–700 nm.

## Formation of protein-bound SLBs on flat surfaces

Flat SLBs were prepared in 96-well glass-bottomed plates (Matrical). Initially, wells were subjected to washing with 1 L of 5% (v/v) Hellmanex III (Hëlma Analytics) overnight while gently vortexed by a magnetic stirrer. Following this, the wells were thoroughly rinsed with deionized water (DI $H_2O$) for 20 cycles, washed with 5 M NaOH for 1 h at 50 °C for three repetitions, and again thoroughly rinsed with DI $H_2O$. Subsequently, the wells were dried using 99.9% nitrogen gas and treated with air plasma in an HP plasma cleaner (Harrick Plasma) for 90 min to remove any remaining organic contaminations on the glass surfaces before SLB formation. SUVs were added to the wells and incubated for 15 min at 42 °C to allow the SUVs to collapse and fuse onto the glass, thereby forming SLBs. The SLBs were then rinsed with PBS to wash away any unbound SUVs. To reconstitute the protein interaction on the lipid membrane, a protein solution of the desired concentration was premixed with the indicated molar ratios in PBS, incubated for 2 min, and added to the well. After 15 min of incubation, unbound proteins were removed by washing with PBS.

## Imaging of protein clustering on flat SLBs

Images were acquired at room temperature using a Nikon Ti2-E inverted microscope equipped with a 100 × 1.45NA Plan-Apo objective lens and a TIRF module (iLasV2 Ring TIRF, GATACA Systems) and an ORCA-Fusion sCMOS camera (Hamamatsu Photonics). Image acquisition was controlled by MetaMorph software (Molecular Devices).

## U2OS cell culture

*Homo sapiens* bone osteosarcoma U2OS cells (ATCC) were maintained in DMEM with GlutaMAX (Gibco) supplemented medium with 10% fetal bovine serum (FBS) and 1% Penicillin/Streptomycin (PS). All cell lines were seeded in 100 mm plastic dishes with ~$1 \times 10^6$ cells per dish. Cells were kept in an incubator with 5% $CO_2$ and 100% relative humidity at 37 °C.

## Immunostaining of FBP17 and F-actin

Cells were fixed in 4% paraformaldehyde (PFA) in PBS for 15 min at room temperature, washed three times with PBS, and permeabilized with 0.1% Triton X-100 in PBS for 5 min. After three washes, samples were blocked in 1% BSA in PBS for 1 h at room temperature. For FBP17 staining, cells were co-incubated with mouse anti-FBP17 primary antibody (dilution 1:100, Thermo Scientific™), phalloidin conjugated to ATTO 565 (dilution 1:500, Hypermol) for 2 h at room temperature, followed by three washes in PBS and incubation with Alexa Fluor™ 488 conjugated secondary antibody (1:500, Invitrogen) for 30 min at room temperature.

## Cell imaging with nanobar-substrates

To enable cell imaging on the nanochip, the chip was attached to the 35 mm cell culture dish (TPP) with a hole punched in the center to expose the nanobar pattern. Before cell plating, the dish substrate was sterilized by UV treatment for 20 min. The surface was coated with 0.2% fibronectin (Sigma-Aldrich) for 30 min to promote cell attachment. U2OS cells were then cultured on the nanobar chip for one day before transfection. U2OS cells were then transiently transfected with the lifeact-mApple/mScarlet-N-WASP/GFP-FBP17 plasmids using Lipofectamine 3000 (Invitrogen, USA) following the manufacturer's protocol and grown overnight on a coated nanobar chip at 37 °C in a $CO_2$ incubator for protein expression. For fixed cell imaging, cells were fixed in 4% PFA in PBS for 15 min at room temperature, washed three times with PBS, and permeabilized with 0.1% Triton X-100 in PBS for 5 min. After three washes, samples were incubated with phalloidin-ATTO 565 (dilution 1:500, Hypermol) for 1 h at room temperature. For live cell imaging, 100 nM LatA or the same volume of DMSO was added to the cells, incubated for 1 h at 37 °C, washed three times with PBS, and changed back to DMEM. During imaging, the cells were maintained at 37 °C with 5% $CO_2$ in an on-stage incubator. For a fixed cell image, U2OS cells were fixed using 3% trichloroacetic acid (TFA) in PBS. Briefly, the cells were washed twice with PBS at room temperature, followed by the addition of ice-cold 3% TFA solution. Cells were incubated for 15 min before being washed three times with PBS to remove residual TFA and stored in PBS at 4 °C until further processing. The fixed cells were stained with Phalloidin-ATTO 488/647 following the manufacturer's protocol before imaging. Cell imaging was then performed with an SDC built around a Nikon Ti2 inverted microscope containing a Yokogawa CSU-W1 confocal spinning head and a 100 X/1.4NA oil immersion objective.

## TIRF actin assembly assay in vitro

The in vitro real-time actin assembly assay was conducted on Biotin-PEG-coated glass slices (Laysan Bio Inc) in six-well chamber slices (Ibidi). The chamber was first blocked by 30 mL HBSA buffer (20 mM HEPES pH 7.4, 1 mM EDTA, 50 mM KCl, 1% (m/v) BSA) and incubated for 30 s. The glass surface was then conjugated with streptavidin by adding 30 mL HEKG10 buffer (20 mM HEPES pH 7.4, 1 mM EDTA, 50 mM KCl, 10% (v/v) glycerol, plus 0.1 mg/mL

streptavidin) and incubating for 1 min. Afterward, free streptavidin was washed away using 1x TIRF buffer (10 mM imidazole, pH 7.4, 50 mM KCl, 1 mM MgCl$_2$, 1 mM EGTA, 50 mM DTT, 0.3 mM ATP, 20 nM CaCl$_2$, 15 mM glucose, 100 mg/mL glucose oxidase, 15 mg/mL catalase, and 0.25% methylcellulose). Next, 30 μL 3X protein-actin mix containing 1.5 μM G-actin (79% purified globular rabbit actin, 20% Oregon Green 488-actin, and 1% Biotin-actin), 200 mM EGTA, 110 mM MgCl$_2$, and the desired protein solution were mixed with 30 μL 2X TIRF buffer and added into the chamber to a final volume of 90 μL to initiate the actin polymerization. Images were acquired as a stack at room temperature with 15-s intervals for 15 min using a Nikon Ti2-E inverted microscope equipped with a 100 × 1.45NA Plan-Apo objective lens and a TIRF module (iLasV2 Ring TIRF, GATACA Systems) and an ORCA-Fusion sCMOS camera (Hamamatsu Photonics). Imaging lasers were provided 488 nm/150 mW (Vortran) and 639 nm/150 mW (Vortran) combined in a laser launch (iLaunch, GATACA Systems). Focus was maintained by hardware autofocus (Perfect Focus System), and image acquisition was controlled by Meta-Morph software (Molecular Device). To quantify the number of actin branches, 13 × 13 μm$^2$ ROIs were chosen from each time point image, and the actin filament branches in each ROI were manually counted. The N number refers to the number of square ROIs.

## Reconstitution of actin polymerization on SLBs

SLBs were formed and coated with proteins, as described above. After acquiring a pre-actin image, the SLBs were rinsed with 60 μL 1X TIRF buffer 3 times. Subsequently, 30 μL 4X actin mix containing G-actin (10% Oregon Green™ 488 labeled), Arp2/3 complex, 0.25 mM ATP, 0.1 mM MgCl$_2$, 0.1 mM EGTA, and the desired protein solution were added and mixed with 30 μL 1X TIRF buffer into the well with gentle pipetting. TIRF images were acquired at 15 s intervals for 15 min for all experiments. Oregon Green 488-actin images were acquired using 100 ms exposure, and the 488-laser power was set to 15%. AF647-FBP17 images were acquired using 200 ms exposure, and the 647-laser power was set to 50%. No detectable crosstalk was observed with these settings.

## Modeling actin nucleation under varying N-WASP localization

We hypothesize that the localization of N-WASP leads to spatially localized activation of Arp2/3, resulting in enhanced nucleation of Arp2/3. Thus, we test whether the spatial localization of N-WASP is adequate to enhance nucleation activity using an agent-based modeling approach. In this model, we do not represent the membrane explicitly. Instead, we specify a spatial distribution of N-WASP concentration as the initial condition and study the role it plays in actin nucleation. Towards this, we first develop a model based on Shannon Entropy to systematically generate spatial copy number maps at progressively increasing N-WASP localization. Then, we simulate the resulting actin nucleation using Mechan-ochemical Dynamics of Active Matter v5.1.0 (MEDYAN v5.1.0).

For this purpose, we consider a 16 × 16 × 1 compartment space, each of size 500 × 500 × 500 nm$^3$ as the reaction volume. N-WASP copy number in each compartment can be specified at various levels of localization. Such copy number maps can be readily converted to probability density functions. Thus, to quantify spatial

localization, we resort to the Shannon Entropy function.

$$H(x,y) := \iint p(x,y)dxdy \tag{1}$$

In the above equation, $p(x,y)$ represents the probability density function. Uniform probability density functions (represent random distribution of N-WASP without any localization) correspond to the maximum values of $H(x,y) = H_{max}$. To compute $H_{max}$, we generate a uniform distribution as the initial condition and the corresponding Shannon Entropy $H_{max}$. As the localization of N-WASP (p(x,y)) increases, the corresponding Shanon Entropy decreases ($H(x,y) \leq H_{max}$). To generate such probability densities, we use the sequential least squares programming (SLSQP) algorithm specified in Python3.0 scipy.optimize.minimize to minimize the following objective function.

$$G(x,y) = (H(x,y) - f.H_{max})^2 \tag{2}$$

In the above equation, $f \in (0,1,]$ represents a factor of localization. When $f = 1.0$, the minimum $G_{min}$ corresponds to a uniform distribution. At lower values, the corresponding minimum yields probability density distributions with increased localization. In this study, we consider $f$ values of 1.0, 0.99, 0.95, 0.8, and 0.5, respectively.

The resulting probability density functions are sampled to compute copy number maps corresponding to the Arp2/3 activator molecule, N-WASP. To ensure that the resulting copy number maps give us the same Shannon Entropy values as the probability density functions, we progressively increase the total concentration and compute the relative error in Shannon entropy between the optimal density function and the sampled copy number map. We find low relative error values (<1%) at concentrations above 1 μM. We consider the total N-WASP concentration to be 1 μM in this study. Please note that while these values do not correspond to experimental values of concentrations studied, the copy number, along with the rate of Arp2/3 activation, gives us a semi-quantitative simulation framework to study the role of localization.

To study actin nucleation resulting from the copy number maps, we use MEDYAN, a C++ stochastic mechanochemical simulator for actin networks. MEDYAN represents filaments explicitly as fibers and can also simulate the dendritic nucleation of actin filaments (Popov et al, 2016). We describe the chemical, mechanical, and mechanochemical frameworks considered in this study.

In MEDYAN, the reaction volume is divided into compartments of size 500 × 500 × 500 nm$^3$. The copy number of chemical species is specified within each compartment. Stochastic diffusion is modeled as random hopping between neighboring compartments. In MEDYAN, reaction propensities are computed by assuming uniform mixing within each compartment. Thus, each compartment has reactions whose propensities are calculated based on the local concentration of reactive species in that compartment. Additionally, actin filaments are explicitly represented as fibers. We assume that every 27 nm of actin filament consists of one actin-binding site that is available for Arp2/3-driven nucleation. Thus, we consider the reactions shown in Fig. EV4 within each compartment.

While $G - actin$ and $Arp2/3_{inactive}$ diffuse freely throughout the reaction volume, N-WASP molecules do not diffuse in our model. This assumption helps us capture the membrane-bound nature of N-WASP. In addition, as Arp2/3 activation is coupled with

N-WASP binding, we assume that the activated Arp2/3 molecules do not diffuse freely.

The chemical evolution of the reaction network explained above is evolved in MEDYAN using a variant of the Gillespie method (Gillespie, 1976) called the next reaction method (Gibson and Bruck, 2000). Briefly, for every reaction $R_\mu, \mu \in [0, N)$ in the reaction network, we can compute reaction propensity $a_\mu = c_\mu \gamma_\mu$. Here, $c_\mu$ represents the mesoscopic rate constant given by, $c_\mu = k_\mu (V_r/N_A)^{n-1}$, where $k_\mu$ is the rate constant of reaction of order $n$ happening in a compartment of volume $V_r$ and $N_A$ represents the Avogadro constant. $\gamma_\mu$ represents the degeneracy of the reaction. For example, for the reaction $A + B \rightarrow C$, $\gamma_\mu = N_A \times N_B$ where $N_A$ and $N_B$ represent copy number of species A and B respectively within a given compartment. Once the propensies are computed, the timestep ($\tau_\mu$) corresponding to each reaction can be computed randomly from $\tau_\mu = (1/a_\mu) ln(1/r_\mu)$, where $r_\mu$ represents a random number between 0 and 1. The reaction with the lowest $\tau_\mu$ is chosen and executed. Then, the subset of reactions $\mu$ that share reaction species with the executed reaction are chosen, and the corresponding propensies and time steps are updated. Next reaction method speeds up this process by storing a dependency graph corresponding to the reaction network and also using an indexed priority queue data structure to efficiently sort the $\tau_\mu$ values. In MEDYAN, each of the chemical events are also coupled with a geometric change in the filament shape. For example, considering the monomer length to be 2.7 nm, (de) polymerization reactions lead to filament extension (reduction) by 2.7 nm. Additionally, dendritic nucleation reaction results in the formation of an offspring filament at a 70° angle with respect to the parent filament. Thus, in addition to stochastic sampling, it is also essential to ensure MEDYAN generates mechanically equilibrated network architectures. To achieve this, MEDYAN uses a mechanical energy representation and an energy minimization protocol.

To ensure the physical realism of generated networks, in this study, we perform conjugate gradient energy minimization of actin networks after every 5 ms of reaction-diffusion perturbations.

Actin filaments are represented as a series of cylindrical fiber segments, each of length $L_{cyl} = 27$nm. As actin filaments resist extensile stresses, the segments are modeled as elastic springs with high spring stiffness ($k_{str} = 100$pN/nm). The stretching energy of a segment (i) is given by,

$$U_i^{str} = k_{str}/2(l - l_0)^2 \tag{3}$$

Here, the cylinder stretching constant is given by $k_{str}$, with current length $l$ and resting length $l_0$. In this study, we choose a resting length $l_0 = 27$ nm corresponding to ten monomers.

Additionally, the filaments can bend along the hinge points according to the bending constant ($k_{bend}$) obtained from the experimentally measured flexural rigidity ($EI = L_p k_B T$) of actin filaments. The bending constant is given by $l_{cyl} k_{bend} = L_p k_B T$, where $L_p, k_B$, and T represent persistence length of actin, Boltzmann constant, and temperature, respectively. In this study, we assume T = 298 K, and $k_B T = 4.11$ pN.nm. The corresponding bending energy of a hinge point between cylinders i and i + 1 is given by,

$$U_{i,i+1}^{bend} = k_{bend}(1 - \cos \theta_{i,i+1}) \tag{4}$$

In addition, we explicitly model the steric repulsion to prevent spatial overlap of actin filament segments. Consider cylinders i and j. The excluded volume potential between the two is given by,

$$U_{ij}^{vol} = K_{vol} \int\int_0^1 \frac{dsdt}{|\vec{r_i} - \vec{r_j}|^4} \tag{5}$$

In the above equation, the excluded volume constant is given by $K_{vol}$, and the distance between points on cylinder i and cylinder j is parameterized by representing the points on cylinder i with s and j with t. Please refer to Floyd et al (Floyd et al, 2021) for a detailed discussion on the excluded volume potential used here.

Finally, actin filament segments that are confined within the reaction volume through a repulsion potential with the planar boundary given as,

$$U_i^{boundary} = \epsilon_{boundary} e^{-d/\lambda} \tag{6}$$

Here, the boundary repulsion depends on the distance from the boundary (d), boundary repulsion energy $\epsilon_{boundary}$ and the screening length $\lambda$. Please refer to Table EV1 for a detailed description of the parameters used in this model (Chandrasekaran et al, 2022a; Chandrasekaran et al, 2022b; Footer et al, 2007; Fujiwara et al, 2002; Fujiwara et al, 2007; Gittes et al, 1993; Mahaffy and Pollard 2006).

The following potentials are considered between the parent and offspring filaments. To ensure that the parent and offspring filaments do not overlap and to ensure that the minus-end of the offspring filament remains at a definite position with respect to the parent filament, we consider a combination of stretching and bending potentials. The length of the bond connecting the binding site on the parent and minus-end of offspring filament is given by, $L_{bond}$ with a resting length $L_0$ and is assumed to have a stretching constant $k_{branch,str}$.

$$U_{branch,stretch} = k_{branch,str}(L_{bond} - L_0)^2 \tag{7}$$

The angle ($\theta_{parent,bond}$) between the parent filament and bond is allowed to fluctuate around $\pi/2$ depending on the bending energy parameter $k_{branch,bend,I}$.

$$U_{branch,bend} = k_{branch,bend,I}(1 - \cos(\theta_{parent,bond} - \pi/2)) \tag{8}$$

To ensure the angle between parent and offspring ($\theta_{parent,offspring}$) is ~70°, we consider a bending potential.

$$U_{branch,bend} = k_{branch,bend,II}(1 - \cos(\theta_{parent,offspring} - 1.222)) \tag{9}$$

To penalize any twist along the bond connecting F-actin binding site and offspring minus-end, we consider a dihedral potential.

$$U_{branch,dihedral} = k_{branch,dihedral} \cos(\vec{n1}, \vec{n2}) \tag{10}$$

$k_{branch,dihedral}$ represents the dihedral energy parameter that penalizes the angle between plane normals ($\vec{n1}, \vec{n2}$). $\vec{n1}$ is the normal of plane encompassing by parent filament vector and the bond vector. $\vec{n2}$ is the normal of plane encompassing the bond vector and the offspring filament vector.

The polymerization rate of filament ends that are close to the boundary is scaled according to the Brownian ratchet model (Mogilner and Oster, 1996).

$$k_{poly} = k_{poly,F=0} e^{-F/F_0} \tag{11}$$

Here, the polymerization rate under zero-force condition $k_{poly,F=0}$ of a filament end is modified to $k_{poly}$ when it experiences force $F$ and a corresponding characteristic force $F_0 = 1.5$ pN.

## Quantification of single particle intensity

To quantify the intensity of AF488-N-WASP and AF647-FBP17 particles, the images were analyzed using Trackmate (Tinevez et al, 2017) within ImageJ. Initially, a preliminary particle analysis was conducted manually, setting the estimated particle diameter to 0.65 μm (10 pixels) for particle detection. Additionally, quality control was performed to filter out false selections originating from the background, ensuring that only visible punctate protein signals were selected. The selected particles then underwent signal intensity analysis in Trackmate. Specifically, 200 out of more than 1000 particles from at least ten images were quantified for each protein combination to ensure statistical robustness.

## Curvature enrichment analysis of FBP17, N-WASP, and actin signal on SLB

The processing and analysis of nanoarray images were conducted using custom-written MATLAB code adapted from a previous study (Miao et al, 2022). In brief, individual nanobar positions in the lipid channel image were located using a square mask (71 × 71 pixels) centered at the nanobar. The same mask was then applied to generate individual nanobar images of the protein channel. Background correction of each individual nanobar image was performed by subtracting a local background image generated using the mesh grid function in MATLAB based on four ROIs at the corners of the image. Background-corrected individual nanobar images with the same nanobar dimensions were averaged across different arrays and experimental repeats to generate averaged images. To quantify the signal at the nanobar-end and nanobar-center, each nanobar was segmented into three ROIs (two nanobar-ends and a nanobar-center). The size of the nanobar-end ROI was adjusted according to the dimension of the nanobar to minimize covering the nearby non-curved center of the nanobar. Intensity values of pixels within each ROI were then integrated.

For comparison between lipid and protein channels, the intensities of background-corrected individual nanobar images were normalized to the percentages of 600 nm nanobar-center intensity acquired in the same image. Protein density at each nanobar was measured by the ratio of normalized protein intensity to normalized lipid intensity. Additionally, a montage of protein distribution on nanobars was obtained from averaged images using Fiji ImageJ. Statistical analysis was carried out using PRISM 10 (GraphPad).

## Curvature enrichment analysis of actin signal in U2OS cells

Background-corrected individual nanobar images with the same nanobar dimensions were averaged across different arrays and experimental repeats to generate averaged images as mentioned above. To quantify the signal at nanobar-end and nanobar-center, each nanobar was manually segmented into four ROIs (two nanobar-ends and a nanobar-center) as described in Fig. 3C. Mean intensity values of pixels within each ROI were measured. The mean intensity at each nanobar-end was normalized to the average mean intensity of two centers for each nanobar size. Additionally, the montage of Lifeact-mApple signal on nanobars was obtained

from averaged images using Fiji ImageJ. Statistical analysis was carried out using PRISM 10 (GraphPad).

## Statistics analysis

All experiments are carried out with two replicates and random sampling or otherwise indicated. All statistical analyses were done by GraphPad Prism (GraphPad). $P$ values were determined by unpaired $t$-test or one-way ANOVA, as appropriate (*$p < 0.05$, **$p < 0.01$, ***$p < 0.001$, ****$p < 0.0001$, and ns no significance).

## Data availability

Source data for all quantified results and raw microscopy images underlying the figures are provided as Source Data files with this manuscript. Additional related datasets have been deposited in Zenodo and are publicly available under the accession https://doi.org/10.5281/zenodo.17635319. Input files used for the simulations, along with the analysis scripts, will be available at https://github.com/RangamaniLabUCSD/FBP_NWASP_Nanopillar upon publication.

The source data of this paper are collected in the following database record: biostudies:S-SCDT-10_1038-S44318-025-00677-w.

## Peer review information

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

## Acknowledgements

We thank Dr. Min Wu (Yale University, USA) and Dr. Mingjie Zhang (SUSTech, China) for sharing plasmids for protein production and discussion. We also acknowledge the support of EEE N2FC, SPMS CDPT, and MSE FACTS for the fabrication and characterization of nanochips. This work was funded by the Singapore Ministry of Education (MOE) Academic Fund Tier 3 (MOE2019-T3-1-012 to YM MOE-MOET32020-0001 to WZ), Tier 2 (MOE-T2EP30220-0009 and T2EP30222-0043 to WZ), and Tier 1 (RG95/19 and RG93/22 to WZ); National Research Foundation Singapore under its Open Fund - Individual Research Grant (MOH-000955 to YM), National Research Foundation Singapore (NRF-NRFI08-2022-0012 to YM), National Institutes of Health, NIGMS R01-GM132106 to PR, and the Human Frontier Science Program Foundation (RGY0088/2021 to WZ), IDMxS (EDUN C-33-18-279-V12) to YM and WZ and the Start-up Grant from Nanyang Technological University to WZ).

## Author contributions

**Kexin Zhu**: Conceptualization; Data curation; Formal analysis; Validation; Investigation; Visualization; Methodology; Writing—original draft; Writing—review and editing. **Xiangfu Guo**: Conceptualization; Data curation; Formal analysis; Validation; Investigation; Visualization; Methodology; Writing—original draft; Writing—review and editing. **Aravind Chandrasekaran**: Conceptualization; Data curation; Software; Formal analysis; Validation; Investigation; Visualization; Methodology; Writing—review and editing. **Xinwen Miao**: Data curation; Formal analysis; Validation; Visualization; Methodology; Writing—review and editing. **Padmini Rangamani**: Conceptualization; Supervision; Funding acquisition; Validation; Visualization; Writing—original draft; Writing—review and editing. **Wenting Zhao**: Conceptualization; Resources; Data curation; Software; Formal analysis; Supervision; Funding acquisition; Validation; Investigation; Visualization; Methodology; Writing—original draft; Project administration; Writing—review and editing. **Yansong Miao**: Conceptualization; Resources; Data curation; Formal analysis; Supervision; Funding acquisition; Validation; Investigation; Visualization; Methodology; Writing—original draft; Project administration; Writing—review and editing.

Source data underlying figure panels in this paper may have individual authorship assigned. Where available, figure panel/source data authorship is listed in the following database record: biostudies:S-SCDT-10_1038-S44318-025-00677-w.

## Disclosure and competing interests statement

The authors declare no competing interests.

# Expanded View Figures

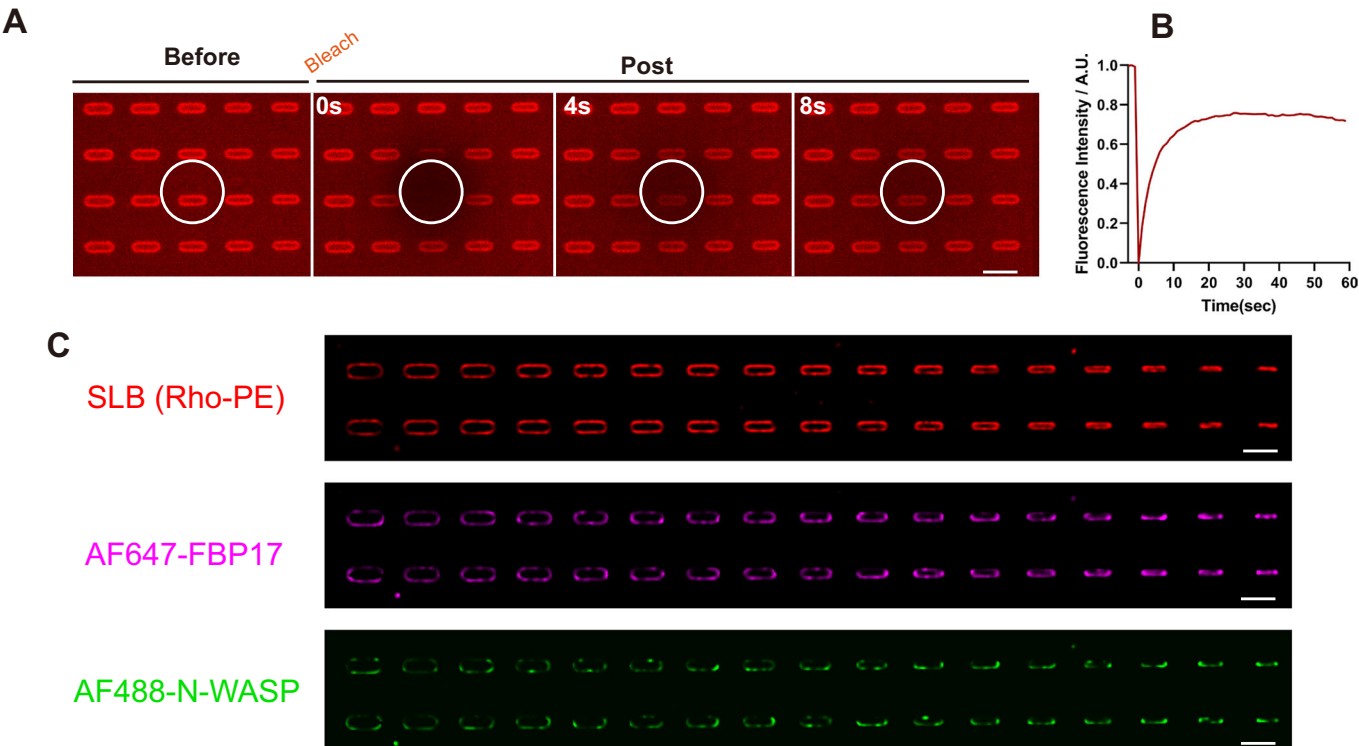

**Figure EV1. Curvature-dependent recruitment of FBP17 and N-WASP to membrane nanotubes.**

(A) Representative FRAP images showing lipid diffusion dynamics on SLBs (10% POPS: 88.5% POPC: 1% PI(4,5)P2:0.5% Rhodamine-PE). White circles indicate the bleached region. Scale bar: 5 μm. (B) Quantification of the normalized fluorescence recovery plot of SLB in (A). Data were normalized fluorescence intensities ($F/F_0$). (C) Preaveraged confocal microscopy images of SLB, AF647-FBP17 and AF488-N-WASP signal on curved SLB, 200–1000 nm bar width with 50 nm interval. Scale bars: 2 μm.

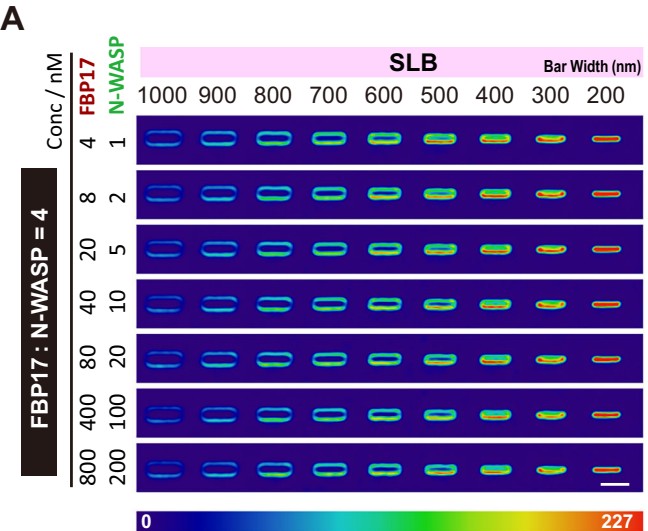

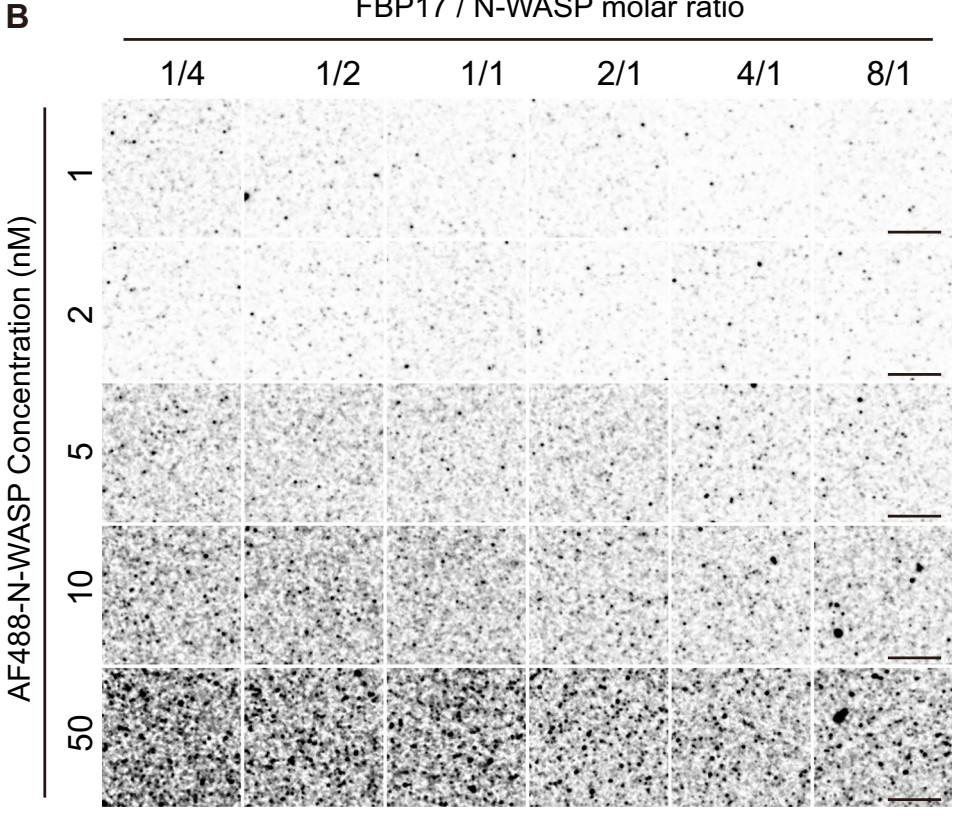

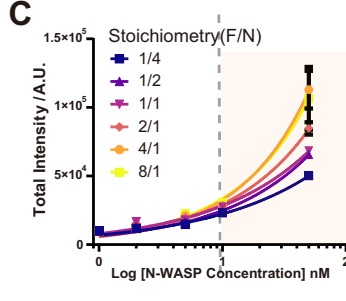

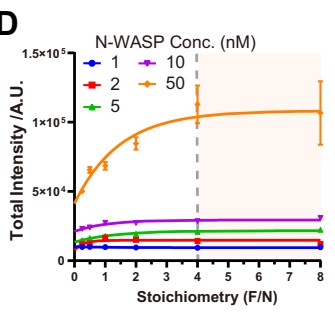

◀ **Figure EV2. N-WASP forms condensate with FBP17 in a concentration- and stoichiometry-dependent manner.**

(A) Heatmap representation of fluorescence intensities from SLBs at different bar widths (200–1000 nm). Each row corresponds to a fixed FBP17:N-WASP molar ratio of 4:1. Scale bar: 2 μm. (B) TIRFM single particle images of AF488-N-WASP at 1, 2, 5, 10, 50 nM on SLB with various stoichiometry of FBP17. Scale bar: 2 μm. (C) Plot of single particle intensity of the N-WASP as a function of concentration on flat SLB in (B). Lines are binding curves fitted with the Hill equation. $N = 1000$ from three repeated experiments, mean ± SEM are shown. (D) Total intensity plot of N-WASP single particles on SLB as a function of F/N stoichiometry on flat SLB in (B). $N = 1000$ from three biological repeats, mean and SEM are shown.

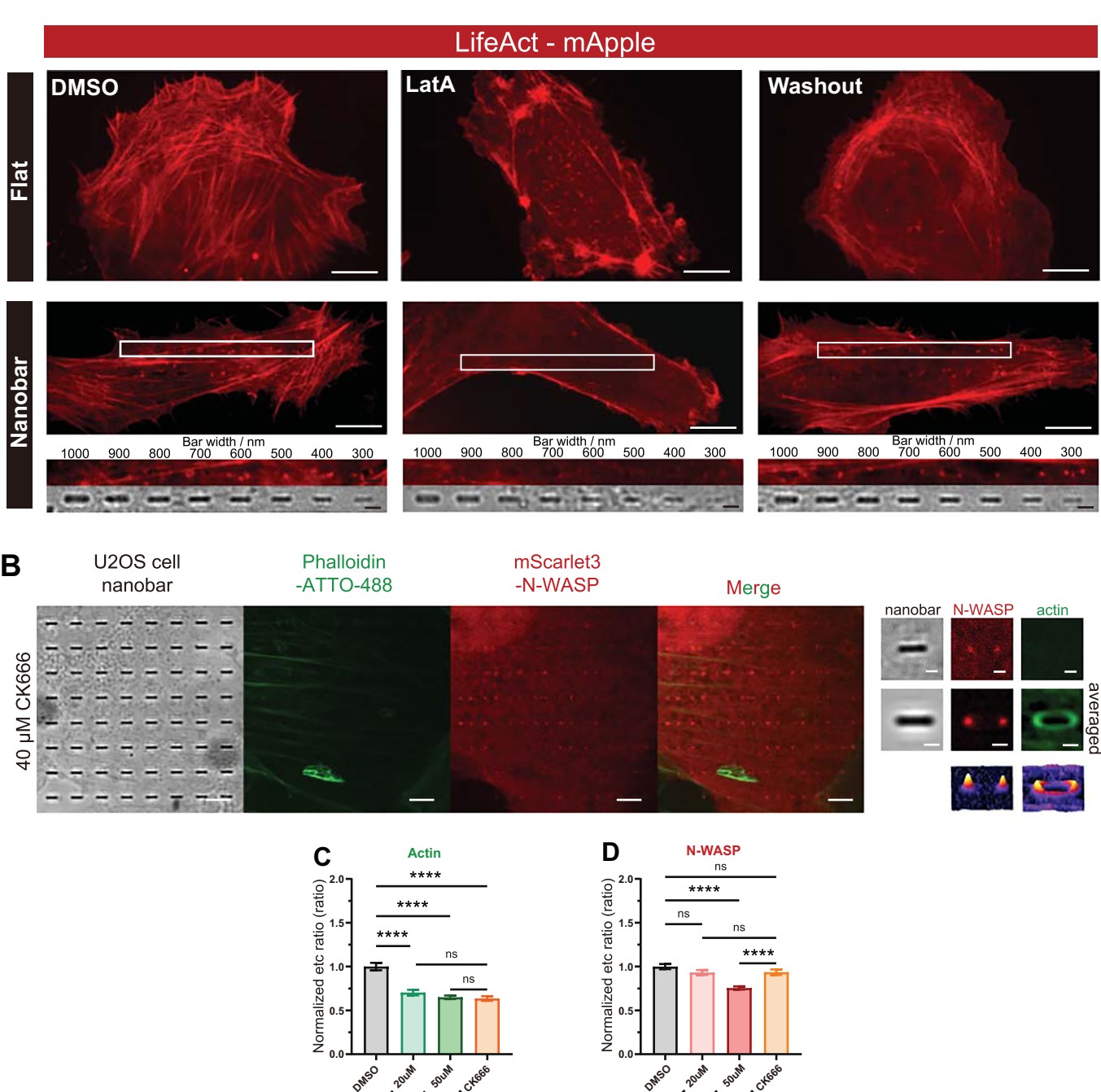

**Figure EV3. Actin dynamics and curvature sensing on flat and nanobar substrates.**

(A) Representative confocal images of Lifeact-mApple-labeled actin filaments in cells treated with DMSO (control), LatA, or after LatA washout, cultured on flat surfaces (top row) or nanobar arrays (bottom row). White boxes in nanobar images indicate regions of interest (ROIs) corresponding to bar widths ranging from 300 to 1000 nm. Scale bars: 10 μm. Zoomed-in views of nanobar ROIs, highlighting actin alignment and curvature adaptation across different bar widths (300–1000 nm). (B) Original and averaged confocal images of U2OS fixed cells cultured on 300-nm-wide nanobars with the expression of mScarlet3-N-WASP and the staining with Phalloidin-ATTO-488. Cells were fixed after a 1 h treatment with 40 μM CK666. Scale bar, 10 μm (left) and 1 μm (right). Contrast(averaged): 12–30. (C) Normalized Actin signal end-to-center ratio at nanobars 300 nm in width. Sample sizes for treatments under DMSO, 20 or 50 μM ML141, 40 μM CK666 were $N = 634, 324, 689$, and 425 nanobar ends, respectively. Each data point represents the mean ± SEM. (D) Normalized mScarlet-N-WASP signal end-to-center ratio at nanobars 300 nm in width. Sample sizes for treatments under DMSO, 20 or 50 μM ML141, 40 μM CK666 were $N = 634, 324, 689$, and 425 nanobar ends, respectively. Each data point represents the mean ± SEM. Statistical analysis was performed using one-way ANOVA followed by Tukey's multiple comparisons test (ns $p > 0.05$, ****$p < 0.0001$).

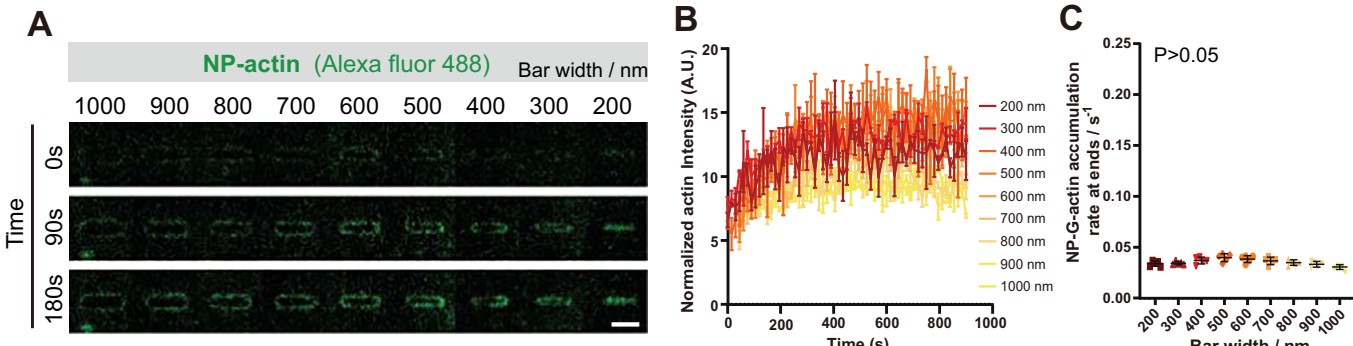

**Figure EV4. Curvature-dependent actin assembly in vitro.**

(A) Averaged confocal images of in vitro reconstitution of 1.5 µM NP-actin (30% AF488 labeled) recruitment at 0/90/180 s on the bilayer at nanobars of 200–1000 nm width with the presence of 200 nM FBP17, 50 nM N-WASP, and 5 nM Arp2/3. Scale bar, 2 µm. (B) Normalized signal density of NP-actin based on their corresponding lipid bilayer intensity. Each point represents mean ± SEM from over 15 nanobars. (C) NP-G-actin accumulation rate at nanobars of 200–1000 nm width. (The linear fitted slope within 105–300 s range in **B**). Each point represents mean ± SEM from over 15 nanopillars.

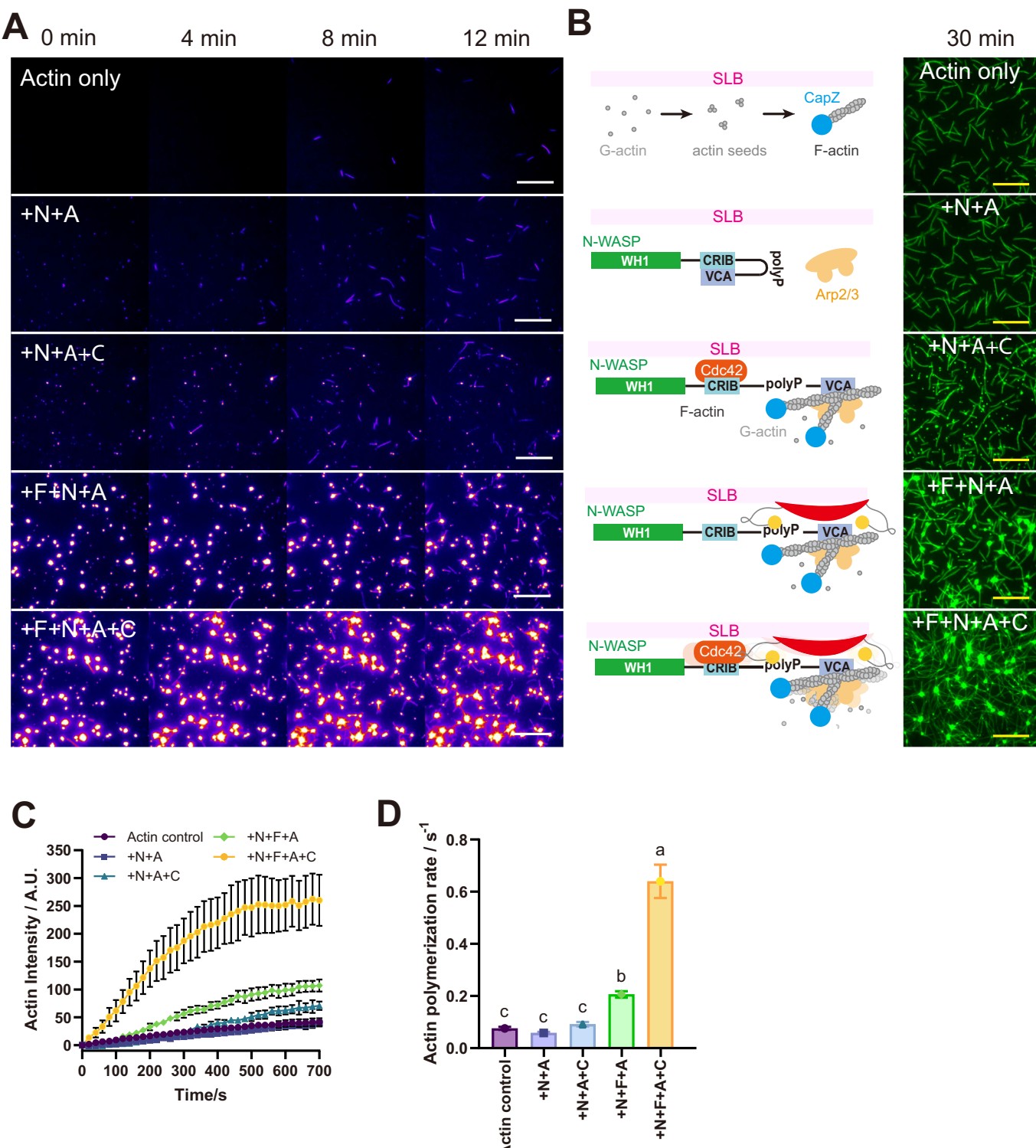

**Figure EV5. Synergistic effects of FBP17, N-WASP, and Cdc42 on actin polymerization with actin elongation inhibitor CapZ.**

(A) Time-lapse fluorescence microscopy images of actin polymerization under different reaction conditions: actin only (control), +N + A, +N + A + C, +F + N + A, and +F + N + A + C. Scale bar: 10 μm. (B) Right: Schematic representation of the experimental setup. The schematics illustrate actin polymerization on SLBs in different combinations of actin regulators (N-WASP, FBP17, Cdc42, Arp2/3, and CapZ). Left: Representative endpoint images (30 min) for each condition are shown. Scale bar: 10 μm. (C) Quantification of mean actin fluorescence intensity over time in (A). (D) Actin polymerization rates: the linear fitted slope within 105–350 s range in (C). $N = 9$. Each point represents mean ± SEM. Statistical analysis was performed using one-way ANOVA followed by Tukey's multiple comparisons test. Letters (a, b, c) denote statistical significance groups ($p < 0.05$).

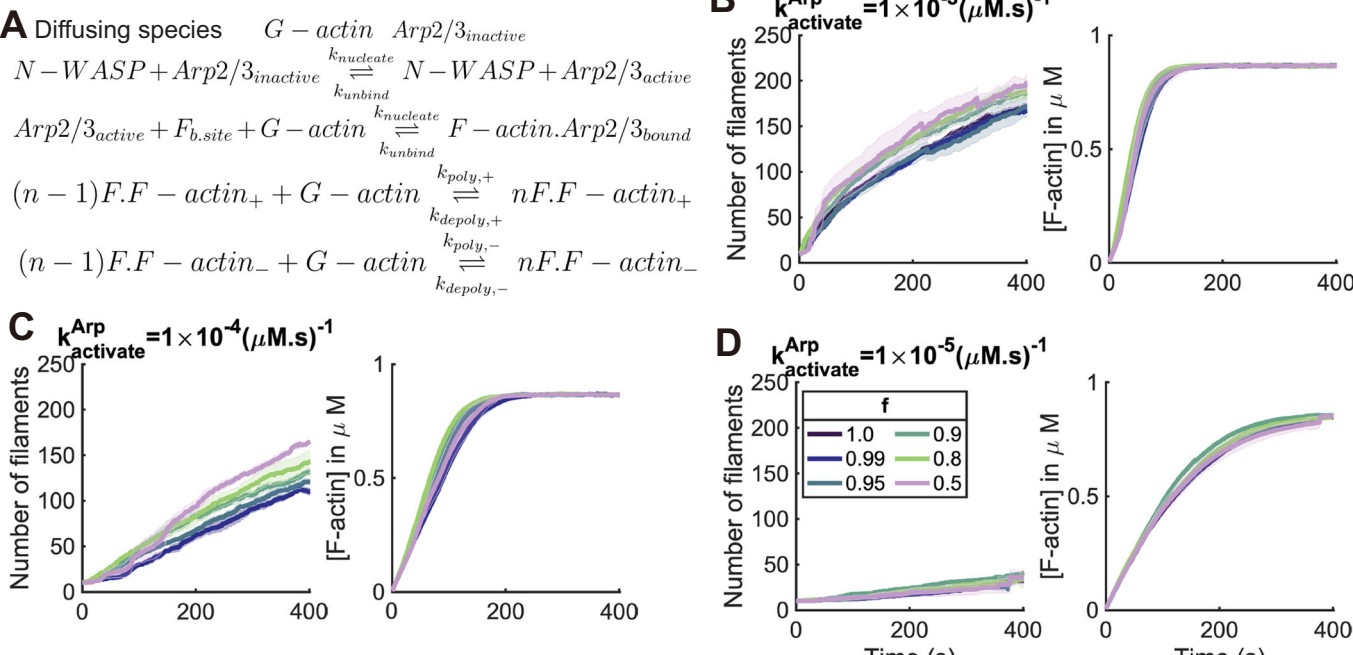

**Figure EV6. Spatial localization of N-WASP and N-WASP-driven activation rates are critical to enhance localized nucleation.**

(A) The set of chemical reactions considered in MEDYAN is shown. G-actin and inactive Arp2/3 are allowed to diffuse throughout the reaction volume. Arp2/3 is activated proportional to the local N-WASP concentration. Active Arp2/3 can bind to a binding site of F-actin (one per ten monomers in this study) to nucleate a new offspring filament. In addition, (de)polymerization reactions at both the plus and minus ends of filaments are considered. (B–D) The activation rate of Arp2/3 plays a critical role in controlling the cooperative nucleation process. Plots show mean and standard error of mean plots at various values of localization factor (f) as time series. Each row corresponds to a particular activation rate value.

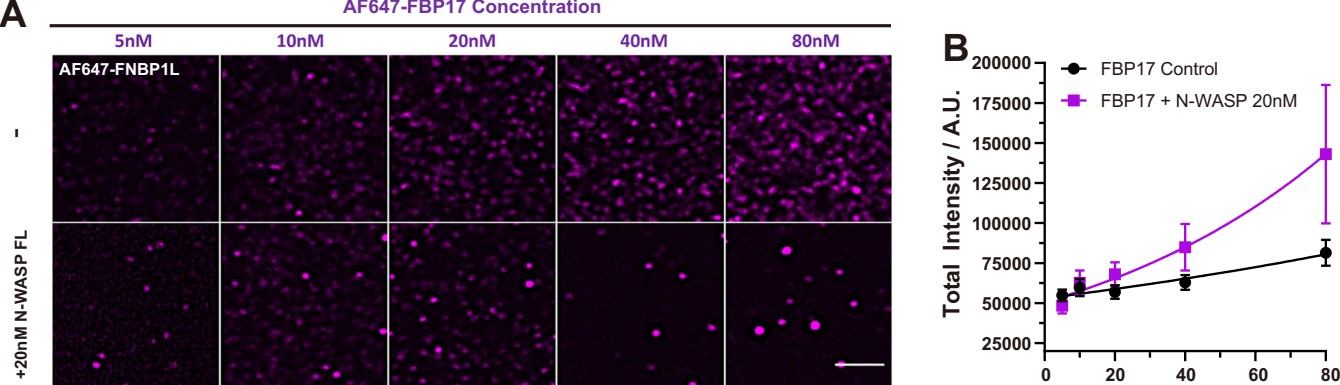

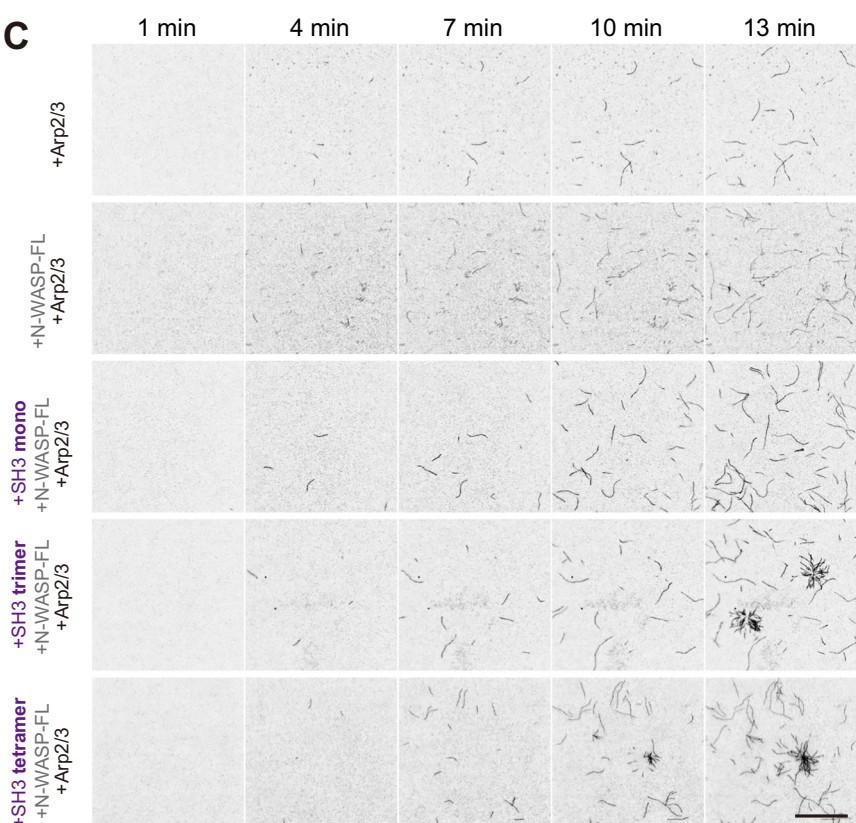

**Figure EV7. AF647-FBP17 single particle images and TIRF actin assembly assay of engineered SH3$^{FBP17}$.**

(A) TIRFM single particle images of AF647-FBP17 at 5, 10, 20, 40, 80 nM on SLB in the absence(upper)/presence(bottom) of 20 nM N-WASP FL. Scale bar, 5 μm. (B) Quantification of total fluorescence intensity of FBP17 as a function of concentration, with or without N-WASP in (B). (C) TIRF images of actin polymerization (10% Oregon-labeled) with different combinations of 80 nM FBP17/oTri-SH3/oTet-SH3, 20 nM N-WASP FL and 5 nM Arp2/3. Scale bar, 10 μm.

