## [Peer Review File · The EMBO Journal]

Membrane curvature initiates Cdc42-FBP17-N-WASP clustering and actin nucleation

Kexin Zhu, Xiangfu Guo, Aravind Chandrasekaran, Xinwen Miao, Padmini Rangamani, Wenting Zhao, and Yansong Miao

Corresponding authors: Yansong Miao (yansongm@ntu.edu.sg) , Wenting Zhao (wzhao@ntu.edu.sg), Padmini Rangamani (prangamani@ucsd.edu)

Review Timeline:

Submission Date:	7th Jun 24
Editorial Decision:	31st Jul 24
Appeal:	2nd Sep 24
Editorial Decision:	18th Sep 24
Revision Received:	13th Feb 25
Editorial Decision:	10th Mar 25
Appeal:	19th Sep 25
Editorial Decision:	28th Oct 25
Revision Received:	18th Nov 25
Accepted:	3rd Dec 25

Editor: Ieva Gailite

Transaction Report:

Dear Yansong,

Thank you for submitting your manuscript for consideration by The EMBO Journal. We have now received three reviewer reports on your manuscript, which are included below for your information. Based on the reviewer comments, we unfortunately had to conclude that the study is not a sufficiently strong candidate for publication in The EMBO Journal.

As you can see, while the reviewers find the topic per se interesting and reviewer #2 is more positive in their assessment, they also raise multiple and substantive concerns with the conclusiveness of the data, the experimental approach, and the transferability of the findings to a more physiological context, which is indicated by all reviewers. Additionally, reviewers #1 and #3 find the study in its current form of limited broader novelty and relevance. Since we require enthusiastic support from majority of our reviewers for publication here, and since all reviewers indicate that further-reaching analyses would be needed to elevate the manuscript to the level they expect for publication here, I am afraid that we cannot offer further consideration of the manuscript at The EMBO Journal.

While we cannot pursue this manuscript further, I would like to suggest a transfer to our not-for-profit open-access sister journal, Life Science Alliance (LSA). I have shared your manuscript and the accompanying reviews with LSA Executive Editor, Eric Sawey, who is interested in these findings, and would like to invite further consideration of this manuscript at LSA pending the following revisions:

- Address Reviewer 1's comments.
- Address Reviewer 2's comments.
- Address Reviewer 3's Major concerns #3 & 4 and the Minor concerns.

We understand that such a revision might need to be re-reviewed, in which case, Dr. Sawey will walk the Reviewers through our transfer process. If you are interested in this option, please use the link below to transfer your manuscript to LSA:

Link Not Available

You do not need to revise the manuscript before transferring it to LSA. Once you transfer, Dr. Sawey will email you an invitation to revise and resubmit, listing the same revision requests as mentioned above. Please feel free to reach out at e.sawey@life-science-alliance.org if you have any questions about the LSA journal, the transfer process or the revisions requested.

Thank you in any case for the opportunity to consider this manuscript. I am sorry that I could not offer better news this time, but I nevertheless hope that you will find the transfer offer of interest.

With kind regards,

Ieva

Referee #1:

K. Zhu et al report about novel results on the curvature-dependent local nucleation of actin structures mediated by N-WASP through its interaction with the F-BAR protein FBP17. They mostly use microfabricated structures with nanobars or pillars of controlled shape, coated by supported bilayers, together with purified proteins (FBP17, N-WASP, Arp2/3, G-actin). The method is not new but the results bring new insights about the interactions and the cooperativity between these molecules when interacting with a moderately curved membrane. In particular, the authors demonstrate that the positive curvature sensitivity of N-WASP depends on its interaction with FBP17, with an optimum N-WASP concentration of 20 nM for 80 nM FBP17. They also show FBP17 facilitates N-WASP-mediated activation of branched actin assembly and their localization. The conclusions related to this part of the paper are correct and justified. Nevertheless, the biological relevance of these results is not clear to me.

However, from the section starting at the line 326 on, the authors claim that FBP17 induces the "condensation" of N-WASP. This refers to a very specific physical transition, and they have no evidence here that a phase separation effectively takes place. Moreover, the observation that "... FBP17 also exhibits clustering behavior after adding N-WASP, and the size of FBP17 clusters is grown stoichiometric (Supplementary Fig. S5A,B). ..." is not supported by the data: Fig. S5B shows clustering of FBP17 even in the absence of N-WASP; the stoichiometric aspect is not quantified. Fig. 5I shows that the N-WASP clusters are more intense in the presence of FBP17. Finally, they use monomeric and trimeric SH3-domain chimera to show that a trivalent interaction with the disordered PolyP region of N-WASP induces N-WASP clustering and actin aster when monovalent one fails. A FBP17 dimer contains only 2 SH3 domains that are differently spaced than on the SH3 chimera and FBP17 is rigid. I imagine it is not possible to alter one SH3 domain only on the dimer, but I am not fully convinced that it is justified to conclude that "our results highlight the indispensable role of multivalent interaction to cluster and activate N-WASP via SH3 domain in curvature sensor FBP17" since the evidence is rather indirect.

The authors also develop a model to describe the curvature-induced sorting of FBP17 and N-WASP on nanobars. They suppose that FBP17 binds to hydrophobic defects in the membrane (that form in a curvature-dependent manner) through the insertion of amphipathic helices of the F-BAR domain. But the F-BAR domain of FBP17 does not contain amphipathic helices... The reference to the review of Gallop & McMahon is not correct. The F-BAR interacts with membrane via negatively charged lipids. Theoretical models already exist, based on curvature-matching, that would be more suitable, although not developed for the nanobar geometry. Regarding the other part of the model on enhanced actin polymerization due to clustering of N-WASP, I am not expert and cannot give an opinion.

To conclude, this paper brings some new results and understanding on the complex FBP17/N-WASP and its role on local actin polymerization. But the hypotheses of theoretical model on enrichment on nanobars are not suitable. Moreover, the section related to condensation and to the role of multivalency in the formation of local clusters should also be revised to consider this paper for publication in EMBO J.

Minor issues:

Fig. 2K: Mis-spelling (Curvatur)

Line 336: Fig. S5A,B. should be Fig 5B.

Line 337: "induce the condensation of N-WASP": Just the recruitment is shown, not the "condensation"

Line 379: "Membrane curvature not only responds to forces": what does it mean?

Line 392: (Almeida-Souza et al): wrong reference. This paper is on CME, not on clathrin-independent endocytosis.

Referee #2:

Review for « Membrane curvature catalyzes actin nucleation through nano-scale condensation of N-WASP-FBP17" by Zhu et al.

The manuscript entitled "Membrane curvature catalyzes actin nucleation through nano-scale condensation of N-WASP-FBP17" by Zhu et al. describes the use of a nanolithography method to control membrane curvature. The authors investigate the role of membrane curvature in the recruitment of BAR proteins and subsequently on the recruitment of actin nucleation-promoting factors and actin polymerization. The manuscript contains various experiments that detail the different steps from the detection of curvature by sensing molecules to the initiation of actin polymerization. The use of a reconstituted system, with precisely designed nanoarrays (previously described in other papers), allows for the decoupling of the processes of curvature detection and actin polymerization.

The main conclusions of the paper are:

- The protein FBP17 is able to precisely sense membrane curvature
- FBP17 and N-WASP form a complex with optimal stoichiometry in curvature-dependent manner
- The condensation of N-WASP drives actin nucleation
- Description of the mechanism by which nanoscale curvature precisely directs protein recruitment and macromolecular condensation to trigger actin nucleation locally.

Overall, the experiments presented in the manuscript are convincing but there are some points that should be addressed before recommendation for the publication of this article in EMBO Journal.

Main points:

1) The organization of Figures 1 and 2 is confusing. Is the same set-up used in both figures? The scheme at the beginning of Figure 2 suggests that the experimental set-up has changed from Figure 1, but I don't think this is the case. The authors should rearrange the figures to make them more comprehensible.

The same remark applies to the end of Figure 4 and Figure 5. There is a scheme at the beginning of Figure 5, but it is unclear whether this is the set-up used at the end of Figure 4. More generally, it is sometimes difficult to understand the exact conditions under which the experiments shown in the figures were carried out. This needs to be clarified.

2) About the experiments in cells:

a. The diagram in Figure 3A is not very clear and the side view is misleading for understanding what is happening in the experiment. It should be improved.

b. Sup Fig3A upper panel (DMSO): on the nanobars, the authors write "highlighted dense F-actin networks at the end of high-curvature regions above the nanobars". From the images, there is no sign of dense F-actin networks at the level of the nanobars. The authors should modify the sentence in the text.

c. It is not easy to understand what the authors mean with this experiment on cells. It does not provide any new information compared with what has been shown in previous studies (as in Lou et al. PNAS 2019 for example). The authors should either delete it or clearly specify the contribution of this experiment to the message of the paper.

3) To ensure that the polymerization observed in Figure 3 is branched actin polymerization and to complement the observations made in Supplemental Figure 3F, the authors should perform a control experiment without the Arp2/3 complex to confirm that the observed localization is not due to linear filament formation dependent on curvature. It would also be interesting to conduct the experiment without N-WASP to see if actin localization on the lipid bilayer is curvature-dependent. These two controls will strengthen the message of Figure 3, demonstrating that branched actin assembly can be modulated by membrane curvature.

Minor points:

1. not independent in terms of polymerization. However, from the movie, it is clear that each bar polymerizes its own actin network. The authors should redo the scheme to make this clearer.

2. Figure 3G: why is there a drop in actin intensity at 500 seconds?

3. In the legends, it is not easy to understand if the big N stands for independent replicates, field of view or nanobars observed. This information should be specified.

4. Sup Figure 3B: there is a typo in "Alexa Fluor".

5. The non-polymerizable G-actin does not appear in the method section. The source of this protein should be included.

6. Fig 4G: is the actin polymerization done on a SLB? This should be more explicit in the figure legend and in the text.

7. Figure 4I: colors should be explained in the legend or on the figure.

8. Figure 4I: Most of the FBP17 clusters do not have actin asters. How do the authors explain this phenomenon? This needs to be addressed for a better understanding of the results. Line 316: the authors refer to Sup Fig 3F to see the FBP17 at the center of actin asters but this is not clear at all from the image shown. More examples should be shown for this statement to be convincing.

9. In the methods "TIRF actin assembly assay in vitro", the authors write that the surface is conjugated with streptavidin. Could the authors explain why they do this step?

Referee #3:

Zhu et al. propose an interesting manuscript aiming to decipher the mechanism by which plasma membrane curvature induced by substrate nanotopography triggers actin nucleation via the condensation of the TOCA protein FBP17, and N-WASP. Through an elegant set of in vitro experiments and modeling, they show that FBP17, which has a strong affinity for highly curved membranes, causes the condensation of N-WASP at sites of high membrane curvature. Interestingly, they provide data indicating that the condensation of FBP17 and N-WASP occurs optimally within a specific range of concentrations and stoichiometry. Their study also outlines the temporal sequence of events: 1{degree sign}) membrane curvature is sensed by FBP17, 2{degree sign}) FBP17 induces N-WASP condensation at high curvature sites, and 3{degree sign}) N-WASP condensates subsequently recruit Arp2/3 complex, which induces actin polymerization. While experiments seem adequately performed and the figures and legends are generally clear and complete, important controls are lacking (see major concerns), and the statistical information could be improved (see minor concerns).

In addition, although this study is qualitative, its major limitation lies in the lack of in cellulo experiments to support the in vitro datasets. This significantly limits the scope of the study, as it leaves room for the argument that their findings may not apply to a biological/physiological context. Furthermore, the authors heavily emphasize the CDC42-independent nature of their described mechanism in the discussion. However, they do not provide any in vitro or in cellulo data addressing CDC42's involvement. Despite referencing the existence of previously described Rho GTPase-independent actin nucleation mechanisms in the discussion, the question remains unresolved for their mechanism, particularly in a physiological cellular context.

In conclusion, the findings of this study present limited novelty in their current form. A substantial amount of revision work would be necessary, including in cellulo data and a deeper exploration of the role of CDC42 in the mechanism, to make this story novel and of high interest and attractiveness for EMBO Journal readers.

Please find our major and minor concerns below.

Major concerns:

1{degree sign}) Role of CDC42 is not explored: Previous in cellulo studies have indicated that CDC42 is crucial for the recruitment of TOCA proteins to plasma membrane nanodeformations, such as CIP4 in Ledoux et al. 2023 (<https://doi.org/10.1126/sciadv.ade1660>). In that study, CIP4 recruitment to nanodeformations strongly depends on CDC42 expression and activity. They also show that the HR1 domain of CIP4 - which interacts with GTP-CDC42 (active form) - is essential for the specific recruitment of CIP4 to highly curved plasma membrane regions. A CIP4 mutant lacking HR1 domain is still recruited to plasma membrane, but less specifically to nanodeformations. In the current manuscript, Zhu et al. did not consider CDC42 in their analyses. In datasets such as in Fig.1C or D, the normalized end density of FBP17 is very close to 1, even on the thinnest nanobars (200 nm) where the value is around ~1.2. This indicates that the enrichment of the protein to high curvature regions of the membrane is not very high. One may wonder if a normalized end density of 1.2 is statistically significant (given the error intervals), which the authors should show. It is necessary and highly interesting to add GTP-loaded CDC42 in in vitro experiments presented in Fig.1 to 4 to see if this increases further the affinity of FBP17 to highly curved membranes, and subsequently the condensation of N-WASP and actin polymerization. This would also provide effective evidence on the putatively CDC42-independent nature of their mechanism, which should be strongly toned down if no additional data are provided.

2{degree sign}) Lack of in cellulo data to support the conclusions of the study: The only experimental data in a cellular model are provided in Fig. 3A-D and Fig. S3A. Unfortunately, experiments with Latrunculin A are not particularly informative. Additional conditions need to be tested to support to the in vitro data. For instance, the authors should look at actin end density on nanobars upon FBP17 (and possibly also CIP4 and FNBP1L) and N-WASP depletion using siRNAs. This cellular system could also be used to further explore if their mechanism is indeed CDC42-independent, and compare the responses obtained to their in vitro data. Without any further experimental evidence in vitro (see major concern #1) and in cellulo, it cannot be claimed that the mechanism described here is strictly CDC42-independent.

3{degree sign}) Lack of controls: Based on data presented in Figures 3 and 4B, the authors propose that FBP17 drives N-WASP-mediated actin assembly at highly curved plasma membrane. This appears overstated, as controls are lacking (e.g. without FBP17).

- In Fig. 4B: How do the authors explain that actin polymerization is already increasing on large nanobars (> 500 nm), where FBP17 and N-WASP are barely recruited? Is there an FBP17- and N-WASP-independent actin polymerization mechanism in vitro? This needs to be clarified.

- Experiments in Fig. 4B and S3F need to be repeated without FBP17 as a control condition (as in Fig. 1G) to unambiguously demonstrate that actin polymerization does not occur (or occurs to a lesser extent) without the specific recruitment of FBP17/N-WASP machinery.

4{degree sign}) N-WASP mutants (Fig. S5C): The authors should show if N-WASP mutants can still condensate in the presence of FBP17. They should also explore if these N-WASP mutants are still able to mediate Arp2/3 recruitment and actin polymerization. This should be explored in vitro and in cellular models. In the current version of the manuscript, N-WASP mutants seem underexploited and do not appear particularly useful.

Minor concerns:

1{degree sign}) The titles of some figures (e.g. Fig. 1, 3, S5...) and subheadings (e.g. line 222) are not informative. They should summarize the main findings of the figure or section.

2{degree sign}) In the abstract, the second sentence (lines 23-25) seems to lack a few words: "Yet, the mechanism by which dynamic membrane curvature prompts quick actin cytoskeletal changes in signaling remains elusive".

3{degree sign}) In the introduction and wherever necessary (e.g. discussion, line 389), TOCA-1 should be presented as FNBP1L, TOCA-2 as FBP17, and TOCA-3 as CIP4. Either one uses TOCA nomenclature, or the alternative names (FNBP1L/FBP17/CIP4), to avoid confusion.

4{degree sign}) Lines 72-75: Wrong citation of Ledoux et al., 2023. This study did not address the recruitment of WASP family proteins to highly curved membranes, but focused on TOCA proteins, CDC42 and actin polymerization.

5{degree sign}) Legend of Fig.1A: On the scheme, blue blocks should be captioned as proline-rich domains.

6{degree sign}) Line 115: Alexa Fluor (and not Flour).

7{degree sign}) Figure 2K, vertical axis title: "Curvature-gated" (and not "Curvatur-gated").

8{degree sign}) Line 191: We further explored (...).

9{degree sign}) Line 217: "FBP arrays" should be replaced by "FBP17 arrays".

10{degree sign}) Figure 2J: To make things clearer to the reader, it should be written that the concentrations indicated on the top of the chart are N-WASP concentrations.

11{degree sign}) Lines 262-263: The authors claim that actin cytoskeleton is "highly aligned with curvature end of nanobars" in reference to Fig. S3F. What they call alignment is not clear. The statement brings confusion, as the referenced figure does not show alignment of actin fibers with nanobars. This should thus be rephrased for clarity.

12{degree sign}) Fig. 4B: Data for FBP17 and N-WASP look the same as in Fig. 1D. Duplicated data must be avoided: either one of the datasets needs to be removed, or it should be specified somewhere on the figure or in the legend that a part of data are reused in Fig. 4B from Fig. 1D.

13{degree sign}) Lines 281-308: This paragraph describes a modeling procedure to predict how N-WASP localization can activate Arp2/3 complex activation. This paragraph should be better introduced and explained, as it is quite difficult to grasp for non-specialists. Also, it is not clear what this modeling part brings to the whole story. This should be either clarified or removed from the manuscript.

14{degree sign}) Lines 327-328: "To investigate... actin polymerization, we (...)". It seems words are lacking in the first part of the sentence. It needs to be rephrased.

15{degree sign}) Line 336: Shouldn't it be "stoichiometrically" rather than "stoichiometric"?

16{degree sign}) In several figures, such as in Fig.5H and I, statistical details are lacking. Fig.5H: Only one actin aster is shown (SH3 trimer condition). How representative is it? Is it a frequent or rare event? What are the numbers of asters observed independently in each experimental condition (SH3 mono vs SH3 trimer)? Fig. 5I: Statistical significance should be depicted.

17{degree sign}) Line 380: "sculpts" rather than "sculpt".

18{degree sign}) Methods, line 503: "The primary amine groups (...) were labeled...".

19{degree sign}) Methods: Values of labeling efficiencies of all proteins used in this study must be provided.

20{degree sign}) Methods, line 635: Remove the point after "10% FBS". PS is an acronym that is not explained before.

21{degree sign}) Methods, line 665: "... were mixed...".

22{degree sign}) In several figure legends (e.g. Fig.2L,M, Fig.5D,G...), the authors write that they quantified a certain number of particles (n = 200) out of over 10,000 particles. Why did they choose 200 particles, not more or not less? Why didn't they analyze all the particles identified? How did they choose the particles? In other figure panels (e.g. Fig. 5I), what does N represent? In Fig. 5I, is N=12 the number of images or the number of actin asters?

23{degree sign}) In addition, in each legend, it should be mentioned how many independent experiments were performed for each dataset. For each image, it should also be indicated the number of independent experiments they are representative of.

24{degree sign}) Statistical information regarding tests that were used and p-values should be reported in the legend of each figure. It should also be mentioned if data normality was checked for each dataset and how.

** As a service to authors, EMBO Press provides authors with the possibility to transfer a manuscript that one journal cannot offer to publish to another EMBO publication or the open access journal Life Science Alliance launched in partnership between EMBO Press, Rockefeller University Press and Cold Spring Harbor Laboratory Press. The full manuscript and if applicable, reviewers' reports, are automatically sent to the receiving journal to allow for fast handling and a prompt decision on your manuscript. For more details of this service, and to transfer your manuscript please click on Link Not Available. **

We sincerely appreciate the insightful and constructive feedback from all three reviewers. The reviewers recognized the novelty of our findings on curvature-dependent actin nucleation mediated by N-WASP and FBP17 but raised concerns about the clarity and biological relevance of some of our conclusions. Reviewer #1 questioned the evidence supporting the "condensation" of N-WASP and suggested revisions to our theoretical model. Reviewer #2 pointed out the need for better figure organization and additional control experiments to validate our findings, particularly regarding actin polymerization. Reviewer #3 emphasized the lack of in-cellulo data and the unexplored role of CDC42, which are crucial for demonstrating the physiological relevance of our mechanism.

We believe we can address all of the reviewers' concerns by conducting the additional experiments requested, refining our theoretical model, improving the clarity of our explanations, reorganizing the figures, and enhancing our discussion. Below, we provide a point-by-point response with revision plans for your review and reconsideration.

Referee #1:

- **Major Concerns:**

1. **Biological Relevance:** In particular, the authors demonstrate that the positive curvature sensitivity of N-WASP depends on its interaction with FBP17, with an optimum N-WASP concentration of 20 nM for 80 nM FBP17. They also show FBP17 facilitates N-WASP-mediated activation of branched actin assembly and their localization. The conclusions related to this part of the paper are correct and justified. Nevertheless, the biological relevance of these results is not clear to me.

Response: Response: We appreciate the reviewer's acknowledgment of our conclusions. Lou et al. (2019, PNAS) and Ledoux et al. (2023, Science Advances) reported that in cells, actin cytoskeleton initiation occurs at high curvature regions and depends on BAR domain proteins. Our in vitro reconstitution assays address a significant knowledge gap by revealing the molecular mechanisms through which a multivalent nano-condensation is initiated upon high curvature recognition and explaining the preference for specific curvature radii. To further enhance the biological relevance of our study, we plan to conduct in-cellulo assays involving the small GTPase CDC42, another known activation pathway. This will allow us to better distinguish between CDC42-mediated activation and curvature/condensation-mediated activation in cells.

2. **Indirect Evidence for Condensation:** However, from the section starting at the line 326 on, the authors claim that FBP17 induces the "condensation" of N-WASP. This refers to a very specific physical transition, and they have no evidence here that a phase separation effectively takes place.

Response: We agree with the reviewer's point that the term 'condensation' may not be specific enough for FBP-N-WASP condensates, especially concerning its broad scale in the context of general phase separation. Initially, a percolation clustering process forms an early-stage condensate by nucleating a core with limited diffusion, which can then evolve into larger condensates by adjusting stoichiometry and size (Lee *et al*, 2023; Miao *et al*, 2023). To avoid confusion for the broader cell biology audience, we will use the terms 'nano-condensation' or 'nano-clustering' to better reflect the assembly state of FBP-N-WASP at nano-curvature. Additionally, we will provide supplementary data to more clearly demonstrate the multivalent clustering-based assembly of FBP-N-WASP clusters with changes in stoichiometry.

3. Moreover, the observation that "... FBP17 also exhibits clustering behavior after adding N-WASP, and the size of FBP17 clusters is grown stoichiometric (Supplementary Fig. S5A,B). ..." is not supported by the data: Fig. S5B shows clustering of FBP17 even in the absence of N-WASP; the stoichiometric aspect is not quantified. Fig. 5I shows that the N-WASP clusters are more intense in the presence of FBP17.

Response: BAR-domain-containing proteins form dimers and retain the ability for dimer-dimer interactions between F-BAR domains, as reported by (Frost *et al*, 2008). Although curvature effects promote seeding and alignment into more stable oligomeric states, these proteins naturally exist in low-affinity, heterogeneous, low-oligomeric forms. A similar scenario can be seen in intermediate promiscuous assemblies. We propose performing interface-disrupting mutations on the BAR domain of FBP17 (F-BAR^{K66E} or F-BAR^{K166A}) to examine if this leads to more homogeneous assemblies, thereby supporting our interpretation.

In addition, we will quantify the stoichiometric growth of FBP17 clusters and present the results in the revised figures. Our original statement referred to the observed increase in cluster size when N-WASP was added, indicating that N-WASP enhances FBP17 clustering. Additionally, Fig. 5I shows increased intensity of N-WASP clusters in the presence of FBP17, further supporting the cooperative interaction between these proteins. We will clarify these points in the manuscript to better reflect our findings.

4. **Lack of Validation for the Design of Trimer SH3:** Finally, they use monomeric and trimeric SH3-domain chimera to show that a trivalent interaction with the disordered PolyP region of N-WASP induces N-WASP clustering and actin aster when monovalent one fails. A FBP17 dimer contains only 2 SH3 domains that are differently spaced than on the SH3 chimera and FBP17 is rigid. I imagine it is not possible to alter one SH3 domain only on the dimer, but I am not fully convinced that it is justified to conclude that "our results highlight the indispensable role of multivalent interaction to cluster and activate N-WASP via SH3 domain in curvature sensor FBP17" since the evidence is rather indirect.

Response: We appreciate the reviewer’s attention to the role of SH3 domain multivalency in N-WASP clustering and activation. In response, we revisited our experimental design and conclusions regarding the trimer SH3 chimera. While direct manipulation of individual SH3 domains within the dimer is challenging, our use of the trimeric SH3 chimera effectively tested the impact of increased valency. Our findings demonstrate that the trimer SH3 chimera induces N-WASP clustering and actin nucleation, reinforcing our hypothesis that multivalency is crucial (Figure 5).

However, address reviewer’s concern in the revision, we here propose a new design that we fuse two SH3 domains at both C- and N-terminal of oTri-CC. In this scenario, the trimeric assembly can mimic the F-BAR dimer-dimer oligomerization pattern which was reported in (Frost *et al.*, 2008). The below panel demonstrates the illustration and AlphaFold3 prediction of hexameric FBP17 and trimeric SH3-oTri-SH3.

5. **Model Inaccuracy for FBP17 Membrane Association:** The authors also develop a model to describe the curvature-induced sorting of FBP17 and N-WASP on nanobars. They suppose that FBP17 binds to hydrophobic defects in the membrane (that form in a curvature-dependent manner) through the insertion of amphipatic helices of the F-BAR domain. But the F-BAR domain of FBP17 does not contain amphipatic helices... The reference to the review of Gallop & McMahon is not correct. The F-BAR interacts with membrane via negatively charged lipids. Theoretical models already exist, based on curvature-matching, that would be more suitable, although not developed for the nanobar geometry.

Response: We apologize for the misunderstanding regarding the model. Our description was erroneously based on N-BAR proteins rather than F-BAR family of proteins. We have rectified the error in this revision by carefully explaining each reaction in the model along with the mechanistic intuition behind each. Further, we have added additional descriptions to clearly convey the assumptions made in the model. In brief, molecular simulations of lipid membranes (Domanska & Setny, 2024; Yesylevskyy et al, 2017) suggests that area-per-lipid increases as a function of membrane curvature. As FBP17 molecules bind to lipid head groups, we argue in our model that the increased area per lipid should improve access between binding sites on FBP17 and lipid head groups. Such optimally oriented lipid head groups are termed (H) in our model.

Referee #2:

- **Main Concerns:**

1. The organization of Figures 1 and 2 is confusing. Is the same set-up used in both figures? The scheme at the beginning of Figure 2 suggests that the experimental set-up has changed from Figure 1, but I don't think this is the case. The authors should rearrange the figures to make them more comprehensible.

The same remark applies to the end of Figure 4 and Figure 5. There is a scheme at the beginning of Figure 5, but it is unclear whether this is the set-up used at the end of Figure 4. More generally, it is sometimes difficult to understand the exact conditions under which the experiments shown in the figures were carried out. This needs to be clarified.

Response: We appreciate the reviewer's feedback and apologize for any confusion. The experimental set-up in Figures 1 and 2 is consistent; the scheme in Figure 2 was intended as an overview, not as an indication of any changes. We will revise the figures and captions for clarity.

For Figures 4 and 5, the experimental set-ups were different tailored for specific purpose: Figure 4 was conducted on a Biotin-PEG-coated glass slide, while Figure 5 used SLBs. We will adjust the schematics to match the experimental conditions in both figures and update the figure legends to clearly describe these conditions, ensuring better understanding of each figure's context.

2. **a)** The diagram in Figure 3A is not very clear and the side view is misleading for understanding what is happening in the experiment. It should be improved.

Response: We agree that the side view may be misleading and could be improved for better understanding. We will revise the diagram to provide a clearer and more accurate representation of the experiment, ensuring it more effectively conveys the key aspects of the setup.

b) The Description of Results: Sup Fig3A upper panel (DMSO): on the nanobars, the authors write "highlighted dense F-actin networks at the end of high-curvature regions above the nanobars". From the images, there is no sign of dense F-actin networks at the level of the nanobars. The authors should modify the sentence in the text.

Response: We appreciate the reviewer's observation regarding the description of dense F-actin networks in the upper panel of Supplementary Figure 3A. We will clarify this better in the revision. The images provided were an averaged composite of several individual images to highlight the actin clusters at the ends of the high-curvature regions above the nanobars, because the actin signal follows a dynamic on-off behavior (please see example in Lou et al. (PNAS 2019) Movie S1 and Zhao et al. (Nature Nanotechnology 2017) Figure 4A). To ensure clarity

and accuracy, we will revise the relevant text in the manuscript accordingly. The reason we have provided the original images were for better transparency. In the revision, we will provide a more detailed description of the image background and ensure clear alignment with the relevant analyzed data.

Figure 4A. Kymograph plots of dynamin2-GFP along lines of nanopillars show repeated appearance and disappearance of dynamin2 spots at nanopillar locations (Zhao et al, 2017)

c) Cellular Experiments Are Not Formative: It is not easy to understand what the authors mean with this experiment on cells. It does not provide any new information compared with what has been shown in previous studies (as in Lou et al. PNAS 2019 for example). The authors should either delete it or clearly specify the contribution of this experiment to the message of the paper.

Response: We appreciate the reviewer's comments, which highlighted the need to clarify our significance and how our findings differ from those of Lou et al. (PNAS 2019). Lou et al. demonstrated actin polymerization at the ends of nanobars and its colocalization with FBP17, dependent on FBP17, but did not provide mechanisms of molecular interplay at the complex actin assembly center. This center involves multiple components, including curvature sensing, actin nucleation, and other actin regulators, as well as considerations of topology and timing.

Our research offers a detailed molecular mechanism that explains the core reactions through a cascade of multivalent interactions. This process progresses from topological effects to nanoscale condensation, ultimately enhancing biochemical reactions. Unlike previous studies, our work focuses not just on dependency but on a spatially and temporally regulated molecular mechanism driven by physicochemical cues, guiding biochemical reactions through macromolecular condensation. These distinctions are the key differences between our current work and previously reported results. We will provide better discussion in the revised version.

- 3. Lack of Control Experiments:** To ensure that the polymerization observed in Figure 3 is branched actin polymerization and to complement the observations made in Supplemental Figure 3F, the authors should perform a control experiment without the Arp2/3 complex to confirm that the observed localization is not due to linear filament formation dependent on curvature. It would also be interesting to conduct the experiment without N-WASP to see if actin localization on the lipid bilayer is curvature-dependent. These two controls will strengthen

the message of Figure 3, demonstrating that branched actin assembly can be modulated by membrane curvature.

Response: We agree with the reviewer's suggestion and will conduct the recommended control experiments. To determine whether the observed actin polymerization is dependent on the Arp2/3 complex and to rule out the possibility of linear filament formation, we will conduct actin polymerization assay on supported lipid bilayers (SLBs) with nanobar patterns, incubated with FBP17, N-WASP, G-actin, CapZ, and Arp2/3 complex (as positive control) and parallelly perform two other groups one without N-WASP, one without Arp2/3 complex.

Referee #3:

- **Major Concerns:**

1. **Role of CDC42 is not explored:** Previous in cellulo studies have indicated that CDC42 is crucial for the recruitment of TOCA proteins to plasma membrane nanodeformations, such as CIP4 in Ledoux et al. 2023 (<https://doi.org/10.1126/sciadv.ade1660>). In that study, CIP4 recruitment to nanodeformations strongly depends on CDC42 expression and activity. They also show that the HR1 domain of CIP4 - which interacts with GTP-CDC42 (active form) - is essential for the specific recruitment of CIP4 to highly curved plasma membrane regions. A CIP4 mutant lacking HR1 domain is still recruited to plasma membrane, but less specifically to nanodeformations. In the current manuscript, Zhu et al. did not consider CDC42 in their analyses. In datasets such as in Fig.1C or D, the normalized end density of FBP17 is very close to 1, even on the thinnest nanobars (200 nm) where the value is around ~1.2. This indicates that the enrichment of the protein to high curvature regions of the membrane is not very high. One may wonder if a normalized end density of 1.2 is statistically significant (given the error intervals), which the authors should show. It is necessary and highly interesting to add GTP-loaded CDC42 in in vitro experiments presented in Fig.1 to 4 to see if this increases further the affinity of FBP17 to highly curved membranes, and subsequently the condensation of N-WASP and actin polymerization. This would also provide effective evidence on the putatively CDC42-independent nature of their mechanism, which should be strongly toned down if no additional data are provided.

Response: We appreciate the reviewer's insightful comments on the role of CDC42 in recruiting TOCA proteins to plasma membrane nanodeformations. We fully acknowledge the well-established role of CDC42 in activating actin polymerization through nucleation-promoting factors. Our intention was not to diminish the importance of CDC42, but rather to highlight a coupling activation mechanism centered on curvature-driven actin nucleation. We recognize the complexity of the dynamic interplay between CDC42-mediated and curvature-mediated activations, which may coexist and be difficult to decouple. This raises an important question about the nuanced relationships between different mechanisms of actin nucleation activation. We here propose to explore the synergistic or stepwise mechanisms of CDC42 with curvature/FBP17-N-WASP complex by conducting additional experiments.

Our aim is to elucidate a synergistic pathway that could operate alongside the CDC42-mediated pathway, driving curvature-mediated actin assembly. This aspect was largely underestimated in cell biology assays due to the dominant role of CDC42 in vivo and the challenges of dissecting multiple-component cascade reactions in living cell. The novelty of our work lies in uncovering a clustering mechanism uniquely induced by membrane curvature and mediated by multivalent SH3 domain interactions. The significance lies with cascade multivalent condensation-mediated enhancement of biochemical reaction, differing from dependency-based relationships, either curvature-actin coupling or CDC42-N-WASP

activation.

Although the reviewer's suggestions are more suitable for future studies and require extensive research, we recognize the value of these comments. Integrating CDC42 into our *in vitro* system could indeed strengthen our study. In the revision, to investigate the potential interplay between CDC42-driven and curvature-mediated N-WASP activation, we plan to conduct additional experiments. By examining how CDC42 and membrane curvature jointly affect FBP17 and N-WASP dynamics, we aim to offer a more comprehensive understanding of actin assembly regulation in response to complex cellular signals.

- Lack of In Cellulo Data to support the conclusions of the study:** The only experimental data in a cellular model are provided in Fig. 3A-D and Fig. S3A. Unfortunately, experiments with Latrunculin A are not particularly informative. Additional conditions need to be tested to support to the *in vitro* data. For instance, the authors should look at actin end density on nanobars upon FBP17 (and possibly also CIP4 and FNBP1L) and N-WASP depletion using siRNAs. This cellular system could also be used to further explore if their mechanism is indeed CDC42-independent, and compare the responses obtained to their *in vitro* data. Without any further experimental evidence *in vitro* (see major concern #1) and *in cellulo*, it cannot be claimed that the mechanism described here is strictly CDC42-independent.

Response: For the *in-cellulo* siRNA assays of FBP17, CIP4, and FNBP1L, we appreciate the reviewer's reference to the study by Ledoux et al. (2023) in *Science Advances*, which thoroughly examined the role of BAR family proteins in curvature sensing using nanobeads. Their findings showed that depleting the three TOCA family proteins, including FBP17, CIP4, and FNBP1L, significantly reduced actin accumulation at curved sites on 100 nm nanobeads, but not on 300 and 500 nm nanobeads. This strongly supports the importance of these proteins in curvature-induced actin assembly.

Our study, however, was designed to explore a different aspect: the visualization and quantification of newly polymerized actin via nucleation. By focusing on the mechanisms that initiate curvature-coupled actin nucleation, we aim to provide new insights into how membrane curvature, as sensed by FBP17, enhances actin nucleation through a cascade reaction that evolves depending on curvature radius. This approach allows us to explore the mechanism by which FBP17 leverages topological cues to biochemical reactions using cascade recruitment and condensation to enhance actin nucleation, which is beyond simply its dependence on BAR proteins.

While we understand the value of conducting additional experiments involving more BAR proteins and siRNA-mediated depletion of N-WASP, we respectfully suggest that reproducing these known findings may not significantly advance our mechanistic understanding of nucleation enhancement, and may extend beyond the current scope of our study.

In our revised manuscript, we will ensure to better clarify the unique mechanisms we have investigated and clearly differentiate our findings from previous studies, emphasizing how our work complements existing knowledge and can inform future studies on different BAR proteins in cellular contexts.

3. **Lack of Controls:** Based on data presented in Figures 3 and 4B, the authors propose that FBP17 drives N-WASP-mediated actin assembly at highly curved plasma membrane. This appears overstated, as controls are lacking (e.g. without FBP17).

- In Fig. 4B: How do the authors explain that actin polymerization is already increasing on large nanobars (> 500 nm), where FBP17 and N-WASP are barely recruited? Is there an FBP17- and N-WASP-independent actin polymerization mechanism in vitro? This needs to be clarified.

- Experiments in Fig. 4B and S3F need to be repeated without FBP17 as a control condition (as in Fig. 1G) to unambiguously demonstrate that actin polymerization does not occur (or occurs to a lesser extent) without the specific recruitment of FBP17/N-WASP machinery.

Response: We thank the reviewer for the insightful comments and agree that further clarification and additional controls are needed to support our conclusions. The increase in actin polymerization observed on larger nanobars (>500 nm), despite the low recruitment of FBP17 and N-WASP, reflected the effectiveness of the underlying cascade biochemical reactions. Even at nanomolar concentrations, active N-WASP is sufficient to initiate actin polymerization. In our reconstitution system, the 200 nm condition could still be more effective than the >500 nm condition, consistent with in-cellulo results reported by Ledoux et al. (2023) in *Science Advances*. However, in-cellulo conditions may involve additional unknown involving factors that compromise the reaction efficiency at curvatures greater than 500 nm. We will clearly discuss this in our revised manuscript to avoid any potential overstatements or misleading information.

To this end, we will conduct the suggested additional experiments where actin polymerization was assessed in the absence of FBP17 (and also in the absence of N-WASP where relevant) on both flat and patterned SLBs.

4. **Underutilization of N-WASP Mutants:** The authors should show if N-WASP mutants can still condensate in the presence of FBP17. They should also explore if these N-WASP mutants are still able to mediate Arp2/3 recruitment and actin polymerization. This should be explored in vitro and in cellular models. In the current version of the manuscript, N-WASP mutants seem underexploited and do not appear particularly useful.

Response: We appreciate the reviewer's suggestion to further investigate the behavior of N-WASP mutants in the presence of FBP17, and we agree that assessing the condensation of

these mutants is important. We are currently conducting additional experiments to evaluate the ability of N-WASP mutants to undergo clustering when co-incubated with FBP17.

Regarding the ability of N-WASP mutants to mediate Arp2/3 recruitment and actin polymerization, we would like to note that these aspects have been thoroughly investigated in several well-documented studies. For example, the work by (Kim *et al*, 2000; Rohatgi *et al*, 2000; SUETSUGU *et al*, 2004; Yarar *et al*, 2002) has extensively characterized how various N-WASP mutants interact with the Arp2/3 complex and regulate actin polymerization. These studies provide a comprehensive understanding of the functional capabilities of N-WASP mutants in these processes.

Figure 5 Activation of N-WASP by mDab1 in vitro. Activation of N-WASP by mDab1 in vitro Pyrene-actin assay to monitor Arp2/3-complex-mediated actin polymerization. Increase in fluorescence indicates actin polymerization. Actin polymerization induced by WT (A), WH1 (B) and proline-rich (C) N-WASP in the presence or the absence of PTB domain of mDab1 is shown. A.U., absorbance units. (S. Suetsugu *et al*. *Biochemical Journal*(2004))

Minor comments:

#13. Lines 281-308: This paragraph describes a modeling procedure to predict how N-WASP localization can activate Arp2/3 complex activation. This paragraph should be better introduced and explained, as it is quite difficult to grasp for non-specialists. Also, it is not clear what this modeling part brings to the whole story. This should be either clarified or removed from the manuscript.

We thank the reviewer for the feedback. We have rewritten this section using simpler language and have added additional details to ensure accessibility for readers of all levels of expertise. The changes made to the manuscript are shown below.

The simulations play a pivotal role in explaining the mechanistic origin of enhanced nucleation from N-WASP clustering. Experimental studies cannot tunably control the extent of NWASP localization

on SLBs. To overcome this limit, our model generates concentration profiles of NWASP at various levels of localization. We use this to study Arp2/3 activation. Further, the simulations show that adequate activation rate of Arp2/3 is necessary to see the localization-driven enhancement of Arp2/3 nucleation (Supplementary Figure 5). We quantify enhanced nucleation using both the number of filaments and the number branches per μm^2 .

We believe that these changes have significantly strengthened the manuscript, and we look forward to your further consideration.

References:

- Domanska M, Setny P (2024) Exploring the Properties of Curved Lipid Membranes: Comparative Analysis of Atomistic and Coarse-Grained Force Fields. *J Phys Chem B* 128: 7160-7171
- Frost A, Perera R, Roux A, Spasov K, Destaing O, Egelman EH, De Camilli P, Unger VM (2008) Structural Basis of Membrane Invagination by F-BAR Domains. *Cell* 132: 807-817
- Kim AS, Kakalis LT, Abdul-Manan N, Liu GA, Rosen MK (2000) Autoinhibition and activation mechanisms of the Wiskott–Aldrich syndrome protein. *Nature* 404: 151-158
- Lee DSW, Choi C-H, Sanders DW, Beckers L, Riback JA, Brangwynne CP, Wingreen NS (2023) Size distributions of intracellular condensates reflect competition between coalescence and nucleation. *Nature Physics* 19: 586-596
- Miao Y, Guo X, Zhu K, Zhao W (2023) Biomolecular condensates tunes immune signaling at the Host-Pathogen interface. *Curr Opin Plant Biol* 74: 102374
- Rohatgi R, Ho H-YH, Kirschner MW (2000) Mechanism of N-Wasp Activation by Cdc42 and Phosphatidylinositol 4,5-Bisphosphate. *Journal of Cell Biology* 150: 1299-1310
- SUETSUGU S, TEZUKA T, MORIMURA T, HATTORI M, MIKOSHIBA K, YAMAMOTO T, TAKENAWA T (2004) Regulation of actin cytoskeleton by mDab1 through N-WASP and ubiquitination of mDab1. *Biochemical Journal* 384: 1-8
- Yarar D, D'Alessio JA, Jeng RL, Welch MD (2002) Motility determinants in WASP family proteins. *Mol Biol Cell* 13: 4045-4059
- Yesylevskyy SO, Rivel T, Ramseyer C (2017) The influence of curvature on the properties of the plasma membrane. Insights from atomistic molecular dynamics simulations. *Scientific Reports* 7: 16078
- Zhao W, Hanson L, Lou H-Y, Akamatsu M, Chowdary PD, Santoro F, Marks JR, Grassart A, Drubin DG, Cui Y *et al* (2017) Nanoscale manipulation of membrane curvature for probing endocytosis in live cells. *Nature Nanotechnology* 12: 750-756

Dear Yansong,

Thank you for contacting me regarding the recent editorial decision on your manuscript. I apologise for the delay in responding to you, as I was out of office in the beginning of September and had to deal with a significant backlog after my return to the office on Monday.

I have now gone through your revision plan, and I appreciate that you are willing to engage in a major revision to address the reviewers' concerns. However, the outcome of the proposed extensive additional analysis is at this point uncertain. Furthermore, there are two aspects that were highlighted by reviewer #1 (whether the reported N-WASP recruitment by FBP17 represents condensation events) and both reviewers #1 and #3 (physiological relevance of the reported mechanism) that will be challenging to address. Based on these considerations, I am afraid that I cannot explicitly invite a revised manuscript.

Nevertheless, I appreciate the extensive nature of the proposed experiments and, should the revision work be successful and provide substantial additional support to your proposed model, I would be willing to reconsider the manuscript, while treating it as a new submission. In this case, I would send it back to the original reviewers for their re-assessment. In such cases, we allow reviewers to make comments on the newly provided data, which might then have to be further addressed.

I appreciate that you contacted us again for further discussion of your work, and I hope that the proposed approach sounds reasonable to you.

With best regards,

leva

leva Gailite, PhD
Senior Scientific Editor
The EMBO Journal
Meyerhofstrasse 1
D-69117 Heidelberg
Tel: +4962218891309
i.gailite@embojournal.org

** As a service to authors, EMBO Press provides authors with the possibility to transfer a manuscript that one journal cannot offer to publish to another EMBO publication or the open access journal Life Science Alliance launched in partnership between EMBO Press, Rockefeller University Press and Cold Spring Harbor Laboratory Press. The full manuscript and if applicable, reviewers' reports, are automatically sent to the receiving journal to allow for fast handling and a prompt decision on your manuscript. For more details of this service, and to transfer your manuscript please click on Link Not Available. **

Referee #1:

K. Zhu et al report about novel results on the curvature-dependent local nucleation of actin structures mediated by N-WASP through its interaction with the F-BAR protein FBP17. They mostly use microfabricated structures with nanobars or pillars of controlled shape, coated by supported bilayers, together with purified proteins (FBP17, N-WASP, Arp2/3, G-actin). The method is not new but the results bring new insights about the interactions and the cooperativity between these molecules when interacting with a moderately curved membrane. In particular, the authors demonstrate that the positive curvature sensitivity of N-WASP depends on its interaction with FBP17, with an optimum N-WASP concentration of 20 nM for 80 nM FBP17. They also show FBP17 facilitates N-WASP-mediated activation of branched actin assembly and their localization. The conclusions related to this part of the paper are correct and justified. Nevertheless, the biological relevance of these results is not clear to me.

Response: We appreciate the reviewer's positive feedback on our work and conclusions. In our revised manuscript, we introduce an additional layer of Cdc42-mediated nucleation regulation, which integrates more effectively with the existing knowledge framework and clarifies biological relevance through extensive new experiments. By performing comprehensive *in-cellulo* imaging and *in vitro* reconstitution studies, we investigated the interplay between curvature-induced local actin polymerization and Cdc42-driven actin nucleation in current Figure 3E-G, 4E-G, 6A-D. For example, according to the *in-cellulo* assay in Figure R1 (current Figure 3E-G), we demonstrate while the partial inhibition of small GTPase Cdc42 reduced actin polymerization without affecting N-WASP curvature localization, stronger inhibition impaired both. This indicates that Cdc42 regulates curvature-mediated actin assembly through two potential mechanisms: directly activating N-WASP for basal actin polymerization and facilitating its association with FBP17 at curved membranes. Since Cdc42 plays a well-established role as signaling molecule during endocytosis, filopodia formation, and membrane protrusions for cell migration, these findings provide a mechanistic link between membrane-curvature-regulated actin polymerization dynamics with various cellular pathways. We now discuss these implications in the revised manuscript.

Figure R1: Effects of ML141 and CK666 on Actin Cytoskeleton and N-WASP Localization in U2OS Cells (Current Figure 3E-G).

A. Original and averaged confocal images of U2OS fixed cells cultured on 300nm-wide nanobars with the expression of mScarlet3-N-WASP and the staining with Phalloidin-ATTO-488. Cells were fixed after a 1h treatment with DMSO, 20μM or 50μM ML141. Scale bar, 10 μm (left) and 1 μm (right).

B. Actin signal end-to-center ratio at nanobars 300nm in width. Sample sizes for treatments under DMSO, 20uM or 50uM ML141 were N=634, 324, 689 nanobar ends, respectively. Each data point represents the mean ± SD.

C. mScarlet-N-WASP signal end-to-center ratio at nanobars 300nm in width. Sample sizes for treatments under DMSO, 20uM or 50uM ML141 were N=634, 324, 689 nanobar ends, respectively. Each data point represents the mean ± SD.

Statistical analysis was performed using one-way ANOVA followed by Tukey's multiple comparisons test (ns p>0.05, **** p<0.0001).

However, from the section starting at the line 326 on, the authors claim that FBP17 induces the "condensation" of N-WASP. This refers to a very specific physical transition, and they have no evidence here that a phase separation effectively takes place.

Response: We acknowledge the reviewer's observation that the term "condensation" may not be immediately clear and could be more precisely described as "sub-percolation clustering." Percolation theory examines the formation and properties of clusters with variable interactive surfaces, focusing on the connectivity and networking of spanning clusters above critical thresholds. In contrast, condensation is a broader term refers to a phase transition where molecules assemble into a denser phase. These concepts are interconnected in the study of phase separation: sub-percolation clustering can represent an early-stage condensate, where nucleation forms a core with limited diffusion, which

can then evolve into larger condensates by adjusting stoichiometry and size (Lee *et al*, 2023; Miao *et al*, 2023).

The FBP17–N-WASP complex engages in multivalent interactions through the SH3 domain and polyproline motifs, which is a pair of well-studied phase separation system (Harmon *et al*, 2017). Likely due to other interaction surfaces and conformational constraints, this complex does not form large, fluidic droplets characteristic of liquid-liquid phase separation. Instead, it exhibits sub-percolation clustering, resulting in smaller, less dynamic assemblies. Additionally, the nanoscale curvature of the membranes studied—on the order of hundreds of nanometers—poses challenges for resolving the internal molecular dynamics of these clusters using standard microscopy techniques.

To provide clarity for the broader cell biology community, we have revised the manuscript and referred to these assemblies as "nano-clusters", accurately reflecting the state of FBP17–N-WASP complexes at nanoscale curvature. Figure R2 illustrates the multivalent clustering of FBP17–N-WASP, highlighting dynamic changes in stoichiometry. Time-lapse imaging further demonstrates a phase transition from the dilute phase to the dense phase, characterized by the formation of nanoscale clusters of FBP17–N-WASP. This process exhibits dynamic coarsening behavior over time, as shown in Figure R3, highlighting the flexible condensation properties.

Figure R2 (Supplementary Figure S2B-D)

Figure R2. N-WASP forms nanoclusters with FBP17 in a concentration- and stoichiometry-dependent manner

B. TIRFM single particle images of AF488-N-WASP at 1, 2, 5, 10, 50 nM on SLB with various stoichiometry of FBP17. Scale bar, 2 μ m.

C. Plot of single particle intensity of the N-WASP as a function of concentration on flat SLB in (B). Lines are binding curves fitted with the Hill equation. N=1000 single particles from 3 repeated experiments, mean and SEM are shown.

D. Total intensity plot of N-WASP single particles on SLB in (A). N=1000 single particles from 3 biological repeats, mean and SEM are shown.

Revision Figure R3

Figure R3. N-WASP clustering and coarsening in the presence of FBP17. Confocal images of 20 nM AF488-N-WASP on SLB with (lower panel) or without (upper panel) 80 nM FBP17. Scale bar, 5 μm .

Moreover, the observation that "... FBP17 also exhibits clustering behavior after adding N-WASP, and the size of FBP17 clusters is grown stoichiometric (Supplementary Fig. S5A,B). ..." is not supported by the data: Fig. S5B shows clustering of FBP17 even in the absence of N-WASP; the stoichiometric aspect is not quantified. Fig. 5I shows that the N-WASP clusters are more intense in the presence of FBP17.

Response: BAR domain-containing proteins, such as FBP17, form dimers and can retain the ability for dimer-dimer interactions (Frost *et al*, 2008a). While membrane curvature facilitates the nucleation and alignment of these proteins into more stable oligomeric structures, they typically exist in low-affinity, heterogeneous, low-oligomeric forms. This behavior is similar to that observed in intermediate promiscuous assemblies (McDonald *et al*, 2015).

In the revised Supplementary Figure S7 B,C (Figure R4), we provide quantitative analysis demonstrating the stoichiometric growth of FBP17 clusters, but its clustering is further enhanced by N-WASP through multivalent interactions, suggesting the macromolecular multivalent interactions promoted FBP17 clustering via nested hierarchical assembly. Furthermore, current Figure 5D shows the enhanced intensity of N-WASP clusters in the presence of FBP17, supporting the cooperative interaction between these proteins. We have clarified these findings in the manuscript to accurately reflect our results.

Revision Figure R4 (Supplementary Figure S7B,C)

Figure R4. N-WASP enhances FBP17 clustering and coarsening.

B) TIRF single particle images of 20nM N-WASP on SLB with 5, 10, 20, 40, 80nM AF648-FBP17. Scale bar, 5 μ m.

C) Plot of single particle intensity of the AF647-FBP17 as a function of concentration on flat SLB in (B). Lines are binding curves fitted with the Hill equation. N=1000 from 3 repeated experiments, mean and SEM are shown.

Finally, they use monomeric and trimeric SH3-domain chimera to show that a trivalent interaction with the disordered PolyP region of N-WASP induces N-WASP clustering and actin aster when monovalent one fails. A FBP17 dimer contains only 2 SH3 domains that are differently spaced than on the SH3 chimera and FBP17 is rigid. I imagine it is not possible to alter one SH3 domain only on the dimer, but I am not fully convinced that it is justified to conclude that "our results highlight the indispensable role of multivalent interaction to cluster and activate N-WASP via SH3 domain in curvature sensor FBP17" since the evidence is rather indirect.

Response: Multivalent interactions between SH3 domains and polyproline-rich motifs primarily depend on the strength and number of these interactions, with connectivity playing a more crucial role than local conformation (Harmon *et al.*, 2017). However, conformational constraints can hinder connectivity when spacers are short and lack flexibility. In this context, SH3 oligomer acts as a 'linker' or 'scaffolder', while N-WASP provides the 'spacer' function. Therefore, clustering at one end is not expected to impair the formation of multivalent interactions with N-WASP. A similar synthetic protein engineering was also successful in demonstrating an actin-binding protein coronin's valency in regulating Arp2/3 activities (Han *et al.*, 2023). Furthermore, the valency provided by SH3 domains is essential for guiding actin assembly on nano-curvature; notably, deletion of the SH3 domain abolishes actin assembly on nano-curvature in cellulo (Lou *et al.*, 2019).

We appreciate the reviewer's attention to the role of SH3 domain multivalency in N-WASP clustering and activation. In response, we revisited our experimental design and conclusions regarding the trimer SH3 chimera. While direct manipulation of individual SH3 domains within the dimer is challenging, our use of the trimeric SH3 chimera effectively mimics the valency when it's in side-by-side oligomerization state (Frost *et al.*, 2008b), with half of the oligomerized FBP17.

Revision Figure R5

Figure R5. Schematic and structural representation of FBP17 and its engineered oligomeric SH3 construct.

(A) Schematic illustration of FBP17, highlighting its F-BAR domain (red) and SH3 domains (yellow). The inset shows the engineered oTri-CC construct (cyan), which replaces the F-BAR domain to examine the role of multivalent interaction.

(B) Structural representation of FBP17 and oTri-SH3 from AlphaFold server. The F-BAR domain (red) forms a side-by-side scaffold, while the SH3 domains (yellow) are flexibly linked. The inset depicts the designed oTri-CC construct (cyan), showing its trimeric coiled-coil architecture and interaction with the SH3 domains (gold), mimicking F-BAR-mediated side-by-side oligomerization.

Secondly, to directly investigate curvature-induced multivalent interaction, we expressed an ENTH-SH3^{FBP17} protein, which is monomeric but can sense membrane curvature, in U2OS cells cultured on the regular nanobar chip (300nm). By observing the end enrichment of N-WASP signals and the actin distribution, we can clearly tell the role played by multivalency in this process. (Figure R6). Our quantitative fluorescence analysis reveals that both FBP17 and ENTH-SH3 sense membrane curvature with similar efficiency, as indicated by comparable end-to-center enrichment ratios on nanobar structures (Figure R6B-C). However, their ability to recruit N-WASP differs markedly. While FBP17 induces strong N-WASP clustering at nanobar ends, ENTH-SH3 shows significantly weaker recruitment (Figure R6B-D). This difference is further reflected in actin polymerization patterns—FBP17 induces a curvature-enriched actin network at highly curved membrane regions, whereas ENTH-SH3 induces a lower level of curvature-sensitive actin organization, leading to a more diffused actin distribution (Figure R6B-E). These results establish that without hierarchical nested multivalent assembly, curvature sensing is insufficient for efficient N-WASP clustering and downstream actin polymerization; instead, multivalent interactions through oligomeric SH3 domains are indispensable for organizing N-WASP into actin assemble hubs.

Revision Figure R6 (current figure 8A-F)

Figure R6. ENTH-SH3^{FBP17} impairs N-WASP recruitment and activation at curved ends.

- A. Schematic representation of curvature-sensing proteins FBP17 and ENTH-SH3 chimera on a curved membrane. Both constructs sense curvature, but only FBP17, with its dimeric SH3 domains, enables multivalent interactions with N-WASP, promoting clustering and actin assembly. ENTH-SH3, a monomeric chimera, lacks this multivalency.
- B. Representative fluorescence images of U2OS cells expressing GFP-FBP17 (top) or GFP-ENTH-SH3 (bottom) along with mScarlet-N-WASP and Phalloidin-ATTO-647 labeled actin on nanobar structures. Scale bars, 10 μ m.
- C. High-magnification images (top) and averaged images (bottom) of nanobar regions (boxed in B), showing FBP17 (green) or ENTH-SH3 (green), N-WASP (red), and actin (magenta) localization. Scale bars, 1 μ m.

Quantification of fluorescence signal ratios at nanobar ends relative to the center (etc ratio) for GFP (D, FBP17 vs. ENTH-SH3), mScarlet-N-WASP (E), and Cy5-actin (F).

The authors also develop a model to describe the curvature-induced sorting of FBP17 and N-WASP on nanobars. They suppose that FBP17 binds to hydrophobic defects in the membrane (that form in a curvature-dependent manner) through the insertion of amphipatic helices of the F-BAR domain. But the F-BAR domain of FBP17 does not contain amphipatic helices... The reference to the review of Gallop & McMahon is not correct. The F-BAR interacts with membrane via negatively charged lipids. Theoretical models already exist, based on curvature-

matching, that would be more suitable, although not developed for the nanobar geometry. Regarding the other part of the model on enhanced actin polymerization due to clustering of N-WASP, I am not expert and cannot give an opinion.

Response: We apologize for the confusion regarding the model. Our description was erroneously based on N-BAR proteins rather than the F-BAR family of proteins. We rectified the error in this revision by carefully explaining each reaction in the model and the mechanistic intuition behind each. Further, we have added additional descriptions to clearly convey the assumptions made in the model. In brief, molecular simulations of lipid membranes (Domanska & Setny, 2024; Yesylevskyy et al, 2017) suggest that area-per-lipid increases as a function of membrane curvature. As FBP17 molecules bind to lipid head groups, we argue in our model that the increased area per lipid should improve access between binding sites on FBP17 and lipid head groups. Such optimally oriented lipid head groups are termed (H) in our model.

To conclude, this paper brings some new results and understanding on the complex FBP17/N-WASP and its role on local actin polymerization. But the hypotheses of theoretical model on enrichment on nanobars are not suitable. Moreover, the section related to condensation and to the role of multivalency in the formation of local clusters should also be revised to consider this paper for publication in EMBO J.

Minor issues:

Fig. 2K: Mis-spelling (Curvatur)

Apologies for the errors. We have now corrected the spelling in current Fig 2M.

Line 336: Fig. S5A,B. should be Fig 5B.

We apologize for the confusion. Fig. S5A,B is different from Fig 5B. Although Figure 5B already showed the FBP17 clusters, Figure S5A,B (new Figure S7B,C) aimed to provide more information like quantification and stoichiometric growth of FBP17 clusters to support the data analysis.

Line 337: "induce the condensation of N-WASP": Just the recruitment is shown, not the "condensation"

We have modified the word to "nanoclusters". The graph in new Figure 7D (previous Figure 5D) showed the increased total intensity of N-WASP particle also indicating the clustering of N-WASP. The reviewer can also refer to Figure R3, which directly shows the clustering and coarsening of N-WASP after adding FBP17.

Line 379: "Membrane curvature not only responds to forces": what does it mean?

We have modified the DISCUSSION for better readability starting from page 14.

Line 392: (Almeida-Souza et al): wrong reference. This paper is on CME, not on clathrin-independent endocytosis.

Thanks for pointing this out. We have now corrected the reference in Page 15.

“There, F-BAR proteins coordinate with other proteins in curvature-related processes such as clathrin-independent endocytosis and the formation of membrane tubules (Roberts-Galbraith & Gould, 2010), suggesting a role for F-BAR-mediated multivalency.”

Referee #2:

Review for « Membrane curvature catalyzes actin nucleation through nano-scale condensation of N-WASP-FBP17" by Zhu et al.

The manuscript entitled "Membrane curvature catalyzes actin nucleation through nano-scale condensation of N-WASP-FBP17" by Zhu et al. describes the use of a nanolithography method to control membrane curvature. The authors investigate the role of membrane curvature in the recruitment of BAR proteins and subsequently on the recruitment of actin nucleation-promoting factors and actin polymerization. The manuscript contains various experiments that detail the different steps from the detection of curvature by sensing molecules to the initiation of actin polymerization. The use of a reconstituted system, with precisely designed nanoarrays (previously described in other papers), allows for the decoupling of the processes of curvature detection and actin polymerization.

The main conclusions of the paper are:

- The protein FBP17 is able to precisely sense membrane curvature
- FBP17 and NWASP form a complex with optimal stoichiometry in curvature-dependent manner
- The condensation of N-WASP drives actin nucleation
- Description of the mechanism by which nanoscale curvature precisely directs protein recruitment and macromolecular condensation to trigger actin nucleation locally.

Overall, the experiments presented in the manuscript are convincing but there are some points that should be addressed before recommendation for the publication of this article in EMBO Journal.

Main points:

1) The organization of Figures 1 and 2 is confusing. Is the same set-up used in both figures? The scheme at the beginning of Figure 2 suggests that the experimental set-up has changed from Figure 1, but I don't think this is the case. The authors should rearrange the figures to make them more comprehensible.

Response: We apologize for the confusion regarding the experimental setups depicted in Figures 1 and 2. The schematic in Figure 2 was intended as an overview and does not indicate any changes from the setup shown in Figure 1. To enhance clarity, we have revised the layouts and captions of Figure 1 and 2 in the manuscript.

The same remark applies to the end of Figure 4 and Figure 5. There is a scheme at the beginning of Figure 5, but it is unclear whether this is the set-up used at the end of Figure 4. More generally, it is sometimes difficult to understand the exact conditions under which the experiments shown in the figures were carried out. This needs to be clarified.

Response: For Figures 4 and 5, the experimental set-ups were different and tailored for specific

purposes: Experiments were conducted on biotin-PEG-coated glass slides for previous Figure 4 and supported lipid bilayers for previous Figure 5. We have now updated the schematics (current Figure 5G and Figure 6A) to accurately represent these experimental conditions and revised the figure legends to clearly describe them, ensuring a better understanding of each figure's context.

2) About the experiments in cells:

a. The diagram in Figure 3A is not very clear and the side view is misleading for understanding what is happening in the experiment. It should be improved.

Response: We agree that the side view in the original diagram may have been misleading. To enhance clarity and accurately represent the experimental setup, we have revised the diagram accordingly (current Figure 3).

b. Sup Fig3A upper panel (DMSO): on the nanobars, the authors write "highlighted dense F-actin networks at the end of high-curvature regions above the nanobars". From the images, there is no sign of dense F-actin networks at the level of the nanobars. The authors should modify the sentence in the text.

Response: We appreciate the reviewer's observation regarding the description of dense F-actin networks in the upper panel of Supplementary Figure 3A. To clarify, the images presented in Figure 3B are averaged composites of several individual frames as exemplified in Supplementary Figure 3A, highlighting actin clusters at the ends of high-curvature regions above the nanobars. This approach was necessary due to the dynamic on-off behavior of the actin signal, as demonstrated in previous studies (please see example in Lou et al. (PNAS 2019) Movie S1 and Zhao et al. (Nature Nanotechnology 2017) Movie S2, see Revision Figure R7). To ensure clarity and accuracy, we have revised the relevant text in the manuscript to better describe the original single time-frame images in Supplementary Figure 3A for supporting Figure 3B, which showed differential density in F-actin clusters following curvature radius.

Revision Figure R7

Figure R7. Montage of Movie S2 of dynamin2-GFP on nanobars shows repeated appearance and disappearance of dynamin2 spots at nanobar locations (Zhao et al, 2017)

c. It is not easy to understand what the authors mean with this experiment on cells. It does not provide any new information compared with what has been shown in previous studies (as in Lou et al. PNAS 2019 for example). The authors should either delete it or clearly specify the contribution of this experiment to the message of the paper.

Response: We appreciate the reviewer's constructive comments, which have helped us clarify the significance of our findings and distinguish them from those of Lou et al. (PNAS 2019). While Lou et al. demonstrated actin polymerization at the ends of nanobars and its colocalization with FBP17, their correlation-based discovery did not elucidate the exciting molecular mechanisms underlying nested hierarchical assembly of FBP17-N-WASP complex for local actin polymerization on curvature. Our newly incorporated Cdc42 studies provide a detailed mechanistic analysis of multicomponent molecular interactions and the activation of actin nucleation on the membrane surface. According to the *in-cellulo* assay in Figure R1 (current Figure 3E-G), we demonstrate while the partial inhibition of small GTPase Cdc42 reduced actin polymerization without affecting N-WASP curvature localization, stronger inhibition impaired both. This indicates that Cdc42 regulates curvature-mediated actin assembly through two potential mechanisms: directly activating N-WASP for basal actin polymerization and facilitating its association with FBP17 at curved membranes. These findings provide mechanistic insights that were not addressed in previous studies and demonstrate how global Cdc42 regulation integrates with local curvature-driven actin remodeling. In the revised manuscript, we have integrated these points to highlight the novelty and distinctiveness of our work compared to previous studies.

Revision Figure R1 (current Figure 3 E-G)

Figure R1. N-WASP signal in U2OS cells after the inhibition of Cdc42

- A. Original and averaged confocal images of U2OS fixed cells cultured on 300nm-wide nanobars with the expression of mScarlet3-N-WASP and the staining with Phalloidin-ATTO-488. Cells were fixed after a 1h treatment with DMSO, 20 μM or 50 μM ML141. Scale bar, 10 μm (left) and 1 μm (right).
- B. Actin signal end-to-center ratio at nanobars 300nm in width. Sample sizes for treatments under

DMSO, 20 μ M or 50 μ M ML141 were N=634, 324, 689, respectively. Each data point represents the mean \pm SD.

- C. mScarlet-N-WASP signal end-to-center ratio at nanobars 300nm in width. Sample sizes for treatments under DMSO, 20 μ M or 50 μ M ML141 were N=634, 324, 689, respectively. Each data point represents the mean \pm SD.

Statistical analysis was performed using one-way ANOVA followed by Tukey's multiple comparisons test ($P < 0.05$).

3) To ensure that the polymerization observed in Figure 3 is branched actin polymerization and to complement the observations made in Supplemental Figure 3F, the authors should perform a control experiment without the Arp2/3 complex to confirm that the observed localization is not due to linear filament formation dependent on curvature. It would also be interesting to conduct the experiment without N-WASP to see if actin localization on the lipid bilayer is curvature-dependent. These two controls will strengthen the message of Figure 3, demonstrating that branched actin assembly can be modulated by membrane curvature.

Response: We appreciate the reviewer's suggestion and have conducted the recommended control experiments to elucidate the roles of N-WASP and the Arp2/3 complex in actin polymerization on nanobar patterns. Using supported lipid bilayers (SLBs) with nanobar patterns, we performed actin polymerization assays under the following conditions: 1) **Complete System (Positive Control, F+N+A+C)**: SLBs incubated with FBP17, N-WASP, Cdc42, Arp2/3 complex, CapZ, and G-actin (current Figure 4E) ; 2) **Without N-WASP(F+A+C)**: SLBs incubated with all components except N-WASP (current Figure 4E); 3) **Without Arp2/3 Complex(F+N+C)**: SLBs incubated with all components except the Arp2/3 complex (current Figure 4E). Our results indicate that the absence of either N-WASP or the Arp2/3 complex abolishes actin polymerization on nanobars, confirming that the observed polymerization in the complete system is branched actin assembly rather than spontaneous or linear filament formation. Conversely, in the absence of FBP17(N+A+C), actin polymerization occurs but lacks curvature sensitivity.

These findings demonstrate that both N-WASP and the Arp2/3 complex are essential for actin polymerization in this context, while FBP17 is crucial for preferential actin assembly to curved membrane sites. We have incorporated these new results into the revised manuscript.

Figure R8 (current Figure 4E-G)

Figure R8. Additional actin assembly assay with combinations of FBP17, N-WASP, Arp2/3, and Cdc42.

E) Averaged confocal images of *in vitro* reconstitution of 3 μM actin (10% Oregon labeled) polymerization at 15min on the bilayer at nanobar of 200-1000 nm width with the presence of 200 nM FBP17, 50 nM N-WASP, 12 nM CapZ and 5 nM Arp2/3. (Red: SLB. Green: actin.) Scale bar, 1 μm.

F) Normalized actin signal intensity based on their corresponding lipid bilayer intensity at nanobars of 200-1000 nm width. Sample sizes for nanobars and flat regions were N=9, 9, 9, 8 and 8, respectively. Each point represents mean ± SEM.

G) The linear fitted absolute slope of actin signal to nanobar width ranging from 200-1000 nm. N=9, 9, 9, 8 and 8, respectively. Each point represents mean ± SD.

Minor points:

1. not independent in terms of polymerization. However, from the movie, it is clear that each bar polymerizes its own actin network. The authors should redo the scheme to make this clearer.

Thanks for the great suggestion. We have redone the scheme in the current Figure 4A to clarify it better.

2. Figure 3G: why is there a drop in actin intensity at 500 seconds?

The drop in actin intensity at 500 seconds was due to a temporary loss of autofocus caused by sample drift during imaging. However, we manually corrected the focus afterward, ensuring that subsequent frames accurately reflect the actin polymerization process. Importantly, this transient dip does not affect our quantitative analysis, as we didn't include the affected frames in the slope calculation window (105-350s).

3. In the legends, it is not easy to understand if the big N stands for independent replicates,

field of view or nanobars observed. This information should be specified.

We have now defined each N clearly in the legends.

4. Sup Figure 3B: there is a typo in "Alexa Fluor".

We have corrected it now in current Supplementary Figure S4A.

5. The non-polymerizable G-actin does not appear in the method section. The source of this protein should be included.

We have included it now in the methods.

6. Fig 4G: is the actin polymerization done on a SLB? This should be more explicit in the figure legend and in the text.

The assay is done on a PEG-Biotin-coated glass slide. We now add a scheme in current Figure 5G to describe the experiment.

7. Figure 4I: colors should be explained in the legend or on the figure.

We have now added detailed captions in Figure 5J.

8. Figure 4I: Most of the FBP17 clusters do not have actin asters. How do the authors explain this phenomenon? This needs to be addressed for a better understanding of the results. Line 316: the authors refer to Sup Fig 3F to see the FBP17 at the center of actin asters but this is not clear at all from the image shown. More examples should be shown for this statement to be convincing.

Response: We appreciate the reviewer's comments. The differences in actin polymerization between Figure 4I and Supplementary Figure 3F arise from distinct experimental setups. Supplementary Figure 3F was performed on a supported lipid bilayer (SLB) with slightly higher protein concentrations to capture the later stages of actin network formation, where FBP17 associates with polymerizing actin and generates new nucleation sites (visible as actin asters). In contrast, Figure 4I was conducted on a glass substrate with lower protein concentrations to focus on the earliest stages of nucleation, rather than the more extensive F-actin networks or FBP17-F-actin interactions observed at higher densities. Furthermore, the SLB likely provides a more dynamic environment that promotes protein-F-actin associations. Finally, our new data (Figure 6) using the SLB platform and various protein combinations further confirm that FBP17 reliably localizes at actin aster centers.

Revision Figure R9 (Dual color images for the last panel of current Figure 6E)

Figure R9. FBP17 highly aligned with the actin aster centers.

TIRF microscopy images showing the colocalization of actin filaments and FBP17. (Left) Actin network labeled in green. (Middle) FBP17 puncta labeled in magenta. (Right) Merged image demonstrating the spatial overlap between actin filaments and FBP17. Scale bar: 20 μm .

9. In the methods "TIRF actin assembly assay *in vitro*", the authors write that the surface is conjugated with streptavidin. Could the authors explain why they do this step?

This is a technical requirement for TIRF-actin assay. Since TIRF microscopy provides high sensitivity for signals on the surface within 100 nm, we need to anchor F-actin in this space. To achieve this, we coated coverslips with Biotin-PEG and allowed Streptavidin to conjugate with Biotin-PEG. After washing away the excess Streptavidin with actin polymerization buffer, we added a G-actin mix, containing 1% Biotin-labeled G-actin, to initiate polymerization. These steps allowed F-actin to be anchored within a proper Z-range for TIRFM.

Referee #3:

Zhu et al. propose an interesting manuscript aiming to decipher the mechanism by which plasma membrane curvature induced by substrate nanotopography triggers actin nucleation via the condensation of the TOCA protein FBP17, and N-WASP. Through an elegant set of *in vitro* experiments and modeling, they show that FBP17, which has a strong affinity for highly curved membranes, causes the condensation of N-WASP at sites of high membrane curvature. Interestingly, they provide data indicating that the condensation of FBP17 and N-WASP occurs optimally within a specific range of concentrations and stoichiometry. Their study also outlines the temporal sequence of events: 1{degree sign}) membrane curvature is sensed by FBP17, 2{degree sign}) FBP17 induces N-WASP condensation at high curvature sites, and 3{degree sign}) N-WASP condensates subsequently recruit Arp2/3 complex, which induces actin polymerization. While experiments seem adequately performed and the figures and legends are generally clear and complete, important controls are lacking (see major concerns), and the statistical information could be improved (see minor concerns).

In addition, although this study is qualitative, its major limitation lies in the lack of *in cellulo* experiments to support the *in vitro* datasets. This significantly limits the scope of the study, as it leaves room for the argument that their findings may not apply to a biological/physiological context. Furthermore, the authors heavily emphasize the CDC42-independent nature of their described mechanism in the discussion. However, they do not provide any *in vitro* or *in cellulo* data addressing CDC42's involvement. Despite referencing the existence of previously described Rho GTPase-independent actin nucleation mechanisms in the discussion, the question remains unresolved for their mechanism, particularly in a physiological cellular context.

In conclusion, the findings of this study present limited novelty in their current form. A substantial amount of revision work would be necessary, including *in cellulo* data and a deeper exploration of the role of CDC42 in the mechanism, to make this story novel and of high interest and attractivity for EMBO Journal readers.

Please find our major and minor concerns below.

We sincerely appreciate the reviewer's insightful comments. The thoughtful suggestions regarding Cdc42 have helped us clarify our key points and incorporate valuable new results, thereby strengthening the physiological relevance and conclusions of our study.

Major concerns:

1{degree sign}) Role of CDC42 is not explored: Previous *in cellulo* studies have indicated that CDC42 is crucial for the recruitment of TOCA proteins to plasma membrane nanodeformations, such as CIP4 in Ledoux et al. 2023

(<https://doi.org/10.1126/sciadv.ade1660>). In that study, CIP4 recruitment to nanodeformations strongly depends on CDC42 expression and activity. They also show that the HR1 domain of CIP4 - which interacts with GTP-CDC42 (active form) - is essential for the specific recruitment of CIP4 to highly curved plasma membrane regions. A CIP4 mutant lacking HR1 domain is still recruited to plasma membrane, but less specifically to nanodeformations. In the current manuscript, Zhu et al. did not consider CDC42 in their analyses.

It is necessary and highly interesting to add GTP-loaded CDC42 in *in vitro* experiments presented in Fig.1 to 4 to see if this increases further the affinity of FBP17 to highly curved membranes, and subsequently the condensation of N-WASP and actin polymerization. This would also provide effective evidence on the putatively CDC42-independent nature of their mechanism, which should be strongly toned down if no additional data are provided.

Response: We appreciate the reviewer's insightful comments regarding the established role of Cdc42 in activating nucleation-promoting factors (NPFs) of the Arp2/3 complex and its relation to CIP4 for curvature-mediated NPF activation. To carefully evaluate the role of Cdc42, we have conducted a series of new *in vitro* and *in-cellulo* assays to dissect the interplay between these two activation mechanisms.

First, our findings show that Cdc42 activates actin polymerization through NPFs, such as N-WASP, as previously documented. In addition, our study further reveals that curvature-induced activation of N-WASP, mediated by the curvature-sensing protein FBP17, operates alongside Cdc42-mediated pathways (Figure R10/current Figure 6, Figure R11/current Supplementary Figure 6, Figure R8/current Figure 4E-G). This dual activation results in a significant increase in curvature-guided local actin polymerization. Systematic reconstitution assays on supported lipid bilayers with and without curvature allowed us to decouple these mechanisms, demonstrating that their coexistence amplifies actin nucleation activity locally on curvature areas.

Our new results suggest that Cdc42 provides a basal level of activation, while membrane curvature determines a hotspot localized enhancement of actin polymerization. This spatial coordination may facilitate dynamic cellular processes, such as wave propagation, that require the combined functions of Cdc42 and FBP17. Here, we have incorporated these new insights into the revised manuscript to highlight the dynamic interplay between Cdc42-mediated and curvature-mediated actin assembly. The new results are now added as Figures 3E-G, 4E-G, 6A-D, Supplementary Figure S4D, S6A-D with more elaboration in the revised discussion. Here, with the following figures, we provide a detailed explanation of the newly performed experiments, the resulting data, and the enhanced conclusions.

Specifically, Figure R10 (new Figure 6) illustrates the cooperative roles of Cdc42, FBP17, and N-WASP in promoting Arp2/3 complex-mediated actin polymerization on SLBs. Time-lapse imaging (Panel B) and quantitative fluorescence intensity measurements (Panel C) reveal the individual and combined contributions of these factors. In the +F+N+A condition, FBP17

significantly enhances actin polymerization, resulting in fluorescence intensity comparable to the +N+A+C condition, where Cdc42 directly activates N-WASP. Notably, the +F+N+A+C condition, with both FBP17 and Cdc42 present, exhibits the most robust actin polymerization, reflected in the highest fluorescence intensity and densest actin filament network. These results indicate that Cdc42 and FBP17 synergistically enhance N-WASP-mediated actin nucleation.

Figure R10 (current Figure 6)

Figure R10. Cdc42 and FBP17 cooperatively regulate actin polymerization on flat SLBs.

B. Timelapse TIRF images of *in vitro* reconstitution of 3 μM actin (10% Oregon labeled) polymerization on the bilayer with the combination of 10nM Cdc42, 20 nM FBP17, 5 nM N-WASP, and 5 nM Arp2/3. Scale bar, 10 μm .

C. Plot of mean actin signal intensity in (B). N=4. Each point represents mean \pm SEM.

D. Actin polymerization rate in (C). (The linear fitted slope within 105-350s range in G). N=4. Each point represents mean \pm SEM.

E. TIRF images of *in vitro* reconstitution of 3 μM actin (10% Oregon labeled) polymerization on the bilayer after 30min with the combination of 10nM Cdc42, 20 nM FBP17, 5 nM N-WASP, and 5 nM Arp2/3. Scale bar, 50 μm .

Next, our findings in Figure R10 further elucidate the cooperative roles of FBP17 (F) and Cdc42 (C) in activating N-WASP-mediated actin polymerization in the presence of CapZ, a barbed-end capping protein that prevents filament elongation. Although both “+C” and “+F+N” conditions individually promote actin polymerization, their combination yields a significantly stronger synergistic effect.

Revision Figure R11 (current Supplementary Figure 6)

Figure R11. Cdc42 and FBP17 cooperatively counteract the inhibition of CapZ protein.

- Timelapse TIRF images of *in vitro* reconstitution of 6nM CapZ and 3 μM actin (10% Oregon labeled) polymerization on the bilayer with the combination of 10nM Cdc42, 20 nM FBP17, 5 nM N-WASP, and 5 nM Arp2/3. Scale bar, 10 μm .
- TIRF images of *in vitro* reconstitution of 3 μM actin (10% Oregon labeled) polymerization on the bilayer after 30min with the combination of 10nM Cdc42, 20 nM FBP17, 5 nM N-WASP, and 5 nM Arp2/3. Scale bar, 50 μm .
- Plot of mean actin signal intensity in (B). N=4. Each point represents mean \pm SEM.
- Actin polymerization rate in (C). (The linear fitted slope within 105-350s range in G). N=4. Each point represents mean \pm SEM.

In addition, Figure R7 also examines the roles of Cdc42, FBP17, N-WASP, and the Arp2/3 complex in actin polymerization on curved membranes. Control experiments omitting N-WASP or the Arp2/3 complex showed a complete loss of both general actin assembly and curvature sensitivity. While Cdc42 and N-WASP can induce general actin polymerization, curvature-specific assembly requires FBP17. FBP17 detects and binds to curved membranes, recruiting and clustering N-WASP at these sites, which leads to localized actin polymerization. The presence of all components—Cdc42, FBP17, N-WASP, and the Arp2/3 complex—results in the most pronounced actin assembly, indicating that FBP17 provides spatial specificity in boosting actin assembly on curvature in the presence of Cdc42-based basal function.

Revision Figure R8 (Figure 4 E-G)

Figure R8 Cdc42 and FBP17 cooperated to regulate curvature-guided actin polymerization

E) Averaged confocal images of *in vitro* reconstitution of 3 μ M actin (10% Oregon labeled) polymerization at 15min on the bilayer at nanobar of 200-1000 nm width with the presence of 200 nM FBP17, 50 nM N-WASP, 12 nM CapZ and 5 nM Arp2/3. (Red: SLB. Green: actin.) Scale bar, 1 μ m.

F) Normalized actin signal intensity based on their corresponding lipid bilayer intensity at nanobars of 200-1000 nm width. Sample sizes for nanobars and flat regions were N=9, 9, 9, 8 and 8, respectively. Each point represents mean \pm SEM.

G) The linear fitted absolute slope of actin signal to nanobar width ranging from 200-1000 nm. N=9, 9, 9, 8 and 8, respectively. Each point represents mean \pm SD.

In datasets such as in Fig.1C or D, the normalized end density of FBP17 is very close to 1, even on the thinnest nanobars (200 nm) where the value is around \sim 1.2. This indicates that the

enrichment of the protein to high curvature regions of the membrane is not very high. One may wonder if a normalized end density of 1.2 is statistically significant (given the error intervals), which the authors should show.

Response: We have now added statistical data to Figure 1E (formerly Figure 1C or D), which demonstrates a curvature-dependent difference in the recruitment of FBP17 and N-WASP, in which N-WASP, in particular, exhibits greater sensitivity to curvature.

2{degree sign}) Lack of *in cellulo* data to support the conclusions of the study: The only experimental data in a cellular model are provided in Fig. 3A-D and Fig. S3A. Unfortunately, experiments with Latrunculin A are not particularly informative. Additional conditions need to be tested to support the *in vitro* data. For instance, the authors should look at actin end density on nanobars upon FBP17 (and possibly also CIP4 and FBNP1L) and N-WASP depletion using siRNAs. This cellular system could also be used to further explore if their mechanism is indeed CDC42-independent, and compare the responses obtained to their *in vitro* data. Without any further experimental evidence *in vitro* (see major concern #1) and *in cellulo*, it cannot be claimed that the mechanism described here is strictly CDC42-independent.

Response: We appreciate the reviewer's insightful comments. Our original intention was to highlight the new activation mechanism rather than suggesting a Cdc42-independent mechanism. As outlined in our previous responses, extensive additional experiments have confirmed the critical role of F-BAR proteins in orchestrating local curvature-mediated actin assembly during these processes. Ledoux et al. (2023) conducted a comprehensive study on the role of BAR family proteins in curvature sensing using nanobeads. Their findings demonstrated that depleting the TOCA family proteins—FBNP1L, FBP17, and CIP4—significantly reduced actin accumulation at curved sites on 100 nm nanobeads, but not on larger beads (300 and 500 nm). This underscores the importance of these proteins in curvature-induced actin assembly. In addition, earlier work also showed that the truncated FBP17 without the SH3 domain led to significantly reduced actin accumulation at the nanobar ends in cells, confirming the essential role of FBP17 for curvature-guided local actin assembly.

Therefore, our study here focuses on elucidating the stepwise regulation spatially with nested hierarchical assembly that regulates local actin nucleation activities, mainly how local membrane curvature, sensed by BAR-domain proteins like FBP17, enhances actin nucleation through multivalent interactions. Such horizontal comparison between BAR domain-containing proteins has already been nicely characterized by Ledoux et al. (2023). In addition, we have also explored the cooperation between local curvature-induced mechanisms and overall Cdc42-mediated actin nucleation. We respectfully acknowledge the value of recommended further experiments, such as those involving additional BAR proteins and siRNA-mediated depletion of N-WASP. But, these

investigations fall beyond the current scope of our study and would not alter our work's primary conclusions or novelty.

In our revised manuscript, we better clarify the unique mechanisms we have investigated and differentiate our findings from previous studies, emphasizing how our work complements existing knowledge and can inform future studies on BAR proteins in cellular contexts. We have included the cooperative role of Cdc42 and FBP17 in curvature-dependent actin assembly (Figures R1, Current Figure 3E-G) and the importance of the multivalent interaction between oligomeric SH3 and PolyP region (Figures R6, Current Figure 8A-F).

Firstly, we have demonstrated the interplays of curvature-induced actin assembly and GTPase Cdc42-mediated actin assembly. Our results confirm that Cdc42 is essential for N-WASP activation and also influences the local curvature-guided hierarchical assembly of N-WASP induced by FBP17. These insights have been incorporated into the revised manuscript, specifically in Figure 3E-G.

Revision Figure R1 (current figure 3E-G)

Figure R1: Effects of ML141 and CK666 on Actin Cytoskeleton and N-WASP Localization in U2OS Cells (Current Figure 3E-G).

A. Original and averaged confocal images of U2OS fixed cells cultured on 300nm-wide nanobars with the expression of mScarlet3-N-WASP and the staining with Phalloidin-ATTO-488. Cells were fixed after a 1h treatment with DMSO, 20µM or 50µM ML141. Scale bar, 10 µm (left) and 1 µm (right).

B. Actin signal end-to-center ratio at nanobars 300nm in width. Sample sizes for treatments under DMSO, 20uM or 50uM ML141 were N=634, 324, 689 nanobar ends, respectively. Each data point represents the mean \pm SD.

C. mScarlet-N-WASP signal end-to-center ratio at nanobars 300nm in width. Sample sizes for treatments under DMSO, 20uM or 50uM ML141 were N=634, 324, 689 nanobar ends, respectively. Each data point represents the mean \pm SD.

Statistical analysis was performed using one-way ANOVA followed by Tukey's multiple comparisons test ($P < 0.05$).

Secondly, to highlight our discovery in curvature-induced multivalent interaction, we expressed an ENTH-SH3^{FBP17} protein, which is monomeric but can sense membrane curvature, in U2OS cells cultured on the regular nanobar chip (300nm). By observing the end enrichment of N-WASP signals and the actin distribution, we can clearly tell the role played by multivalency in this process. (**Figure R6**) Our quantitative fluorescence analysis reveals that both FBP17 and ENTH-SH3 sense membrane curvature with similar efficiency, as indicated by comparable end-to-center enrichment ratios on nanobar structures (Figure R6B-C). However, their ability to recruit N-WASP differs markedly. While FBP17 induces strong N-WASP clustering at nanobar ends, ENTH-SH3 shows significantly weaker recruitment (Figure R6B-D). This difference is further reflected in actin polymerization patterns—FBP17 induces a curvature-enriched actin network at highly curved membrane regions, whereas ENTH-SH3 induces a lower level of curvature-sensitive actin organization, leading to a more diffused actin distribution (Figure R6B-E). These results establish that curvature sensing is insufficient for efficient N-WASP clustering and downstream actin polymerization; instead, multivalent interactions through oligomeric SH3 domains are indispensable for organizing N-WASP into actin assemble hubs.

Revision Figure R6 (current figure 8A-F)

Figure R6. ENTH-SH3FBP17 impairs N-WASP recruitment and activation at curved ends.

D. Schematic representation of curvature-sensing proteins FBP17 and ENTH-SH3 chimera on a curved

membrane. Both constructs sense curvature, but only FBP17, with its dimeric SH3 domains, enables multivalent interactions with N-WASP, promoting clustering and actin assembly. ENTH-SH3, a monomeric chimera, lacks this multivalency.

- E. Representative fluorescence images of U2OS cells expressing GFP-FBP17 (top) or GFP-ENTH-SH3 (bottom) along with mScarlet-N-WASP and Phalloidin-ATTO-647 labeled actin on nanobar structures. Scale bars, 10 μm .
- F. High-magnification images (top) and averaged images (bottom) of nanobar regions (boxed in B), showing FBP17 (green) or ENTH-SH3 (green), N-WASP (red), and actin (magenta) localization. Scale bars, 1 μm .
- G. Quantification of fluorescence signal ratios at nanobar ends relative to the center (etc ratio) for GFP (D, FBP17 vs. ENTH-SH3), mScarlet-N-WASP (E), and Cy5-actin (F).

In summary, our revised manuscript explicitly clarifies that Cdc42 contributes to actin assembly in our system, and we have incorporated new *in-cellulo* experiments to support our conclusions. Importantly, our study provides novel mechanistic insight into how multivalent interactions within the FBP17-N-WASP complex are essential for curvature-guided actin assembly.

3{degree sign}) Lack of controls: Based on data presented in Figures 3 and 4B, the authors propose that FBP17 drives N-WASP-mediated actin assembly at highly curved plasma membrane. This appears overstated, as controls are lacking (e.g. without FBP17).

- In Fig. 4B: How do the authors explain that actin polymerization is already increasing on large nanobars (> 500 nm), where FBP17 and N-WASP are barely recruited? Is there an FBP17- and N-WASP-independent actin polymerization mechanism *in vitro*? This needs to be clarified.

- Experiments in Fig. 4B and S3F need to be repeated without FBP17 as a control condition (as in Fig. 1G) to unambiguously demonstrate that actin polymerization does not occur (or occurs to a lesser extent) without the specific recruitment of FBP17/N-WASP machinery.

Response: We appreciate the reviewer's comments and have conducted additional control experiments to strengthen our conclusions.

In our reconstitution system, we observed increased actin polymerization on larger nanobars (>500 nm), despite low recruitment of FBP17 and N-WASP. This suggests that even at nanomolar concentrations, active N-WASP can initiate actin polymerization.

It's important to note that *in-cellulo* conditions involve additional regulators controlling overall actin production, such as negative regulators of the Arp2/3 complex and profilin for monomer binding. It is impractical to include all related regulatory factors in actin reconstitution assays, which are often designed based on the focused activity steps. Here, we focus on the fundamental aspects of actin nucleation. This reductionist approach allows for specific dissection of the actin nucleation process, a common strategy widely used in actin biochemistry studies.

To avoid potential overstatements, we have revised the manuscript to clearly discuss these points (on Page 9). Additionally, we performed experiments assessing actin polymerization on patterned SLBs in the absence of FBP17 (Figure R12, extracted from current Figure 4E, F). The

results indicate that though Cdc42 can elevate basal actin polymerization, while curvature-responsive actin assembly requires FBP17. This underscores FBP17's role in sensing membrane curvature and recruiting N-WASP locally to facilitate topographic information-guided actin polymerization.

Revision Figure R12 (extracted from current Figure 4E, F)

Figure R12. FBP17 enhances curvature-dependent actin recruitment at nanobar structures.

(A) Averaged fluorescence images of SLB with nanobar structures of varying widths (1000–200 nm). The top row shows lipid fluorescence (red); the middle row (N+A+C) shows actin recruitment in the presence of N-WASP (N), Arp2/3 complex (A), and Cdc42 (C); the bottom row (F+N+A+C) shows actin recruitment when FBP17 (F) is included in the system. White outlines indicate nanobar regions. Scale bar, 1 μ m.

(B) Quantification of lipid-normalized actin fluorescence signal at nanobar ends as a function of nanobar width.

4{degree sign} N-WASP mutants (Fig. S5C): The authors should show if N-WASP mutants can still condensate in the presence of FBP17. They should also explore if these N-WASP mutants are still able to mediate Arp2/3 recruitment and actin polymerization. This should be explored *in vitro* and in cellular models. In the current version of the manuscript, N-WASP mutants seem underexploited and do not appear particularly useful.

Response: Thanks for the question that allows us to clarify our intention and experiment design. We conducted these mutant experiments to demonstrate that previously characterized activated forms of N-WASP do not require clustering to nucleate actin, indicating that Cdc42-mediated and clustering-mediated activation proceeds via distinct, non-overlapping mechanisms. We have now performed an improved control experiment to show that valency is key in activating N-WASP by comparing the activity of monomeric ENTH-SH3^{FBP17} with that of dimeric FBP17 (currently Figure 8). In addition, new assays conducted in the presence of Cdc42 clarified how these two distinct activation pathways are coordinated. In summary, we agree with the reviewer that the N-WASP mutants are not particularly useful anymore and have replaced them with evidence-based on ENTH-SH3^{FBP17} and Cdc42 data.

Minor concerns:

1{degree sign}) The titles of some figures (e.g. Fig. 1, 3, S5...) and subheadings (e.g. line 222) are not informative. They should summarize the main findings of the figure or section.

We have now changed some subheadings and figure titles to better summarize our findings.

2{degree sign}) In the abstract, the second sentence (lines 23-25) seems to lack a few words: "Yet, the mechanism by which dynamic membrane curvature prompts quick actin cytoskeletal changes in signaling remains elusive".

We have modified the words in the revised manuscript.

3{degree sign}) In the introduction and wherever necessary (e.g. discussion, line 389), TOCA-1 should be presented as FNBP1L, TOCA-2 as FBP17, and TOCA-3 as CIP4. Either one uses TOCA nomenclature, or the alternative names (FNBP1L /FBP17/CIP4), to avoid confusion.

We have modified the terms in the revised manuscript.

4{degree sign}) Lines 72-75: Wrong citation of Ledoux et al., 2023. This study did not address the recruitment of WASP family proteins to highly curved membranes, but focused on TOCA proteins, CDC42 and actin polymerization.

We have removed the wrong reference in revised manuscript.

5{degree sign}) Legend of Fig.1A: On the scheme, blue blocks should be captioned as proline-rich domains.

We have corrected the typo.

6{degree sign}) Line 115: Alexa Fluor (and not Flour).

We have corrected the typo.

7{degree sign}) Figure 2K, vertical axis title: "Curvature-gated" (and not "Curvatur-gated").

We have corrected the typo.

8{degree sign}) Line 191: We further explored (...).

We have amended the sentences in the revised manuscript.

9{degree sign}) Line 217: "FBP arrays" should be replaced by "FBP17 arrays".

We have corrected the typo in the revised manuscript.

10{degree sign}) Figure 2J: To make things clearer to the reader, it should be written that the

concentrations indicated on the top of the chart are N-WASP concentrations.
We have added the label of (N-WASP conc.) on Figure 2L in the revised manuscript.

11{degree sign}) Lines 262-263: The authors claim that actin cytoskeleton is "highly aligned with curvature end of nanobars" in reference to Fig. S3F. What they call alignment is not clear. The statement brings confusion, as the referenced figure does not show alignment of actin fibers with nanobars. This should thus be rephrased for clarity.

We deleted the statements and now use new data and quantification to support our findings in Figure 4E-G, where the contribution of each component was dissected, and the radius-dependent actin polymerization was also demonstrated.

12{degree sign}) Fig. 4B: Data for FBP17 and N-WASP look the same as in Fig. 1D. Duplicated data must be avoided: either one of the datasets needs to be removed, or it should be specified somewhere on the figure or in the legend that a part of data are reused in Fig. 4B from Fig. 1D. As the reviewer suggested, we have now added "The FBP17, N-WASP and Actin curves are integrated data from Figure 1E and Figure 4D here to elaborate the parameterization of the model in (A)." to the caption of Figure 5B.

13{degree sign}) Lines 281-308: This paragraph describes a modeling procedure to predict how N-WASP localization can activate Arp2/3 complex activation. This paragraph should be better introduced and explained, as it is quite difficult to grasp for non-specialists. Also, it is not clear what this modeling part brings to the whole story. This should be either clarified or removed from the manuscript.

We thank the reviewer for the suggestion. We have rewritten this section using simpler language and have added additional details to ensure accessibility for readers of all levels of expertise. The changes made to the manuscript are shown below.

"The simulations play a pivotal role in explaining the mechanistic origin of enhanced nucleation from N-WASP clustering. Experimental studies cannot tunably control the extent of NWASP localization on SLBs. To overcome this limit, our model generates concentration profiles of NWASP at various levels of localization. We use this to study Arp2/3 activation. Further, the simulations show that an adequate activation rate of Arp2/3 is necessary to see the localization-driven enhancement of Arp2/3 nucleation (Supplementary Figure 5). We quantify enhanced nucleation using both the number of filaments and the number of branches per μm^2 ."

14{degree sign}) Lines 327-328: "To investigate... actin polymerization, we (...)". It seems words are lacking in the first part of the sentence. It needs to be rephrased.

We have rephrased the sentences in the revised manuscript on Page 12.

"To investigate the cascade reaction in nanoclustering FBP17 and N-WASP complex for actin polymerization, we next examined whether homo-dimeric FBP17 can cluster full-length N-WASP

through the direct interaction between its SH3 domains and the disordered PolyP region (271-391aa) of N-WASP (Fig. 1A).”

15{degree sign}) Line 336: Shouldn't it be "stoichiometrically" rather than "stoichiometric"?
We have amended the wording as suggested on the Page 12.

16{degree sign}) In several figures, such as in Fig.5H and I, statistical details are lacking. Fig.5H: Only one actin aster is shown (SH3 trimer condition). How representative is it? Is it a frequent or rare event? What are the numbers of asters observed independently in each experimental condition (SH3 mono vs SH3 trimer)? Fig. 5I: Statistical significance should be depicted.

First, we employed nanomolar concentrations to capture the early stages of nucleation while minimizing potential overcrowding. Second, we performed the assay on a flat surface without curvature-induced local accumulation, resulting in events observed at low density. Here, we included a larger-field TIRF time-lapse in R14 (current Supplementary Figure S7F), providing a broader view of these events and typically featuring one to two actin asters per field. In addition, Figure R10 further highlights the strong alignment of FBP17–WASP-driven actin polymerization at the highly polymerized late stage. Finally, we have added the corresponding p-value to Figure 5I to indicate statistical significance and specified the sample size as “N=12 ROIs.”

Revision Figure R14 (current Figure S7F)

Figure R14. Larger field TIRF timelapse for monomer/trimer/tetramer SH3 in actin assembly assay.

(F) TIRF images of actin polymerization (10% Oregon-labeled) with different combinations of 80 nM FBP17/oTri-SH3/oTet-SH3, 20 nM N-WASP FL and 5 nM Arp2/3. Scale bar, 10 μ m.

17{degree sign}) Line 380: "sculpts" rather than "sculpt".

Thanks, we corrected it now.

18{degree sign}) Methods, line 503: "The primary amine groups (...) were labeled...".

Thanks, we corrected it now.

19{degree sign}) Methods: Values of labeling efficiencies of all proteins used in this study must be provided.

The labeling efficiencies vary from protein to protein and batch to batch (FBP17 is ~50-60%, and N-WASP is ~30-40%). However, our indicated labeled fractions in figure legend were counted in the efficiency. We have clarified our calculation in the revised Methods on Pages 20-21.

20{degree sign}) Methods, line 635: Remove the point after "10% FBS". PS is an acronym that is not explained before.

Thanks, we have corrected both.

21{degree sign}) Methods, line 665: "... were mixed...".

Thanks, we have corrected it now.

22{degree sign}) In several figure legends (e.g. Fig.2L,M, Fig.5D,G...), the authors write that they quantified a certain number of particles ($n = 200$) out of over 10,000 particles. Why did they choose 200 particles, not more or not less? Why didn't they analyze all the particles identified? How did they choose the particles? In other figure panels (e.g. Fig. 5I), what does N represent? In Fig. 5I, is $N=12$ the number of images or the number of actin asters?

Given the varying sample sizes in our datasets, we consistently selected 200 random particles to ensure a sufficiently large sample for statistical analysis and fair comparisons across experimental conditions. To avoid bias, we used Excel's random sampling function, which gives each particle an equal chance of selection and maintains the objectivity of our analysis. In Figure 5I (now Figure 7I), $N=12$ refers to the number of square ROIs. We have added these clarifications to the Methods section (Page 26).

23{degree sign}) In addition, in each legend, it should be mentioned how many independent experiments were performed for each dataset. For each image, it should also be indicated the number of independent experiments they are representative of.

We have now indicated in the methods that we performed 2 independent experiments otherwise indicated in the figure legends, and we also have the individual original images to be uploaded in the source file.

24{degree sign}) Statistical information regarding tests that were used and p-values should be reported in the legend of each figure. It should also be mentioned if data normality was checked for each dataset and how.

We have now included the statistical information regarding the tests used and the corresponding p-values in the legend of each figure. Additionally, we have specified whether data normality was

assessed for each dataset and the method used for this evaluation.

Reference

- Domanska M, Setny P (2024) Exploring the Properties of Curved Lipid Membranes: Comparative Analysis of Atomistic and Coarse-Grained Force Fields. *J Phys Chem B* 128: 7160-7171
- Frost A, Perera R, Roux A, Spasov K, Destaing O, Egelman EH, De Camilli P, Unger VM (2008a) Structural Basis of Membrane Invagination by F-BAR Domains. *Cell* 132: 807-817
- Frost A, Perera R, Roux A, Spasov K, Destaing O, Egelman EH, De Camilli P, Unger VM (2008b) Structural basis of membrane invagination by F-BAR domains. *Cell* 132: 807-817
- Han X, Hu Z, Surya W, Ma Q, Zhou F, Nordenskiöld L, Torres J, Lu L, Miao Y (2023) The intrinsically disordered region of coronins fine-tunes oligomerization and actin polymerization. *Cell Reports* 42
- Harmon TS, Holehouse AS, Rosen MK, Pappu RV (2017) Intrinsically disordered linkers determine the interplay between phase separation and gelation in multivalent proteins. *eLife* 6: e30294
- Kim AS, Kakalis LT, Abdul-Manan N, Liu GA, Rosen MK (2000) Autoinhibition and activation mechanisms of the Wiskott–Aldrich syndrome protein. *Nature* 404: 151-158
- Lee DSW, Choi C-H, Sanders DW, Beckers L, Riback JA, Brangwynne CP, Wingreen NS (2023) Size distributions of intracellular condensates reflect competition between coalescence and nucleation. *Nature Physics* 19: 586-596
- Lou H-Y, Zhao W, Li X, Duan L, Powers A, Akamatsu M, Santoro F, McGuire AF, Cui Y, Drubin DG *et al* (2019) Membrane curvature underlies actin reorganization in response to nanoscale surface topography. *Proceedings of the National Academy of Sciences* 116: 23143-23151
- Mcdonald A, Nathan, Kooi V, W., Craig, Ohi D, Melanie, Gould L, Kathleen (2015) Oligomerization but Not Membrane Bending Underlies the Function of Certain F-BAR Proteins in Cell Motility and Cytokinesis. *Developmental Cell* 35: 725-736
- Miao Y, Guo X, Zhu K, Zhao W (2023) Biomolecular condensates tunes immune signaling at the Host-Pathogen interface. *Curr Opin Plant Biol* 74: 102374
- Roberts-Galbraith RH, Gould KL (2010) Setting the F-BAR: Functions and regulation of the F-BAR protein family. *Cell Cycle* 9: 4091-4097
- Rohatgi R, Ho H-YH, Kirschner MW (2000) Mechanism of N-Wasp Activation by Cdc42 and Phosphatidylinositol 4,5-Bisphosphate. *Journal of Cell Biology* 150: 1299-1310
- SUETSUGU S, TEZUKA T, MORIMURA T, HATTORI M, MIKOSHIBA K, YAMAMOTO T, TAKENAWA T (2004) Regulation of actin cytoskeleton by mDab1 through N-WASP and ubiquitination of mDab1. *Biochemical Journal* 384: 1-8
- Yarar D, D'Alessio JA, Jeng RL, Welch MD (2002) Motility determinants in WASP family proteins. *Mol Biol Cell* 13: 4045-4059
- Yesylevskyy SO, Rivel T, Ramseyer C (2017) The influence of curvature on the properties of the plasma membrane. Insights from atomistic molecular dynamics simulations. *Scientific Reports* 7: 16078
- Zhao W, Hanson L, Lou H-Y, Akamatsu M, Chowdary PD, Santoro F, Marks JR, Grassart A, Drubin DG, Cui Y *et al* (2017) Nanoscale manipulation of membrane curvature for probing endocytosis in live cells. *Nature Nanotechnology* 12: 750-756

Dear Yansong,

Thank you for submitting your revised manuscript for consideration by The EMBO Journal. I have now received comments from two of the original reviewers, which are included below for your information.

Unfortunately, both reviewers find that some of their initially raised points were not addressed satisfactorily, with the concerns of reviewer #3 being more extensive, especially regarding the broader advance of the findings and insufficient integration of the existing knowledge in the interpretation of the findings, as also raised by reviewer #1 regarding the modelling approach. Furthermore, reviewer #3 finds the provided in cellulose validation insufficient.

Due to our "single major revision round" policy, and since we require strong enthusiasm from all reviewers to proceed with the manuscript after an appeal, I am afraid that I cannot offer further steps towards publication here.

That being said, in the interest of a rapid publication of the study I have discussed your manuscript and referee comments with my colleague Martina Rembold at our sister journal EMBO Reports. I am glad to say that she would be interested in considering your manuscript for publication at their journal revised as outlined below:

- Given the concerns raised by referee 1 regarding the validity of the theoretical section on curvature sorting, these data should be removed, as suggested.
- The interplay between Cdc42 and FBP17 should be strengthened. Referee 3 suggests a number of experiments, all relying on an FBP17 mutant lacking the HR-1/REM-1 domain (point 1, point 2, point 3) that appear feasible and would address this concern.
- It will not be necessary to provide further in cellulose experiments (point 4) and to test FBP17 point mutants that disrupt dimerization/oligomerization (point 3, 2nd paragraph).

Please feel free to contact Martina at m.rembold@emboreports.org if you would like to discuss the proposed revisions further. You do not need to revise the manuscript prior to transfer. Once you have initiated the transfer, Martina will send you an invitation to revise, outlining the scope of revision.

If you find the transfer option of interest, please use the link below to transfer the manuscript:

Link Not Available

Thank you in any case for the opportunity to consider this manuscript. I am sincerely sorry that I could not communicate more positive news, and I very much hope that you will find the transfer of interest.

With kind regards,

leva

leva Gailite, PhD
Senior Scientific Editor
The EMBO Journal
Meyerhofstrasse 1
D-69117 Heidelberg
Tel: +4962218891309
i.gailite@embojournal.org

Referee #1:

In their revised version, the authors present several new datasets highlighting the role and importance of Cdc42, as well as additional evidence supporting the significance of FBP17's multivalent interaction with N-WASP by using an alternative curvature sensor, ENTH, which exhibits only monovalent interactions. The manuscript is now significantly enriched and clarified compared to its initial version, reinforcing its overall message.

Regarding the model describing the curvature-induced sorting of FBP17 and N-WASP on nanobars, I appreciate the authors' efforts to clarify their approach. However, I still have serious concerns about this model. It is evident that the model is

phenomenological, with curvature dependence introduced artificially via an equation analogous to Bell's law for the binding rate of FBP17. The physical meaning of the force parameter "alpha" remains unclear. Furthermore, in the supplementary information, the authors still assume that the BAR domain interacts with hydrophobic patches of the membrane, whereas this interaction is primarily electrostatic. In reality, when the membrane bends, the distance between head groups increases, leading to a decrease in charge density, which should in turn reduce the protein's affinity for the membrane. Additionally, the reported values for the 1D diffusion of proteins (100 $\mu\text{m/s}$) seem unrealistically high, and there is no attempt to compare the kinetic parameters used in the simulations with experimental data. Overall, I do not see what the model contributes to the paper, particularly given that its assumptions are not well supported by existing knowledge on FBP17.

On a minor note, there are still a few typos or grammatical issues that should be addressed.

In conclusion, I believe the paper is now suitable for publication in EMBO Journal, but I strongly recommend removing the theoretical section on curvature sorting.

Referee #3:

In this revised version of their manuscript, Zhu et al. present novel data, primarily in vitro with some in cellulo. Notably, they provide a more thorough investigation of the role of CDC42 in their proposed mechanism, suggesting that curvature sensitivity originates from FBP17, while significant actin nucleation and polymerization require CDC42. Their findings demonstrate a clear synergistic role for the tripartite complex FBP17, CDC42, and N-WASP. While this version of the manuscript is substantially improved with the inclusion of new data, certain aspects still require further investigation (see comments below). Overall, while the amount of data and effort in this study is substantial, elegant, and potentially valuable to the community, the novelty of the findings is moderate in light of existing literature. This study primarily expands previously studied mechanisms by employing in vitro setups that enable precise control over a broad range of molecular players and membrane deformation sizes.

1. Does CDC42 interaction with FBP17 influence FBP17 recruitment to curved membranes and its curvature sensitivity properties? This question remains unresolved. Experiments similar to those in Figure 1E (measuring end densities of FBP17 and N-WASP) should be conducted in the presence of CDC42, as well as with an FBP17 mutant lacking the HR-1/REM-1 domain. Such experiments would clarify whether the interaction between these two proteins is necessary for FBP17's ability to sense membrane curvature. In their study, Ledoux et al. (2023) provided in cellulo data using CIP4 mutants (Fig. 4F-I), suggesting that CIP4's interaction with CDC42 is crucial for its optimal recruitment to plasma membrane nanodeformations. Zhu et al. have an excellent experimental setup that would allow them to test this hypothesis in vitro across a broader range of membrane curvatures.

2. In the rebuttal letter and the revised manuscript (particularly in the discussion), despite the new data highlighting the importance of CDC42 in inducing proper actin nucleation and polymerization in a curvature-dependent manner, the authors continue to treat CDC42 and FBP17 as largely independent factors. This is particularly evident in the schematics of the new Figure 6 and Supplementary Figure 6, where the well-documented interaction between CDC42 and FBP17 via the HR-1/REM-1 domain is not depicted at all. Given that FBP17, N-WASP, and CDC42 form a tripartite complex, this omission is significant.

> Does it make sense to propose two parallel activation mechanisms for actin nucleation and polymerization? The new Supplementary Figure 6 demonstrates that robust actin nucleation occurs only when the tripartite complex is formed, and is significantly reduced when any one of the partners is absent. Additionally, Figure 4F clearly indicates that the curvature-dependent actin polymerization is strongest when all molecular components are present (condition +F+N+A+C).

> Does disruption of the HR-1/REM-1 domain in FBP17 abolish the cooperative/synergistic effect of FBP17 and CDC42 on actin polymerization on both flat and curved membranes? As mentioned above, experiments using FBP17 mutants lacking the HR-1/REM-1 domain would be highly informative.

3. New data with ENTH-SH3 (Figure 8A-F): comparison between multivalent interactions via oligomeric SH3 (FBP17) and monovalent interactions via monomeric SH3 (ENTH-SH3). These new data appear somewhat like comparing apples and oranges. The observed differences between FBP17-SH3 and ENTH-SH3 may not be solely attributable to their oligomeric versus monomeric states. Another overlooked factor is that FBP17 can interact with CDC42, whereas ENTH cannot. Moreover, literature suggests that certain ENTH domains interact with RhoGAPs in yeast, which in turn catalyze CDC42 inactivation (<http://dx.doi.org/10.1073/pnas.0510513103>). This raises the possibility that ENTH might inhibit CDC42 and actin polymerization. Consequently, ENTH-SH3 does not seem to be an ideal control.

To obtain a clearer picture, an important additional control would be a mutated version of FBP17 lacking the HR-1/REM-1 domain, thereby preventing its interaction with CDC42. If actin assembly is impaired under these conditions, it would suggest that the formation of the FBP17/CDC42/N-WASP tripartite complex is required and may be more critical than the oligomerization question. Further, examining FBP17 point mutants that disrupt dimerization/oligomerization while preserving membrane-binding capacity could provide additional insight, though identifying such mutants might be challenging.

4. Lack of in cellulo data: While the authors have included some additional cellular data, they remain limited. More in cellulo validation of their in vitro findings would significantly strengthen the manuscript, although they considered these as out of scope of their manuscript in their previous answers (i.e. siRNA treatment against their molecular players).

5. Additional minor note: The ENTH-SH3 expressing plasmid is not described in Table S3. From which protein is the ENTH domain derived?

6. Minor comment on Figure 4F: Why is the Y-axis truncated and not continuous?

In summary, this revised version of the manuscript is significantly improved, but several key aspects, particularly regarding the interplay between CDC42 and FBP17, still warrant further investigation. Addressing these points would enhance the overall impact and mechanistic depth of the study. In addition, while the novelty of the findings is moderate, the substantial amount and high quality of the data make this work potentially valuable to the community.

** As a service to authors, EMBO Press provides authors with the possibility to transfer a manuscript that one journal cannot offer to publish to another EMBO publication or the open access journal Life Science Alliance launched in partnership between EMBO Press, Rockefeller University Press and Cold Spring Harbor Laboratory Press. The full manuscript and if applicable, reviewers' reports, are automatically sent to the receiving journal to allow for fast handling and a prompt decision on your manuscript. For more details of this service, and to transfer your manuscript please click on Link Not Available. **

Response to Reviewers – Resubmission

We sincerely thank the reviewers for their valuable feedback and suggestions, which allowed us to clarify the mechanisms section. Their insights guided us to undertake substantial additional work that deepened our understanding of the progressive, hierarchical assembly of Cdc42, FBP17, and N-WASP under curvature control. We mapped how Cdc42-driven actin polymerization works in conjunction with curvature-driven polymerization, highlighting both synergy and differences across a range of curvature radii (mimicking dynamic *in vivo* curvature changes).

Below, we provide a point-by-point response to each of the remaining concerns. In this version, we have further strengthened the manuscript with additional *in vitro* and *in cellulo* experiments and have removed the theoretical modeling section as recommended.

Referee #1

Comment: In their revised version, the authors present several new datasets highlighting the role and importance of Cdc42, as well as additional evidence supporting the significance of FBP17's multivalent interaction with N-WASP by using an alternative curvature sensor, ENTH, which exhibits only monovalent interactions. The manuscript is now significantly enriched and clarified compared to its initial version, reinforcing its overall message.

Regarding the model describing the curvature-induced sorting of FBP17 and N-WASP on nanobars, I appreciate the authors' efforts to clarify their approach. However, I still have serious concerns about this model. It is evident that the model is phenomenological, with curvature dependence introduced artificially via an equation analogous to Bell's law for the binding rate of FBP17. The physical meaning of the force parameter "alpha" remains unclear. Furthermore, in the supplementary information, the authors still assume that the BAR domain interacts with hydrophobic patches of the membrane, whereas this interaction is primarily electrostatic. In reality, when the membrane bends, the distance between head groups increases, leading to a decrease in charge density, which should in turn reduce the protein's affinity for the membrane. Additionally, the reported values for the 1D diffusion of proteins (100 $\mu\text{m/s}$) seem unrealistically high, and there is no attempt to compare the kinetic parameters used in the simulations with experimental data. Overall, I do not see what the model contributes to the paper, particularly given that its assumptions are not well supported by existing knowledge on FBP17.

On a minor note, there are still a few typos or grammatical issues that should be addressed. In conclusion, I believe the paper is now suitable for publication in EMBO Journal, but I strongly recommend removing the theoretical section on curvature sorting.

Response:

We appreciate the reviewer's critical evaluation of our theoretical model. As suggested, we have now removed the theoretical model. In addition, we have thoroughly corrected all typographical and grammatical errors in the revised manuscript.

Referee #3

In this revised version of their manuscript, Zhu et al. present novel data, primarily *in vitro* with some *in cellulo*. Notably, they provide a more thorough investigation of the role of Cdc42 in their proposed mechanism, suggesting that curvature sensitivity originates from FBP17, while significant actin nucleation and polymerization require Cdc42. Their findings demonstrate a clear synergistic role for the tripartite complex FBP17, Cdc42, and N-WASP. While this version of the manuscript is substantially improved with the inclusion of new data, certain aspects still require further investigation (see comments below). Overall, while the amount of data and effort in this study is substantial, elegant, and potentially valuable to the community, the novelty of the findings is moderate in light of existing literature. This study primarily expands previously studied mechanisms by employing *in vitro* setups that enable precise control over a broad range of molecular players and membrane deformation sizes.

Response: We express our gratitude to the reviewer for their insightful and thoughtful feedback on our revised manuscript. The reviewer recommended clarifying the role of Cdc42 within the curvature-FBP17-driven mechanism we proposed, highlighting that the tricomponent relationship involving FBP17, Cdc42, and N-WASP lacked clarity. In our initial revision, we incorporated Cdc42 and demonstrated its critical overarching role in activating actin polymerization. In this revision, we have specified their hierarchical assembly for activating actin polymerization in different scenarios in a curvature-radius-dependent manner.

We believe the current revision has allowed us to address the reviewer's concerns about novelty. We believe that our high-quality quantitative work provides deep new insights into the molecular details underlying the interplay and functional relationships, particularly in the context of fine-tuning mechanisms. The mechanisms reported here represent a substantial contribution beyond simply adding some information to previously reported or mapped components. Identifying the components involved is important, but this should not overshadow the value of a detailed dissection of their interplay in specific contexts. Such analysis is essential for understanding hierarchical regulation across different scenarios and membrane curvatures. We contend that identifying the functional relationships within a multicomponent system—combining basal activity with topologically determined activities—offers profound insight into how molecular interplay facilitates spatiotemporal regulation of actin assembly. This constitutes a significant advancement in the study of actin remodeling, rather than a mere incremental advance. Although previous research has separately explored the roles of BAR domain proteins, Cdc42, and N-WASP in membrane remodeling and actin assembly, our study uniquely integrates these elements into a framework controlled by membrane curvature.

We acknowledge that our first revision did not adequately detail the multicomponent assembly and its functions, both *in vivo* and *in vitro*. Following the reviewer's great suggestions, we conducted a pivotal experiment using an HR1 deletion mutant of FBP17. This allowed us to dissect the combined effects on FBP17 by preventing Cdc42 recruitment while preserving its ability to sense curvature. By integrating Cdc42 with both wild-type FBP17 and the HR1-deletion mutant, our updated results in the second revision elucidated the hierarchical assembly and synergistic roles of FBP17, Cdc42, and N-WASP—insights that required this stepwise dissection and *in vitro* reconstitution to uncover.

Our new results enable us to decode spatially regulated variables, such as membrane curvature gradients, protein–protein interactions within multicomponent complexes, and small GTPase activity—elements that are challenging to isolate in *in vivo* assays alone due to their combinatorial effects. We believe these enhancements have substantially strengthened our work, providing a more robust model of the molecular mechanism. This research provides valuable insights for investigators exploring the coordination of membranes and the cytoskeleton, curvature sensing, and signaling-driven remodeling.

Here we thank the reviewer for the excellent comments to improve the clarity and impact of the manuscript. We also hope the reviewer sees that we have carefully addressed the comments and that our new experiments reasonably support our arguments through substantial work and detailed mechanistic analysis.

(R3Q1): Does Cdc42 interaction with FBP17 influence FBP17 recruitment to curved membranes and its curvature sensitivity properties? This question remains unresolved. Experiments similar to those in Figure 1E (measuring end densities of FBP17 and N-WASP) should be conducted in the presence of Cdc42, as well as with an FBP17 mutant lacking the HR-1/REM-1 domain. Such experiments would clarify whether the interaction between these two proteins is necessary for FBP17's ability to sense membrane curvature. In their study, Ledoux et al. (2023) provided *in cellulo* data using CIP4 mutants (Fig. 4F-I), suggesting that CIP4's interaction with Cdc42 is crucial for its optimal recruitment to plasma membrane nanodeformations. Zhu et al. have an excellent experimental setup that would allow them to test this hypothesis *in vitro* across a broader range of membrane curvatures.

Response:

We thank the reviewer for raising this point regarding the influence of Cdc42-mediated FBP17 recruitment in this curvature-sensing mechanism. To address this, we performed both *in cellulo* and *in vitro* experiments, including conditions using an FBP17 mutant lacking the HR1 domain.

In our *in cellulo* experiments (**Revision Figure 1, new Figure 3H-K**), we analyzed endogenous FBP17 localization in U2OS cells cultured on nanobar arrays under control (DMSO) and Cdc42-inhibited (ML141) conditions. While the overall FBP17 signal was reduced upon Cdc42 inhibition, the characteristic two-dot enrichment pattern—indicative of preferential **localization to high-curvature bar ends—remained across conditions**. The quantitative analysis also showed that **radius-dependent enrichment remained** statistically significant across conditions. These findings indicate that Cdc42 contributes to FBP17 membrane recruitment, but its absence does not abolish FBP17's intrinsic ability to sense membrane curvature, suggesting two different mechanisms.

In parallel, we also observed a reduction in actin polymerization at nanobar ends under ML141 treatment in highly curved regions (<500 nm), suggesting that Cdc42 plays an overarching role in actin assembly by activating N-WASP and by enhancing FBP17–N-WASP clustering on curved membranes. Notably, this result does not refute our proposed mechanism in which curvature sensing activates actin nucleation through the hierarchical assembly of FBP/WASP clusters

locally at a nanometer-sized zone. Rather, it clarifies how distinct multicomponent systems regulate actin nucleation in localized cellular regions, complementing the role of overarching master regulators. Together, these mechanisms provide a precise mechanistic map of hierarchical regulation. By analogy, understanding the spatiotemporal regulation of downstream pathways is as crucial as identifying the initial receptor that triggers the cascade or the key players at intermediate steps.

To further dissect the intertwined mechanisms, we performed *in vitro* protein recruitment assays using supported lipid bilayers on nanobar substrates (**Revision Figure 2, New Figure 4C-H**), testing four conditions: FBP17 alone, FBP17 with Cdc42, FBP17 Δ HR1 mutant alone, and FBP17 Δ HR1 with Cdc42. In the absence of Cdc42, wild-type FBP17 displayed robust curvature radius-dependent binding. This curvature preference is supported by the end-to-side intensity ratios measured across a population of nanobars. Cdc42 not only enhanced the overall membrane association of FBP17 but also promoted its localization to lower-curvature (flatter) regions, indicating that Cdc42-controlled basal levels of FBP17 fine-tune its curvature sensitivity. Consistently, the Δ HR1 mutant was unresponsive to Cdc42, demonstrating that the HR1 domain is essential for Cdc42-mediated enhancement of FBP17 membrane binding. These findings suggest that curvature sensitivity is an intrinsic property of FBP17 but is modulated by Cdc42, likely through the increased membrane recruitment (higher local ‘dose’) of FBP17. Consequently, Cdc42 enables FBP17 to sense curvature earlier *in vivo*, where curvature generation begins at a lower curvature before progressing to a higher-curvature stage. Once higher curvature forms, locally stabilized FBP17 can function more independently of Cdc42 to drive local hierarchical assembly and activation of the nucleation complex. We have now incorporated this newly defined mechanism into the Results, Discussion, and refined model figure (**Figures 3, 4, and 9**).

Revision Figure 1 (New Figure 3H-K)

Figure R1. *In cellulo* analysis of Cdc42-dependent recruitment of endogenous FBP17 and actin to membrane curvature.

(H and J) Representative confocal images of U2OS cells cultured on nanobar arrays and stained for endogenous FBP17 (H) and F-actin (phalloidin, J), under DMSO control or Cdc42 inhibitor (ML141 at 20 μ M and 50 μ M). Scale bar=2 μ m.

(I and K) Quantification of curvature enrichment based on the end intensity of FBP17 (I) and F-actin (K) signals.

Data are shown as mean \pm SEM; n = number of bar ends pooled from multiple cells; statistical significance was determined using one-way ANOVA.

Revision Figure 2 (New Figure 4C-H)

Figure R2. *In vitro* reconstitution of Cdc42- and HR1-dependent regulation of FBP17 membrane recruitment.

(C and F) Representative images of Alexa Fluor 488–labeled FBP17 or its Δ HR1 mutant proteins recruited to SLBs on nanobar arrays under the indicated conditions: FBP17 alone, FBP17 + Cdc42, FBP17 Δ HR1 alone, and FBP17 Δ HR1 + Cdc42. Scale bar=2 μ m.

(D,E and G,H) Quantification of signal end-to-center ratio on bars (D and G) and end signal density (E and H).

Data are shown as mean \pm SEM; n = number of bars from at least 2 independent experiments; statistical significance was assessed by one-way ANOVA.

Comment (R3Q2):

In the rebuttal letter and the revised manuscript (particularly in the discussion), despite the new data highlighting the importance of Cdc42 in inducing proper actin nucleation and polymerization

in a curvature-dependent manner, the authors continue to treat Cdc42 and FBP17 as largely independent factors. This is particularly evident in the schematics of the new Figure 6 and Supplementary Figure 6, where the well-documented interaction between Cdc42 and FBP17 via the HR-1/REM-1 domain is not depicted at all. Given that FBP17, N-WASP, and Cdc42 form a tripartite complex, this omission is significant.

Does it make sense to propose two parallel activation mechanisms for actin nucleation and polymerization? The new Supplementary Figure 6 demonstrates that robust actin nucleation occurs only when the tripartite complex is formed, and is significantly reduced when any one of the partners is absent. Additionally, Figure 4F clearly indicates that the curvature-dependent actin polymerization is strongest when all molecular components are present (condition +F+N+A+C).

Does disruption of the HR-1/REM-1 domain in FBP17 abolish the cooperative/synergistic effect of FBP17 and Cdc42 on actin polymerization on both flat and curved membranes? As mentioned above, experiments using FBP17 mutants lacking the HR-1/REM-1 domain would be highly informative.

Response:

We would like to clarify that, although we propose independent mechanisms for mechanistic dissection supported by distinct biochemical activities, we have never claimed—and did not intend to claim—that the curvature-dependent mechanism operates independently of Cdc42. We fully agree with the reviewer that the relationship between these mechanisms deserves clearer elaboration. Any previous confusion may have arisen from the lack of detailed examples, as also noted by the reviewer. In the current revision, we have addressed this point by incorporating quantitative analysis using the reviewer-suggested Δ HR1 mutant for comparison.

We have updated Figures 4 and 5 to clarify the functional interaction between Cdc42 and the HR-1/REM-1 domain of FBP17. These new data and illustrations addressed the interplay of the tripartite complex (Cdc42, FBP17, and N-WASP) in cooperating actin nucleation.

To examine whether and how disruption of Cdc42-FBP17 interaction compromises curvature-sensitive actin polymerization, we conducted *in vitro* actin assembly assays on both flat and nanobar-patterned SLBs, using a mutant lacking the HR-1 domain (FBP17 Δ HR1, referred to as FdHR1).

First, we found that the Δ HR1 mutant (FdHR1) preserved curvature-dependent binding but showed reduced recruitment at low-curvature bar ends compared to wild-type FBP17 (+F+N+A+C). This indicates that curvature sensing by FBP17 is intrinsic (**Revision Figure 3, new Fig. 4L–M**), although Cdc42 still contributes to FBP17 recruitment to curved membranes. Interestingly, overall actin polymerization at low-curvature regions was reduced, suggesting that Cdc42 recruits more FBP17 to these flatter areas to initiate actin polymerization earlier (**Revision Figure 3, new Figure 4L–N**), consistent with the fine-tuning of FBP17's curvature sensing described above. The presence of active Cdc42 enhances FBP17 recruitment in low-curvature regions, indicating that the tricomponent assembly Cdc42, FBP17, and N-WASP varies by

curvature radius and thereby contributes to actin polymerization differentially following curvature progression during membrane bending. Specifically, Cdc42 plays a more prominent role in low-curvature areas. At low curvature, Cdc42 triggers an early wave of actin polymerization to initiate membrane bending. In contrast, FBP17–N-WASP clusters are most effective at boosting actin assembly once high curvature is present, likely supporting rapid membrane morphogenesis in later stages of curvature development.

Second, using flat SLBs without curvature, we observed that FdHR1 fails to synergize with Cdc42 in promoting actin assembly. Compared to the robust actin nucleation observed with wild-type FBP17, N-WASP, and Cdc42, replacing FBP17 with FdHR1 led to significantly slower actin polymerization (**Revision Figure 4, new Figure 5B–D**).

Together, these results demonstrate that the same tripartite complex—Cdc42, FBP17, and N-WASP—produces distinct effects on FBP17 curvature sensing and FBP17-mediated actin polymerization according to different curvature radii. This supports the idea that their combined activities are precisely regulated to coordinate membrane morphogenesis over time throughout curvature progression, from initial membrane formation to radius expansion.

Revision Figure 3 (New Figure 4 L-N)

Figure R3. The Cdc42–FBP17 interaction is dispensable for curvature sensing but required for maximal actin polymerization in low-curvature regions.

(L) Representative confocal images of actin polymerization on SLBs with nanobar arrays of indicated bar widths (1000–200 nm). Actin polymerization was reconstituted using combinations of FBP17 (F), Cdc42 (C), N-WASP (N), and Arp2/3 complex (A), including the FBP17 HR1-deletion mutant (FdHR1), as labeled. Scale bar: 2 μ m.

(M) Quantification of actin signal intensity at bar ends, normalized to lipid signal intensity, plotted against bar width.

(N) Curvature sensing coefficients for each reconstitution condition, calculated from the slope of end-to-side intensity change over curvature. Different letters indicate statistically significant differences ($p < 0.05$, one-way ANOVA).

Data are shown as mean \pm SD from two independent experiments.

Revision Figure 4 (New Figure 5 B-D)

B

Figure R4. The HR-1 domain of FBP17 is essential for cooperative actin nucleation with Cdc42 on flat membranes.

(B) Time-lapse(left) or representative(right) images of actin polymerization on flat SLBs reconstituted with N-WASP, Arp2/3 (A), and Cdc42, in the presence of either wild-type FBP17 (+F+N+A+C, top) or FBP17 Δ HR1 mutant (FdHR1, bottom). Schematics(middle) illustrate the proposed molecular assembly, highlighting that the HR-1 domain of FBP17 is required for efficient Cdc42 interaction and tripartite complex formation.

(C) Quantification of actin polymerization kinetics over time based on fluorescence intensity measurements. N=4 ROIs.

(D) Initial actin polymerization rates derived from the slope of the curves in (C). N=4 ROIs.

Data are shown as mean \pm SEM from two independent experiments. Statistical significance was assessed using one-way ANOVA; different letters indicate significantly different groups ($p < 0.05$). Scale bars: 5 μ m.

Comment (R3Q3):

New data with ENTH-SH3 (Figure 8A-F): comparison between multivalent interactions via oligomeric SH3 (FBP17) and monovalent interactions via monomeric SH3 (ENTH-SH3). These new data appear somewhat like comparing apples and oranges. The observed differences between FBP17-SH3 and ENTH-SH3 may not be solely attributable to their oligomeric versus monomeric states. Another overlooked factor is that FBP17 can interact with Cdc42, whereas ENTH cannot. Moreover, literature suggests that certain ENTH domains interact with RhoGAPs in yeast, which in turn catalyze Cdc42 inactivation (<http://dx.doi.org/10.1073/pnas.0510513103>). This raises the possibility that ENTH might inhibit Cdc42 and actin polymerization. Consequently, ENTH-SH3 does not seem to be an ideal control.

To obtain a clearer picture, an important additional control would be a mutated version of FBP17 lacking the HR-1/REM-1 domain, thereby preventing its interaction with Cdc42. If actin assembly is impaired under these conditions, it would suggest that the formation of the FBP17/Cdc42/N-WASP tripartite complex is required and may be more critical than the oligomerization question. Further, examining FBP17 point mutants that disrupt dimerization/oligomerization while preserving membrane-binding capacity could provide additional insight, though identifying such mutants might be challenging.

Response:

We agree with the reviewer's comments on ENTH-SH3. In light of this, we have now used the HR-1 domain strategy to address this point and removed the ENTH-SH3 data.

The concept of the FBP17/Cdc42/N-WASP tripartite complex alone does not explain the full molecular mechanisms by which their combination and hierarchical assembly dynamically regulate curvature radius-dependent actin polymerization. Their joint function relies on oligomerization, which is central to understanding how FBP17-N-WASP clustering exerts distinct effects depending on topology, curvature, and dosage under Cdc42 control. Thus, both the tripartite relationship and clustering represent two interdependent but coupled layers essential for their interplay and function.

Here, we conducted additional experiments using two FBP17 mutants: (1) FBP17 Δ HR1, which lacks the HR-1 domain required for Cdc42 interaction, and (2) a FBP17 point mutant (K166A), designed to disrupt F-BAR domain tip-tip interaction and thereby reduce higher-order assembly (Frost, Perera et al. 2008) while preserving membrane-binding capacity. Both mutants were tested in U2OS cell-based assays. The FBP17 Δ HR1 mutant showed impaired actin assembly, supporting the roles of Cdc42 in recruiting FBP17 and activating N-WASP. Additionally, the K166A mutant also exhibited reduced actin polymerization, highlighting the importance of FBP17 multivalent clustering in promoting efficient actin nucleation. Notably, these effects were observed in the presence of Cdc42, allowing us to dissect the contributions of Cdc42 versus the curvature-coupled FBP17-N-WASP clustering mechanism.

The above-mentioned mechanisms and their interplay are now presented in the revised manuscript in **Figure R5/Figure 8 and Figure 9**.

Revision Figure 5 (New Figure 8 B-K)

Figure R5. Multivalent interactions via Cdc42 binding and F-BAR oligomerization are required for efficient N-WASP recruitment and actin assembly at curvature sites.

(B) Schematic of FBP17 domain structure, the Cdc42-binding-deficient truncation (FdHR1), and the oligomerization-deficient mutant (K166A). FBP17 contains an F-BAR domain, an HR1 domain for Cdc42 binding, and an SH3 domain for PRM binding to N-WASP.

(C–E) Single-molecule imaging of wild-type FBP17 and K166A. Representative images (C), quantification of total intensity (D) and frequency distribution of cluster intensities (E).

(F–H) U2OS cells cultured on nanobar arrays and stained for FBP17 variants (F), N-WASP (G), and F-actin (phalloidin, H).

(I–K) Quantification of nanobar end enrichment for FBP17 (I), N-WASP (J), and F-actin (K).

Data is presented as mean \pm SEM from 2 independent experiments. Statistical significance was determined using unpaired two-tailed t-tests or two-way ANOVA; * $p < 0.05$, ** $p < 0.01$, *** $p < 0.001$, **** $p < 0.0001$. Scale bars: 2 μ m.

Comment (R3Q4):

Lack of in cellulo data: While the authors have included some additional cellular data, they remain limited. More in cellulo validation of their in vitro findings would significantly strengthen the manuscript, although they considered these as out of scope of their manuscript in their previous answers (i.e., siRNA treatment against their molecular players).

Response:

We thank the reviewer for encouraging additional in-cellulo validation, and in response, we performed siRNA-mediated FBP17 knockdown in U2OS cells to quantify actin enrichment at nanobar ends. Despite reduced FBP17 expression, the residual FBP17 signal still exhibited curvature-dependent localization, indicating that its intrinsic curvature sensitivity is preserved. However, we observed both reduced overall actin intensity and attenuated curvature-dependent actin enrichment, suggesting that FBP17 is required not only to amplify actin nucleation but also to enable full curvature-coupled actin assembly in cells (**Figure R6**).

To further support the role of Cdc42, we also performed pharmacological inhibition using ML141 (see **Revision Figure 1**). Together, these in cellulo data align well with our *in vitro* findings, demonstrating that cooperative assembly of the FBP17–Cdc42–N-WASP complex is essential for curvature-sensitive actin nucleation. These experiments revealed that disruption of any single component of the tripartite complex (FBP17, Cdc42, or N-WASP) results in a marked reduction in actin polymerization, thereby reinforcing the physiological relevance of our proposed mechanism. We believe this set of experiments provides a robust cellular validation of our mechanistic model.

Revision figure 6

Figure R6. siRNA knockdown of FBP17 reduces actin assembly while preserving curvature sensitivity in cells.

(E) Representative confocal images of U2OS cells transfected with negative control (NC) siRNA or FBP17-targeting siRNA (siFBP17), immunostained for endogenous FBP17, and cultured on nanobar arrays of indicated widths (1000–300 nm).

(F) Quantification of FBP17 bar-end intensity showing effective knockdown of FBP17 while maintaining curvature-dependent enrichment in residual signal.

(G) Representative images of phalloidin-stained F-actin in NC siRNA and siFBP17-treated cells.

(H) Quantification of actin bar-end intensity showing reduced overall actin assembly and attenuated curvature-dependent enrichment following FBP17 knockdown.

Data are presented as mean \pm SEM from multiple cells pooled across 2 independent experiments. Scale bars: 2 μ m.

5. Additional minor note: The ENTH-SH3 expressing plasmid is not described in Table S3. From which protein is the ENTH domain derived?

Response:

We have removed the ENTH-SH3 data and related text/figures. Accordingly, the ENTH-SH3 construct no longer appears in Table S3.

6. Minor comment on Figure 4F: Why is the Y-axis truncated and not continuous?

Response:

We thank the reviewer for pointing this out. The Y-axis in previous Figure 4F (now Figure 4M) is continuous, and the axis labels were simply split into two parts for clarity in presentation. To avoid any misunderstanding, we have revised the figure legend to explicitly state that the Y-axis is continuous: “Normalized actin signal intensities relative to corresponding lipid bilayer intensities at nanobars of 200–1000 nm width. *Y-axis values are continuous, but axis labels are shown in two parts for clarity.* Sample sizes were N = 9, 9, 9, 9, 8 and 11 averaged nanobars, respectively. Data are presented as mean \pm SEM.”

In summary, this revised version of the manuscript is significantly improved, but several key aspects, particularly regarding the interplay between CDC42 and FBP17, still warrant further investigation. Addressing these points would enhance the overall impact and mechanistic depth of the study. In addition, while the novelty of the findings is moderate, the substantial amount and high quality of the data make this work potentially valuable to the community.

Response:

We appreciate the reviewer's constructive summary and thoughtful evaluation of our work.

We want to emphasize that we are also truly grateful for the new suggestions, such as using an FBP17 Δ HR1 mutant, which is a powerful approach that not only allowed us to address the concerns but also to uncover more detailed mechanisms. This enabled us to better define the interplay of the FBP17/Cdc42/N-WASP complex with respect to FBP17 recruitment levels, membrane curvature, and the role of FBP17 oligomerization, thereby improving clarity and enhancing the significance of our study.

REFERENCES

Frost, A., R. Perera, A. Roux, K. Spasov, O. Destaing, E. H. Egelman, P. De Camilli and V. M. Unger (2008). "Structural basis of membrane invagination by F-BAR domains." Cell **132**(5): 807-817.

Dear Yansong,

Thank you for submitting a revised version of your manuscript. I have now gone through the incorporated revisions, and I find that they sufficiently address the remaining concerns by reviewers #1 and #3. There now remain only a few editorial points that need to be addressed before I can extend official acceptance of the manuscript:

1. Please submit a complete author checklist, which you can download from our author guidelines (<https://www.embopress.org/pb-assets/embo-site/EMBO%20Press%20Author%20Checklist-1642513524327.xlsx>). Please insert information in the checklist that is also reflected in the manuscript. The completed author checklist will also be part of the Review Process File.
2. Please correct the order and the headings of the manuscript sections to: Abstract / Keywords / Introduction / Results / Discussion / Methods / Data Availability / Acknowledgements / Disclosure and Competing Interests Statement / References / Figure Legends.
3. Please upload the manuscript text in .docx format. All figures should be uploaded as separate, high resolution figure files. Please include the figure legends in the manuscript text file after "References". Please rename the suppl. Figures into Figure EV1 - EV7. Their legends should be after the main figure legends, under the heading "Expanded View Figure Legends".
4. Please include funding sources in the "Acknowledgements" section.
5. CRedit has replaced the traditional author contributions section because it offers a systematic, machine-readable author contributions format that allows for more effective research assessment. Please remove the Authors Contributions from the manuscript and use the free text boxes beneath each contributing author's name in our online submission system to add specific details on the author's contribution. More information is available in our guide to authors.
6. Please rename "Competing interests" section into "Disclosure and competing interests statement" (further info: <https://www.embopress.org/page/journal/14602075/authorguide#conflictsofinterest>)
7. Please remove the supplementary tables from the manuscript text, upload them as separate files and rename them into Table EV1 - EV3.
8. Please rename movies into Movie EV1-EV4 and update the callouts accordingly. The legends should be removed from the manuscript text file and zipped with each movie file. Further information is available here: <https://www.embopress.org/page/journal/14602075/authorguide#expandedview>
9. Figure panels 8C-E are not mentioned in the manuscript text; please add the corresponding callouts.
10. When submitting your revised manuscript, please do not include the Reagents and Tools Table in the Methods section of the manuscript but upload it as a separate file choosing the file type "Reagent Table".
11. At EMBO Press we ask authors to provide source data for the main manuscript figures. You will receive a separate email with instructions for providing source data with your revised manuscript, including how to upload and organise the files.
12. Papers published in The EMBO Journal are accompanied online by a 'Synopsis' to enhance discoverability of the manuscript. It consists of A) a short (1-2 sentences) summary of the findings and their significance, B) 3-4 bullet points highlighting key results and C) a synopsis image that is 550x300-600 pixels large (width x height, jpeg or png format). You can either show a model or key data in the synopsis image. Please note that the image size is rather small and that text needs to be readable at the final size.

Please note that our data editors are still working on the figure legends, and I will share their requests in a follow-up email.

With best wishes,

Ieva

We realize that it is difficult to revise to a specific deadline. In the interest of protecting the conceptual advance provided by the work, we recommend a revision within 3 months (26th Jan 2026). Please discuss the revision progress ahead of this time with the editor if you require more time to complete the revisions.

The authors addressed the remaining editorial issues.

Dear Yansong,

Thank you for addressing the remaining editorial points. I am now pleased to inform you that your manuscript has been accepted for publication.

Before we forward your manuscript to our publishers, we would like to propose some edits in the manuscript title, abstract and synopsis (please see the attached file). We have also prepared a short blurb that will accompany the title of your manuscript in our online system. Please take a look and let me know if any corrections are needed.

You may qualify for financial assistance for your publication charges - either via a Springer Nature fully open access agreement or an EMBO initiative. Check your eligibility: <https://link.springer.com/journal/44318/how-to-publish-with-us>

If you have any questions, please do not hesitate to contact the Editorial Office. Thank you for this contribution to The EMBO Journal and congratulations on a nice study!

With best wishes,

Ieva

Please note that it is The EMBO Journal policy for the transcript of the editorial process (containing referee reports and your response letters) to be published as an online supplement to each paper. If you should prefer removal of any referee-only figures included in the point-by-point response(s), e.g. because they may still be used for future publication or because they have been reproduced from published work by others, please do let us know immediately via response email.

More information is available here: <https://link.springer.com/partners/embo-press/editorial-policies#Peer%20review>